# How Human Trafficking Fuels Erosion of Liberal Democracies—In Fiction and Fact, and from within and without

**Jill E.B. Coster van Voorhout**

Faculty of Law, University of Amsterdam, 1018 WV Amsterdam, The Netherlands; j.costervanvoorhout@uva.nl

**Abstract:** On the same day that the human trafficker Ms. Ghislaine Maxwell was sentenced to 20 years' imprisonment, many people closely watched the sixth hearing of the House Select Committee on the attack of the United States Capitol on 6 January 2021 (28 June 2022). What, if anything, do these ostensibly varied crimes have in common? Seeking to answer this fundamental question, this article explores the usually under-researched connection between trafficking in persons and the documented decline of liberal democracies worldwide. Globally, democratic societies governed by the rule of law appear to be under assault, and therefore this article explores relevant examples of how human trafficking contributes to the erosion of liberal democracy, in fiction and fact, and from within and without. In other words, this article takes us from 'Pizzagate' to profits.

**Keywords:** human trafficking; organized crime; erosion of democracy and the rule of law; organized subversive crime

## 1. Introduction

'If slavery is not wrong, nothing is wrong', as Abraham Lincoln stated.[1] The wrongs of human trafficking, nowadays also oftentimes known as modern-day slavery, are in this Special Issue examined in the context of the purported crisis in liberal democracies around the world (e.g., Diamond 2019; Economist Intelligence Unit 2021; Mounk 2019; Levitsky and Ziblatt 2018; Guriev and Treisman 2022).[2] Today's apparent global democratic deconsolidation has been attributed to everything from the influence of polarization, economic inequality, social media to white male rage. As valid as these points may be, few of their analyses fully consider the deep historical roots of what types of crime advance the erosion of liberal democracies and the extent to which trafficking in persons plays a role. An in-depth analysis of both human trafficking and the erosion of democracy and the rule of law worldwide is complex because their essential elements are difficult to detect due to their mostly hidden nature (e.g., Shelley 2012; Levitsky and Ziblatt 2018; Snyder 2018; Coster van Voorhout 2020). Therefore, to study all these issues comprehensively, this article takes every relevant angle needed: human trafficking in fiction and fact, and corrosion from within and without a liberal democracy or, in this interconnected world, multiple liberal democracies. From all four angles, this article seeks to answer the following research question: Should some forms of human trafficking be dealt with as a new and unique category of crime or accompanied by an aggravating circumstance that emphasizes its corrosive effect on liberal democracy and, if so, under what conditions would that be useful in law and in research?[3]

The methodology used to answer this central research question is a combination of normative ('how should it be?') and, to a limited extent where taken from other predominantly doctrinal sources, empirical research methods ('what is it?'), while social scientific, historical, and political scientific literature is referred to where relevant. One reason for stressing the importance of needing to obtain empirical findings in future research is the criminological research finding that police interventions that are founded only in theoretical assumptions risk contributing to strengthening criminal networks instead of weakening

them (e.g., Luna-Pla and Nicolás-Carlock 2020; Thurner et al. 2018; De Domenico et al. 2019; Helbing et al. 2015; van Engers et al. 2018; Duijn and Sloot 2015; Duijn et al. 2014). The predominantly doctrinal analysis in this article therefore points out ways to find useful empirical insights in further research, so as to assist a more data-driven response and draft, as an ultimate aim, an agenda for resilience for liberal democracies.

The substantive emphasis of this article is on criminal law, with a conceptual and geographic focus on the Netherlands, the United States (US), and Russia. Starting with the Netherlands, Dutch scholars and policymakers are currently seeking to develop a new concept that addresses the challenge of crime that erodes its established liberal democracy (e.g., Kruisbergen et al. 2021; Tjeenk Willink 2021; Tops and Tromp 2017; Boutellier et al. 2019; Huisman and Kleemans 2017). The Dutch did not (yet) face an outright violent attack on its democracy and rule of law such as the storming of the US Capitol on 6 January, 2021. However, just as the Americans, the Dutch struggle with finding adequate solutions for, inter alia, high-impact murders that damage the trust of the general population in its relatively consolidated democratic society governed by the rule of law. While American scholars and policymakers call for dealing more adequately with January 6th by introducing a not yet existent domestic terrorism charge, the Dutch are trying to devise a concept that delineates how a crime can be committed with the intent to, or effect of, subverting a liberal democracy.[4] This article therefore explores to what extent this solution sought by the Dutch could be helpful in other contexts such as in the United States. This article does so, by focusing on a crime that has not been fully examined in relation to this new Dutch concept either: human trafficking. The relevance of this novel Dutch concept for human trafficking is examined in two ways, moreover. First, this article explores whether this new Dutch concept should or should not encompass human trafficking and, if so, in what manner. Second, examined is whether a focus on human trafficking can help us fundamentally understand the more systemic issues of criminality that advances the erosion of liberal democracies by determining who are (forcibly) involved. Turning as a final issue to the third country included below, Russia, this state is mentioned because of its export of conspiracies including about human trafficking to liberal democracies (e.g., Snyder 2018; Miller-Idriss 2022). While this spreading of disinformation that attacks the truth in and about liberal democracies calls for a further examination of Russia, this country is also relevant in relation to the crime of human trafficking itself. For example, on 19 July 2022, Russia was put on the state-sponsored human trafficking list, thus indicating an often-overlooked aspect of trafficking in persons not committed by organized criminal groups but by the state itself (TIP report 2022).[5] Additionally, Russia's fuller invasion of Ukraine has caused increasing (internal) migration, thus making more persons vulnerable to exploitation by both state-run and organized crime group-led human trafficking. To conclude and in order to make the limitations of this article explicit, because of its limited scope, fields of law other than criminal law and countries other than the Netherlands, United States and Russia will have to be reserved for future research.[6]

*Roadmap* This article is divided into six parts. First, a description follows of how the abuse of the phenomenon of human trafficking by conspiracists contributes to the undermining of democracy and the rule of law (Section 2). In the subsequent part, this article turns from fiction to fact, by focusing on whether the actual crime of human trafficking can have a similar corrosive effect (Section 3). Thereafter, a section is dedicated to the potential interplay between fiction and fact and internal and external corrosion (Section 4). Thereupon, the penultimate section deals with the usefulness of this new Dutch concept of 'crime that undermines democracy and the rule of law' for dealing with all these problems (Section 5). Finally, a conclusion is drawn and recommendations for further research are made (Section 6).

## 2. Problem 1: Conspiracies about Human Trafficking

Conspiracists spread false claims about human trafficking like wildfire, as the two most influential conspiracies of our time, Pizzagate and QAnon, demonstrate. To put

it in all too familiar COVID-19 terms, human trafficking may very well be the present-day 'patient zero' of disinformation. Disinformation is known to sow distrust in liberal democracies by attacking the truth[7] and eroding trust in public and private institutions and their representatives.[8] Such attacks on the truth complicate the understanding of the general public of what actually constitutes trafficking in persons and what does not. Similarly, claims that state actors fail to deal with what are false claims about human trafficking can also encourage a feeling among the general public that their state is lacking or at least ineffective in its response. The fact that human trafficking is such a popular theme for conspiracists raises a number of questions. How do they abuse human trafficking in their conspiracies? Do they focus on a specific form of human trafficking and, if so, why do they do so? What are some of the effects that we can empirically notice? Do these conspiracies indeed contribute to the erosion of liberal democracy, for example because they motivate real-world, physical violence?

### 2.1. Conspiracies about Human Trafficking including Pizzagate and QAnon

Before describing in more detail the conspiracies of Pizzagate and QAnon, it is important to note why this article deliberately refers to conspiracies rather than conspiracy theories (e.g., Miller-Idriss 2022). As Ms. Cynthia Miller-Idriss delineates, a conspiracy theory would offer some explanations, while a conspiracy 'only' raises suspicion oftentimes without offering any such edification as to why claims would take place. She explains: 'And the new conspiracies are different in nature from the ones that came before. They are a 'new conspiracism', as the scholars Russell Muirhead and Nancy Rosenblum describe it: conspiracy without the theory. Classic conspiracism helps ordinary people making sense of a 'disorderly and complicated world' by taking events that seem to defy explanation (like the 9/11 attacks or John F. Kennedy's assassination) and framing them as the organized work of a group of powerful people. But the new conspiracism 'dispenses with the burden of explanation', relying on innuendo, suggestion, and repetition—with legitimation of conspiracies and accusations built through retweets, forwards, reposing, and *social* validation'. (Miller-Idriss 2022, p. 57). Today, conspiracists resort more to conspiracies than to conspiracy theories. This is a disturbing trend, especially because Miller-Idriss finds and reports on empirical evidence of right-wing conspiracies having gone mainstream (cf. Riggleman and Walker 2022). This mainstreaming makes it more difficult to trace fictions about crimes like human trafficking in, and distinguish them from other opinions in, general popular culture. In the criminal case concerning the 'Unite the Right' rally held in Charlottesville on 11 and 12 August 2017, the court accepted the expert testimony of Professors Blee and Simi that 'various white supremacist groups often utilize strategies of deniability, including 'double-speak', as experts describe it; outsiders would only hear an innocuous comment or think the speaker was making a tasteless joke, while those 'in the know' would understand the hidden meaning'.[9] This same 'double-speak' is found in relation to the two conspiracies Pizzagate and QAnon.

Pizzagate is the name of the conspiracy that United States' Democratic representatives traffic persons especially children from pizza restaurants. Pizzagate first went viral during the 2016 US presidential election cycle. On social media outlets, conspiracists falsely claimed that emails from the campaign chair of then candidate Ms. Hillary Clinton, Mr. John Podesta, contained coded messages. The emails, which had been published by Wikileaks, for instance referred to cheese pizza (c.p.) which would allegedly stand for child pornography (c.p.). Deriving the last part of its name from Watergate, supposedly the New York City Police had discovered the human trafficking ring, which had even allegedly been confirmed by the Federal Bureau of Investigations, but the Democratic elite supposedly covered it up. This conspiracy did not only reach and perhaps influence many voters in the US, but it also linked to actual, physical violence. In response to Pizzagate according to the convicted person himself, in 2017, a gunman stormed a Washington, DC, pizza restaurant Comet Ping Pong to save children he believed were being held in a basement by this alleged human trafficking ring run by Democratic operatives. He traveled from North Carolina to

the US capital to investigate the conspiracy and fired a rifle inside the restaurant to break the lock on a door to a storage room during his search (Kang and Goldman 2016). The pizza restaurant did not have a basement. The restaurant owner and staff also received death threats from conspiracists (Kang 2016). In interviews to the press, the convicted gunman never admitted that Pizzagate was a conspiracy and thus apparently is still held in its grip today, or perhaps even in its newer iteration: QAnon (Goldman 2016; Abrams 2016).

The QAnon conspiracy was borne out of Pizzagate (e.g., Simon Wiesenthal Center 2020; Kang and Frenkel 2020). Pizzagate resurged in 2020, mainly due to QAnon. While initially it was spread by only the far-right, it has since been widely circulated on TikTok by teens who do not appear to hold that belief (Sommer 2020). More recently, the conspiracy has developed and become less political and partisan in nature, with less emphasis on Ms. Hillary Clinton and more on the alleged worldwide elite of child sex-traffickers (Simon Wiesenthal Center 2020; Kang and Frenkel 2020; Tian 2021). QAnon seemingly evolved on 4chan, and later 8chan (also known as 8 kun), building on some of the themes developed by Pizzagate and the discussions online that followed. 4chan and 8chan are anonymous online forums (imageboards) where users can post messages and images without having to register for an account. A person or group posted messages as 'Q' on all these mostly unregulated fora. The QAnon conspiracy concerns a global cabal that is supposedly responsible for the most urgent world problems such as crime, war, poverty and disease. Former US President Trump was presumed to wage a secret war against these global human traffickers and others involved in the so-called 'deep state'. Forming the top of the QAnon conspiracy hierarchy are the Rothschild family, the Saudi Royal Family and hedge fund manager and initiator of international pro-democracy and pro-rule-of-law non-governmental organizations Mr. George Soros. 'These three so-called 'puppet masters' purportedly control the deep state with a vast combined wealth which they use to manipulate political bodies and financial institutions around the world. They are supposedly aided by collaborators and puppets. These include prominent liberal, primary Democratic, political figures such as Bill and Hillary Clinton, Barack and Michelle Obama, Hollywood icons including Oprah, Ellen DeGeneres, Tom Hanks and Rita Wilson, and religious figures, including the Pope. They face accusations including operating child sex trafficking rings, and practicing satanic rituals' (Simon Wiesenthal Center 2020, p. 3). According to the Public Religion Research Institute think tank, the share of Americans who believe in the main tenets of the QAnon conspiracy movement has increased slightly from 14% in March 2021 to 18% in March 2022 (PRRI 2022). Some 18% think violence might be necessary to save the country and 16% hold that the government, media and financial worlds are controlled by Satan-worshipping pedophiles, according to four surveys carried out in 2021 (*Ibid.*). In the Netherlands, QAnon has an estimated 12.000 followers and some of them were found to partake in anti-COVID measure protests as well as accuse local politicians of sex crimes (NOS 2022).[10]

During the storming of the Capitol on January 6th, 2021, several persons were wearing QAnon t-shirts and brought other such paraphernalia. As Miller-Idriss argues: 'But the past year also has been a lesson for us all about the fact that ordinary people do not only encounter extreme ideas in their everyday lives—they can also be mobilized to violent action by those extreme ideas—in ways that affect the stability of our democracy. After months of widespread propaganda and disinformation about an illegitimate US presidential election, thousands of Americans violently stormed the US Capitol on 6 January 2021, in a spontaneous coalition that united unlawful militias, white supremacist extremists, QAnon conspiracy theorists, Proud Boys, and ordinary Trump voters. They mobilized around one central objective: to thwart a democratic process certifying the presidential election' (Miller-Idriss 2022, p. xiv). Currently, there are still criminal and other investigations ongoing into whether it was indeed a spontaneous coalition or rather a much more organized seditious real-life (not fictional) conspiracy. But before examining how such investigations should be conducted (See Section 5), it is important to give four explanations as to how conspiracies about human trafficking already appear to advance the corrosion of liberal democracies, as follows.

## 2.2. Four Explanations for the Corrosive Effect of Conspiracies about Human Trafficking

First, the conspiracy about a global cabal that traffics vulnerable people, especially women and children, for the purpose of sexual exploitation is reminiscent of the equally widely disclaimed 'Protocols of the Elders of Zion', a Russian falsification that gained traction in Nazi Germany.[11] The historian Mr. Timothy Snyder who conducts historical research into the connection between Russia and its influence on the decline of liberal democracy in Europe and the United States directly ties Pizzagate to attempts to erode democracy and the rule of law in the US through decades-long efforts by President Putin of Russia. Snyder explains: '(o)n October 2016, Trump seemed to be in trouble when a tape revealed his view that powerful men should sexually assault women. Thirty minutes after that tape was published, Russia released the emails of the chairman of Clinton's campaign, John Podesta, thereby hindering a serious discussion of Trump's history of sexual predation. Russian trolls and bots then went to work, trivializing Trump's advocacy of sexual assault and guiding Twitter users to the leak' (Snyder 2018, p. 233). In an attempt that is also known as 'whataboutism', one man's allegations of actual sexual crimes are countered by spreading false claims about non-existing sex crimes. As of 2012, according to Snyder, President Putin has been seeking to undermine liberal democracies by spreading such disinformation so as to distract from his oligarchical rule at home.[12] That 'rule by the wealthy few' is based on, primarily, its income from oil and gas, and, secondarily, its policy or pattern of human trafficking in its (il)legitimate economy (TIP report 2022).[13] The spreading of disinformation ensures capturing of attention of the general public, also oftentimes as a distraction from problems in Russia, which has always been intertwined with the money needed for the oftentimes targeted persuasion of the electorate, especially to ensure positive results in electoral politics.[14]

Second, as for the sexual nature of Pizzagate and QAnon, Mr. Jason Stanley explains in his chapter on sexual anxiety in his book on how fascism works that: '(i)f the demagogue is the father of the nation, then any threat to patriarchal manhood and the traditional family undermines the fascist vision of strength. These threats include the crimes of rape and assault, as well as so-called sexual deviance. The politics of sexual anxiety is particularly effective when traditional male roles, such as that of family provider, are already under threat by economic forces' (Stanley 2018, p. 127). Comparably, Snyder argues that '(h)uman sexuality is an inexhaustible raw material for the manufacture of anxiety' (Snyder 2018, p. 51; Goldenberg et al. 2021). While neither Stanley nor Snyder explains that emphasis on sexuality in relation to human trafficking and its potential effect on the global decline of liberal democracies, their arguments are relevant for both topics that are central to this article. Several conspiracies center on grooming, for example, using the seed of truth of an actual criminal case like the charge of the co-conspirator of the indicted but deceased sex trafficker Mr. Jeffrey Epstein, Ms. Ghislaine Maxwell, and giving it the more ominous and less facts-based connotation in conspiracies about this crime.[15] On the issue of disinformation, Snyder describes President Putin's attempts to offer 'masculinity as an argument against democracy' (Snyder 2018, p. 53). President Putin does so by placing 'heterosexuality within Russia and homosexuality beyond' which is, of course 'factually ludicrous, but the facts were beside the point' (Snyder 2018, p. 51). The concept of fake news and disinformation are inventions from Russia that traveled from east to west (Snyder 2018, p. 11). This emphasis on sexuality also offers a further explanation than the one provided above by Miller-Idriss for the topic of this article, human trafficking, because it is quite noticeable that conspiracists emphasize sex trafficking rather than any other forms. Those other forms of human trafficking include labor trafficking, the forced commission of crimes, forced services like domestic servitude or, where not criminalized, forced begging, and for the purpose of organ removal, as will be dealt with in more detail in the next section (Shelley 2013; Coster van Voorhout 2009, 2020). According to Snyder, President Putin seems to have resurrected 'Leonid Brezhnev's permanent enemy'—'the decadent West had returned: but this time the decadence would be of a more explicitly sexual variety' (Snyder 2018, p. 53) This method of spreading disinformation as to how sex trafficking is supposedly rampant

in liberal democracies indeed seems to emphasize the purported depraved nature and decadency of the West. Consequently, Russians and others under the Putin government's influence may not find democracy and the rule of law an attractive alternative to their current rule. Moreover, if citizens in liberal democracies believe these conspiracies, they lose trust in their own state and, as a corollary, may no longer be able to distinguish fiction from fact about both human trafficking and democracy and the rule of law.[16] Such confusion also contributes to a loss of trust in both public and private institutions as well as their representatives, especially those charged with legislating anti-human trafficking efforts and those tasked with investigating and adjudicating truthfully in criminal cases such as investigators, banks that report unusual transactions about related money laundering, prosecutors, (where available in the legal system) victim representatives, and judges.

Third, this conspiracy-based creation of anxiety and attempt to keep people living in the moment with no view of a better future no longer remains merely passive. For example, like Snyder, Miller-Idriss explains: 'Language that positions white women or children as especially in need of protection or defense has proven to be highly effective at mobilizing far-right violence and action', such as the aforementioned gunman storming the pizza restaurant Comet Ping Pong and possibly the attack on the husband of the Speaker of the House Mr. Paul Pelosi (Miller-Idriss 2022, pp. 53–54). She adds: 'Similar calls urge whites to defend white territory and fight against 'criminal migrants' (Miller-Idriss 2022, pp. 53–54).[17] No longer do the consumers of conspiracies avert violence; now they actively storm restaurants and the US capitol[18] in what nowadays may even be deemed a 'holy battle versus good and evil' (Riggleman and Walker 2022). With claims of 'save the children' and killings to prevent the false so-called 'great reset' in which a supposed world elite is executing a secret plan to replace the white race through mass migration, this has already led to actual violence in Norway and Germany, for example.[19]

Finally, Snyder warns us for 'the totalitarian implications of the selective public release of private communications' (Snyder 2018, p. 233). He explains: 'Totalitarianism effaces the boundary between the private and public, so that it is normal for us all to be transparent to power all the time. The information that Russia released concerned real people who were serving important functions in the American democratic process; its release to the public affected their psychological state and political capacity during an election. It mattered that people who were trying to run the Democratic National Convention were receiving death threats over cell phone numbers that Russia had made public'. (Ibid.) Snyder adds: 'As in Poland in 2015, so in the United States in 2016: no one considered the totalitarian implications of the selective release of private communications'. (Ibid.) Also he concludes '(a)ll of this mattered at the highest level of politics, since it affected one major political party and not the other. More fundamentally, it was a foretaste of what modern totalitarianism is like: no one can act in politics without fear, since anything done now can be revealed later, with personal consequences'. (Ibid, p. 234). This particular warning about the way in which false claims about human trafficking are based on leaked private information so as to dissolve the distinction between the private and public realm is relevant.[20] Such conspiracies or the potential of making allegations of sex crimes that the public finds hard to distinguish from fact indeed make state officials live in fear over having their private information made public and abused in the form of false allegations of their involvement in sex crimes.

### 2.3. Conclusions

To conclude, conspiracies about human trafficking that seemingly have gone global already appear to contribute to democratic deconsolidation worldwide because of at least four reasons: (i) they are an attack on the truth which makes it harder for the general population to distinguish fact from fiction on topics like human trafficking and comparable issues that are essential for a facts-based understanding of democracy and the rule of law; (ii) they scandalize democracy, democratic leaders and citizens through sex crime allegations; (iii) they mobilize persons to commit violence; and (iv) they seek to erase the

division between the public and private realm well-known in totalitarian rule. Summarily put, conspiracies about human trafficking already seem to advance the erosion of liberal democracies because they involve fundamental assaults on the truth[21], mobilization for violence, and risks for gradually and insidiously introducing totalitarianism. But before explaining further how to deal with these conspiracies about human trafficking, it is important to turn to the question whether the real-world offense of trafficking in persons can have a comparable corrosive effect.

### 3. Problem 2: Actual Human Trafficking Cases

Today's perhaps best-known human trafficking case around the world, the yet referenced Epstein case, prompted a bipartisan group of four US Senators to conclude:

> 'These events have ignited a crisis of public trust in the Department [of Justice] and exacerbated the erosion of trust that the American people have in our institutions of republican self-government more broadly'.[22]

These Senators thus stipulate that the –dealing with the– Epstein case caused the general public to lose the trust that is essential for the social contract of a liberal democracy due to damage done to the integrity of its public –and perhaps also private– institutions and their representatives. If their conclusion is correct, human traffickers like Epstein indeed go beyond exploiting state structures for financial or other gain, by rather eroding them. The fact that human trafficking can potentially have such a corrosive effect on a liberal democracy, much like the aforementioned conspiracies about trafficking in persons like Pizzagate and QAnon, prompts a series of questions. Do some forms of human trafficking indeed create this sense among the general population of losing trust in the state? What are some of the corroding factors that we can empirically notice? Is this corrosive effect sufficiently examined, both in absolute terms and in relative terms – the latter meaning compared to studies of the aforementioned conspiracies–, and, if not, what could explain this (difference)?

*3.1. Human Trafficking Cases including the Epstein Case*

Before describing the Epstein case and its corrosive effect in more detail, it is important to first address global facts and data about human trafficking because it is oftentimes wrongly assumed that this crime only affects a few states around the globe. Comparably, it is a common misunderstanding that, given that both the aforementioned conspiracies of Pizzagate and QAnon and the Epstein case concern sex trafficking, this is the only form of human trafficking.

First, trafficking in persons occurs in every region of the world. For example, about 50,000 human trafficking victims were detected and reported by 148 of 155 reporting countries in 2019 (UNODC 2020). States can be the origin, transit or destination country for victims, or even a combination of all (Ibid.).

Second, human trafficking covers an enormous spectrum, ranging from the sex sphere to business sectors like agriculture, restaurants, transportation, forced commission of –drug or other– crimes or begging and organ removal (Shelley 2013; Coster van Voorhout 2009, 2020). The commodities produced by trafficked persons range from clothing to electronics and food, and the profits made by their sale are often not easily distinguished from licit flows of money.

Last, human trafficking is a highly complex crime[23]; a fundamental human rights violation[24]; and an offense that cannot always easily be distinguished from other crimes like migrant smuggling—if a migrant first consents to illegal entry but must pay off the travel debt through forced (sex) labor, for example (e.g., Aronowitz 2009; Cho 2015; Campana and Varese 2016; UNODC 2020).

Before describing more details of the Epstein case, it must be emphasized that this description is inevitably hindered by the fact that human trafficking in all of its manifestations is a predominantly hidden crime. The same accounts for (human-trafficking related) money

laundering and corruption. For both the Epstein case but also more generally, this limits our understanding about all such (connected) offenses. This refers to what in criminological literature is known as the *dark figure* (Biderman and Reiss 1967; Skogan 1977). Hampered by the fact that only few human trafficking (related) crimes get reported, we may never know how many trafficking in persons and/or related financial and economic crimes there really are and how many offenders, enablers, and victims are involved (Cf. Cruyff et al. 2020).[25]

Another important complicating factor for our full understanding of the crime of human trafficking is that few countries in the world have prioritized either its prosecution or confiscation of assets (ILO 2014; Shelley 2012, p. 242; Coster van Voorhout 2020).[26] As a consequence, human trafficking is a low-risk high-profit offense. This is problematic, because human trafficking is an estimated $150 billion 'industry' worldwide.[27] Although such approximations are hard to make, human trafficking is thus deemed to be 'only' less lucrative than global drug trafficking (UNODC 2011b). Nonetheless, while so much money is involved, human trafficking is still usually not a priority for law enforcement, especially not with a focus on its monetary gain, and oftentimes not beyond the limited emphasis on sex trafficking.

Discussing the importance of the potential corrosive effect of a human trafficking case like the Epstein case also requires highlighting known trends about this crime. Focusing on its profitability, it must be stressed that there are at least four reasons why the afore-mentioned estimated $150 billion globally in 'human-trafficking profits' is most likely an underestimation.. First, forced commission of crimes such as couriering of drugs is progressively being recognized as a new form of human trafficking (UNODC 2020).[28] Hence, some of the money now ascribed to the estimated proceeds from drug trafficking, $300 billion worldwide, or to other crimes such as forced robberies or pickpocketing may therefore have to be added to human trafficking 'profits'.[29] Second and as a related issue, criminals tend to be opportunistic in that they go where 'business' opportunities drive them. As a cynical remark about the latter, commodities like drugs can be used once only, whereas persons can be exploited over and over (Coster van Voorhout 2009, 2020). In other words, persons are a continuous source of 'income' to which criminals will likely resort more in the future, especially as more persons become vulnerable to exploitation. As a third and connected issue, due to the many misconceptions about human trafficking and a failure to recognize that one crime (human trafficking) can be hidden in another (e.g., drug trafficking or where prostitution is criminalized), victims are sometimes mistaken for perpetrators.[30] Now labeled the blind spot in the context of crimes that undermine the liberal democracy of the Netherlands, there are not only indications of especially (vulnerable) youth being recruited into drug crime, but also as so-called 'hitters' or 'spotters' (CKM 2022). Hitters do the actual killing of individuals, usually for low payment. Spotters are on the lookout during such an assassination. Fourth and final, in the last five years, the number of estimated human trafficking victims has gone up with 25%, to 50 million people worldwide (cf. Shelley 2021; cf. ILO, IOM and Walk Free 2022). This means, first of all, that there are more trafficked persons today than ever enslaved before in history. Additionally, this denotes thatthis crime is most likely becoming even more lucrative than when fewer victims were exploited. While the number of victims are increasing globally, there is however a 45% drop in prosecutions of human trafficking since its first year of registration of 2015 (Ibid.; TIP report 2022, p. 62).[31] Also, there is a 24% decrease in the worldwide victim identification rate since the year of the highest mark, 2019 (TIP report 2022). Moreover, recently, due to COVID-19, the share of children among detected human trafficking victims around the globe has tripled (UNODC 2020, p. 3). Men, women and children are increasingly vulnerable to being trafficked within and across borders, moreover. In the future, more persons will thus become exposed to human trafficking, since globally the number of people forced to flee their homes is the highest since World War II (UNHCR 2022, p. 3). All expectations are that climate change, wars, conflicts including over resources like water and essential raw elements, and increased social and economic inequality will create further (internal) migration and thus vulnerability to exploitation.

These global data and trends demonstrate that the prosecution, protection of victims, and prevention of human trafficking requires the whole of global society working in concert. Liberal democracies, together with autocratic states with oligarchical rule –especially those whose economies rest heavily on human trafficking–, companies, NGOs, the general public, and others will thus have to collaborate if they are to address this increasing problem. Insofar as responsibility of companies like banks conducting corporate social responsibility to (in)directly counter human trafficking in supply chains is concerned, this crime is nowadays often known as modern slavery (Coster van Voorhout 2020). To sum up, this urgency about the increasing scourge of human trafficking itself also shows the significance of exploring its –potentially growing– corrosive effect on one or more liberal democracies. Against this backdrop based on global data and trends about human trafficking, the next sub-section examines the facts and procedure of the Epstein case, so as to study its potential corrosive effect on one or more liberal democracies.

- The Epstein case: Facts and procedure

Given that the aforementioned four US Senators deemed that the Epstein case had a corrosive effect on the US's liberal democracy, it is important to first explain how this criminal investigation started in 2005 with a report by a mother of a 14-year-old girl who alleged that the then 52-year-old Mr. Epstein had sexually abused her daughter.[32] That investigation by initially local police and thereafter the Federal Bureau of Investigations revealed that, between approximately 1999 and 2007, Mr. Epstein and multiple co-conspirators assembled a network of at least thirty-four underage girls whom he sexually abused at his mansion in Palm Beach, Florida.[33] Following the FBI's investigation, by May 2007, the prosecution office completed an 82-page prosecution memo and a 53-page draft indictment against Mr. Epstein, charging him with federal crimes related to the sex trafficking of minor victims.[34] The federal prosecutor, Mr. Acosta, set a tentative date of 15 May 2007 to indict Mr. Epstein. However, on 24 September 2007, Mr. Acosta concluded a non-prosecution agreement[35] with Mr. Epstein, without notifying the known thirty-four victims moreover.[36] In return for federal immunity, Mr. Epstein agreed to plead guilty to two low-level state solicitation of prostitution charges and serve eighteen months in the county jail.[37] Mr. Epstein received work release from that jail and spent considerable time not imprisoned but in his office. On 8 July 2019, at least fourteen years after the initial criminal investigation, the prosecutor in the Southern District of New York (SDNY), Mr. Berman, indicted Mr. Epstein for sex trafficking of minors.[38] In addition to Palm Beach, Florida, the sex trafficking operation was suspected of also having reached into at least New York, New Mexico, the US Virgin Island of St. James and London, the United Kingdom, while also being connected most likely to another co-conspirator in France.[39] Mr. Acosta, later on 12 July 2019, resigned from his position of Secretary of Labor in the Trump Administration. The US labor department oversees the large government agency dedicated to anti-human trafficking and anti-child labor efforts. On 10 August 2019, due to Mr. Epstein's suicide in pre-trial detention, the aforementioned SDNY prosecutor Berman discontinued the case.[40] Other cases including a defamation case against one of Mr. Epstein's lawyers, Mr. Alan Dershowitz, by one of Epstein's named victims, Ms. Virginia Giuffre[41], are still ongoing.[42]

- The Epstein case: Its corrosive effect

The US Senators were not the only ones who claim that the Epstein case undermined the US's liberal democracy. The investigative journalist Ms. Julie K. Brown, whose work prosecutor Mr. Berman credits as having been important for his office's prosecution of Mr. Epstein, calls it *perversion of justice* (Brown 2021).[43] Brown, who reported on the Epstein case relatively early on, in 2018, chose this title with its double meaning so as to emphasize the damage done to, not only victims and by extension their families and communities, but also the rule of law and potentially democracy in the US. She aptly summarized this corrosive effect as: 'Epstein got away with his crimes because nearly every element of society allowed him to get away with them. Professional, legal, and moral

ethics were set aside for a broken system of values that places corporate profits, personal wealth and political connections, and celebrity above some of the most sacred tenets of our faiths, our teachings, and our democracy' (Brown 2021, p. xiv).[44] One of the four US Senators who came to the aforementioned conclusion, Senator Ben Sasse, then-Chairman of the Senate Judiciary Oversight Subcommittee, also confirmed this understanding of this corrosive effect in his letter to the then-Attorney General William Barr, as follows: '(t)he idea that wealth and connections can buy injustice—the only plausible explanation for such pathetically soft terms for a serial child rapist at the heart of a massive international criminal enterprise—is wholly and completely inconsistent with the basic notions of fairness and equality that undergird the rule of law enshrined in our Constitution'.[45] As already follows from his reference to the international nature of this trafficking operation, Senator Sasse at least alludes to the fact that this corrosive effect was felt beyond America, despite the fact that most of the currently publicly known sex trafficking seems to have occurred on US territory. As Senator Sasse explains: '(m)oreover, the notion that one individual's plea could shield a whole class of potential co-conspirators of uncertain size and identity from legal liability would—if treated as enforceable—pioneer a new model for one fall guy to shield all members of a criminal enterprise from accountability to the law (Ibid.). Here, two examples will be given about two other liberal democracies that were impacted by the Epstein case. First, this corrosive effect was also felt in the United Kingdom, because after no apparent criminal investigation by the British, Mr. Epstein's known British co-conspirator, Ms. Maxwell, was convicted in the US for conspiracy to commit and commission of sexual exploitation including at her London residence.[46] As another example, Prince Andrew from Britain laid down all his functions upon a BBC interview that sought to address allegations by the aforementioned Ms. Virginia Giuffre at, inter alia, the aforementioned London residence of Ms. Maxwell. Second, in France, seemingly without a previous criminal investigation by the French, Mr. Jean-Luc Brunel, who reportedly founded his modeling agency with Mr. Epstein's money, was arrested for using this agency to recruit victims that also ended up in Epstein's scheme.[47] The French, as the Americans, have had to discontinue this criminal investigation because, like Mr. Epstein, Mr. Brunel was found to have committed suicide in a similar fashion as Mr. Epstein. Mr. Epstein was also traveling back from France to the United States before he got arrested on the SDNY's charges, which worried prosecutor Mr. Berman because of the precedent of Mr. Roman Polanski who had never been extradited from France to the US. In addition to thus affecting at least three established liberal democracies, the Epstein case also negatively impacted Mr. Epstein's bank. Deutsche Bank entered into a $150 million settlement because of its failure for years to properly monitor account activity conducted on behalf of the life-long registered sex offender Mr. Epstein.[48] The settlement record shows that Mr. Epstein made suspicious payments to his (alleged) co-conspirators, wired money to Russian models, and made a connected cash withdrawal of $100,000 for 'tips and household expenses'. Having seen the details of the Epstein case, which after many years did come to light (at least somewhat), it is fair to conclude that it serves as an illustration of how a complex, networked human trafficking case can advance the corrosion of multiple liberal democracies and compromise a global financial institution that ensures financial flows between multiple states around the world.

### 3.2. Four Factors That May Explain the Corrosive Effect of Human Trafficking Cases like the Epstein Case

While the four Senators, the investigative journalist and even crime writers[49] all call attention to this corrosive effect of the Epstein case, at least insofar as the US is concerned, strikingly, the extensive literature[50] and case law review of (connected[51]) criminal cases[52] conducted for this article found hardly any exploration thereof. Consequently, this sub-section cannot resort to explanations of this corrosive effect based on in-depth analyses from reviewed academic research or (criminal) cases[53], which also sets it apart from comparable damage done to liberal democracies by conspiracies about human trafficking (See Section 2).

Considering the importance of examining this corrosive effect, also against the background of the global data and trends about human trafficking more generally (See Section 3.1.), this article will therefore present its own analysis based on literature concerning democratic deconsolidation more broadly (Cf. Section 1, e.g., Diamond 2019; Mounk 2019; Levitsky and Ziblatt 2018; Guriev and Treisman 2022). In recognition of the fact that liberal democracies no longer get violently overthrown as often but rather 'erode slowly, in barely visible steps', this analysis shows, as much as possible, separate factors that explain the overall corrosive effect of the Epstein case (Levitsky and Ziblatt 2018, p. 3). While realizing that the Epstein trafficking operation is still being investigated so that much about this case can still surface[54], for now the following four corrosive factors are distinguished.[55]

First, the Epstein case appears to have compromised the integrity of the rule of law and impartial justice, as can best be detected by following the 'wealth and influence' which he used mostly stealthily[56] rather than openly[57] in order to 'from the beginning of the case marshal [ . . . ] the weaknesses of the criminal justice system to his benefit' (Brown 2021, p. 77). As far as the origin of Mr. Epstein's wealth and influence is concerned, it is unclear how much money he made through sex trafficking itself. However, it is likely that Mr. Epstein had set up surveillance of all his properties that may have been used to blackmail those who sexually exploited the underage victims trafficked by him and his co-conspirators.[58] What is known about Mr. Epstein's (possible other) source of wealth and influence is that he advised affluent others about tax evasion and avoided taxes himself.[59] While tax avoidance is usually lawful, it can still be 'awful' in the sense that criminologist Passas explains it ('lawful but awful'; Passas 2005). This latter aspect will be revisited below in Section 4 because extracting through tax avoidance money from the economy where the wealth was accumulated that could otherwise be spent publicly including on strengthening democracy and the rule of law, for example, can also have a corrosive effect on liberal democracy (See Section 4). But returning to Mr. Epstein for now, his career may not have consisted of only tax avoidance. A person connected to Mr. Epstein, Mr. Hoffenberg, who testified to the grand jury that Mr. Epstein was the mastermind behind their joint scheme, was held accountable for crimes of tax evasion, mail fraud, and obstruction of justice in 1995. Mr. Epstein was not convicted for such crimes. Whatever the (il)legal source of Mr. Epstein's wealth and influence, it allowed him to ensure the provision of, inter alia, legal assistance to most victims and his co-conspirators who predominantly were from poorer socioeconomic backgrounds.[60] The effect of the latter use of his wealth and influence can also be seen in a court ruling about how Mr. Epstein gained immunity for sex trafficking through the aforementioned non-prosecution agreement (NPA)[61]: 'Worse, it appears that prosecutors worked hand-in-hand with Epstein's lawyers—or at the very least acceded to their requests—to keep the NPA's existence and terms hidden from victims. And to be clear, the government's efforts appear to have graduated from passive nondisclosure to (or at least close to) active misrepresentation'.[62] The latter means that the thirty-four known victims remained in the dark about the immunity provided to Mr. Epstein in exchange for still unknown activities in support of law enforcement, if any.[63] Mr. Epstein has thus been able to use his wealth and influence to not be fully prosecuted in 2008[64], only to have reportedly continued and widened his sex trafficking operation for at least a decade, until being indicted on 8 July 2019.[65]

Second, in terms of potential damage done to the integrity of democracy, some of Mr. Epstein's wealth and influence appears to have contributed to ensuring his local, state and federal political connections[66] from both political parties in the United States[67] and other powerful contacts in science, the legal community and academia, for example.[68] It is difficult to assess the degree to which money spent on politics by Mr. Epstein including on the US mainland and the US Virgin island of St. James[69] has played a role in the Epstein case because of lacking transparency on the role of money in US politics more generally (e.g., Mayer 2017; Bernstein 2020; Reich 2021; Whitehouse 2019). Correspondingly, it is unknown to what extent in the Epstein case investigators and prosecutors were insulated

from political interference or not. Comparably, it is difficult to detect the influence of (previous) criminal justice representatives on politics or the impact of the Epstein case on the checks and balances between the legislative, executive and judicial branches of government more widely (Whitehouse 2022). However, this case does indicate how the dividing line between politics and the criminal justice system cannot always easily be drawn in more general terms. For example, as mentioned above, the former federal prosecutor, Mr. Acosta, later became Secretary of Labor in the Trump Administration and resigned when his role in yielding his prosecutorial discretion to not at least fully prosecute Epstein became national news.

Third, in terms of social and economic inequality, Mr. Epstein was able to at a minimum exploit the poorer socioeconomic conditions of most of the thirty-four female victims and his (un)named co-conspirators. So even very local social and economic inequality—here between Palm Beach County and Palm Beach 'Millionaires Row', Florida—can allow a trafficker to exploit victims who moreover did not obtain complete justice for the harm done to them. As has become visible in the Epstein case, sometimes a perpetrator does not even have to use the wealth or (corresponding) power to compromise public and private institutions and their representatives, but the mere threat of being able to use or withdraw such money can already deter criminal investigations or due diligence efforts.

Finally and as a related point, in terms of lacking international coordination between and among liberal democracies and with an international bank, as said above, there is, first, no evidence that Ms. Maxwell from the United Kingdom and most likely Mr. Brunel from France were investigated in those jurisdictions nor that these states cooperated with the US to investigate the Epstein case. Also, evidently the global financial institution Deutsche Bank did not do its due diligence under its anti-money laundering obligations for a high-risk client and did not support one or more state investigations either. To conclude, while some aspects of the Epstein case are still ongoing, this exploration of the case thus far demonstrates how human traffickers can indeed ignite a crisis of public trust in its rule-of-law institutions and representatives and exacerbate the erosion of trust that people have in their institutions of liberal democracy more broadly. Extrapolating from the Epstein case to a larger implication, many of those human traffickers will, unlike Mr. Epstein seems to have done, not have to earn their money through any other crime than human trafficking because it is, as said, an estimated $150 billion 'industry' worldwide (ILO 2014).

### 3.3. Conclusions

To conclude, there are at least four categories of reasons that explain the need for gaining a better understanding of how, just as conspiracies about human trafficking, the actual crime itself also already appears to contribute to the subversion of liberal democracies worldwide: (i) financially, because of its lucrativeness that has demonstrably had a corrosive effect on (the integrity of) public and private institutions and their representatives; (ii) developmentally, because of its ever-evolvement into newer (also lucrative) forms like forced commission of -drug- crimes, for example; (iii) structurally, because of its embedding in (il)licit state structures; and (iv) conceptually, because at least some relevant behavior (of –Western– enablers) is not criminalized whereas it does facilitate human trafficking and can result in corrosion of liberal democracies and global financial institutions. Summarily put, complex, networked human trafficking cases can advance the erosion of liberal democracies due to the wealth and power used to undermine state structures, the general loss of trust by the general population in the state's ability and willingness to respond, and the attacks on principles like accountability to and equality before the law. But before explaining further how to deal with the corrosive effect of actual human trafficking, it is important to explore whether there may at present be an even more complicated interplay between fictions and facts about trafficking in persons and between corrosion happening within and from without liberal democracies.

## 4. Problems 3 and 4: The Interplay between Fictions and Facts and Corrosion from within and without

Given that the conspiracies and the actual crime all concern the same topic of human trafficking, it is worth examining whether these issues of fiction and fact pose separate problems or rather constitute a challenging interplay nowadays. Moreover, seeing that autocracies whose economies are heavily based on human trafficking export conspiracies about this phenomenon to liberal democracies and possibly Russian victims were trafficked in Epstein's operation, it is worth examining whether internal and external corrosion happen in siloes or rather interact as well.

### 4.1. The Interplay between Conspiracies and Actual Human Trafficking

Conspiracies about human trafficking already have consequences for real-life trafficking in persons, as can be illustrated with the first example of a group of 20 US Republicans –many of whom are well known for spreading the Pizzagate and QAnon conspiracies– who recently voted against anti-human trafficking legislation.[70] The latter makes it more difficult to detect actual victims of human trafficking within the liberal democracy of the United States that is already facing the consequences of the real-world Epstein case. It remains to be seen whether criminals like human traffickers have used their wealth and influence to hamper the legislative, executive or judicial processes concerning this crime in this liberal democracy, but both the US and other such liberal democracies are moreover seeing no end in sight to the creation of new conspiracies about the deaths of Mr. Epstein and Mr. Brunel who are speculated to not have committed suicide but rather supposedly have been murdered.[71] Such conspiracies may fuel others, and may even distract from properly examining the corrosive effect of the actual crimes committed by Epstein and his co-conspirators by the justice systems of the US and France as well as possibly the UK and Deutsche bank or other financial institutions because fact and fiction can no longer be distinguished as easily.

These conspiracies about Mr. Epstein and Mr. Brunel and related conspiracies remain a continuing source for the new conspiracism that dispenses with the need of giving any explanation. All these (new) conspiracies about human trafficking, especially QAnon, moreover, now appear to be the animating 'philosophy' and glue that holds together the different factions of the Far Right in the United States that was responsible for January 6th and appears to plan more such attacks.[72] This may serve to recruit new members of such groups who may be forced to commit crimes, a possible new form of human trafficking, moreover.

As a fourth and final example of how fictions and facts about human trafficking interact nowadays, this aforementioned, anti-democratic and anti-rule-of-law movement seems to go beyond the bounds of the United States, as the examples of the recent elections in Sweden and Italy but also longer running examples like Hungary, Poland, the Philippines, India, Ethiopia, Brazil, Russia, Turkey and China demonstrate (Rachman 2022). This latter observation links to the importance of examining how both fictions and facts concerning human trafficking do not only erode a liberal democracy from within, but may also interact with erosion from without.

### 4.2. The Interplay between Corrosion from within and without

Internal and external corrosion no longer appear to be separate phenomena but rather constitute an even more complex interplay nowadays, as can be elucidated by 'following the money'. For this purpose, it is important to emphasize once more that much is unknown about the funding and planning of not only the spreading of conspiracies but also of the commission of the actual crime of human trafficking. Therefore, it is helpful to, first, emphasize another example than the Epstein case because Deutsche Bank is not the only bank that shows that the moral, societal and legal obligation to safeguard the integrity of the global financial system against—the corrosive effects of—human trafficking was not upheld. That is, the 2020 sex trafficking cases in Australia and the Philippines resulted

in Australia's second-largest bank Westpac having to conclude the highest transaction in Australian corporate history of $1.3 billion.[73] Westpac had to admit to 23 million breaches of anti-money laundering and terrorist financing laws relating to international transfers and transactions for five years. However, just as in the Epstein case, its corrosive effect was hardly studied, which moreover serves to explain that this same corrosive effect by other forms of human trafficking that come to light even less than sex trafficking has not been the subject of such research either. Those other forms include labor trafficking, organ removal, forced begging or the forced commission of crimes. While this lack of study of corrosion from within a liberal democracy is already problematic, victims of all these forms of human trafficking in a country like Russia no longer get legal recourse at their (international) courts for domestic or cross-border human trafficking.[74] Not only does this hinder victim detection and protection in Russia, but also liberal democracies which are transit or destination countries in cases involving Russia will thus no longer detect this crime through victim reports. While victim detection rates have already gone down globally, as mentioned above on the basis of global data and trends about human trafficking, it cannot stand that the corrosive effect of this crime remains under-studied, knowing that a much more proactive approach will have to be taken if we are to uncover the crime in the first place. This prompts the question why there is a lack of in-depth analyses of the corrosive effect of human trafficking, both in absolute terms and relatively speaking as compared to conspiracies like Pizzagate and QAnon. The best answer that can be provided is that the actual crime of human trafficking is *doubly hidden*.[75] Whereas conspiracies inevitably become public, at least after some time, its funding and planning oftentimes remains just as secret as for the actual crime of human trafficking. However, with real-world human trafficking, *both* the crime itself *and* the money earned and spent usually remain secret. Following this reasoning, it thus also makes sense to focus on the corrosive effect for both conspiracies about and the actual crime of human trafficking, by 'following the money'. This approach that focuses on financial flows is even more important for actual human trafficking cases than conspiracies about the phenomenon, given that, as demonstrated in the previous sections, much more is currently known about those fictions than the facts.

For corrosion from within, the understanding that money earned through criminal offenses such as human trafficking, 'dirty money' or, where (il)legal money is spent anonymously, 'dark money', can corrode a liberal democracy from within, has already in 2013 been condemned by US Supreme Court Justice Breyer as: 'Where enough money calls the tune, the general public will not be heard'.[76] Compared to the estimated amount of dark money spent on political campaign financing in the United States since the Supreme Court case of *Citizens United v. Federal Election Commission*, $1 billion, human traffickers 'make' at least 150 times more money globally (Whitehouse 2019, p. xiii). So, it is, in such economic terms, a reasonable topic for further examination (Bun et al. 2019).

For corrosion from without[77], it is known that oligarchs of countries whose economies rest heavily on human trafficking like Russia store their dirty and dark money in liberal democracies, exactly because these states offer the rule-of-law protections needed to maintain access to that wealth (Snyder 2018; Vogl 2022; Guriev and Treisman 2022).[78] As Snyder explains: 'Tyrants first hide and launder their money, then use it to enforce authoritarianism at home—or export it abroad. Money gravitates to where it cannot be seen, which in the 2010s was in various offshore tax havens. This was a global problem: estimates of just how much money was parked offshore, beyond the reach of national tax authorities, ranged from $7 trillion to $21 trillion. The United States was an especially permissive environment for Russians who wanted to steal and then launder money. Much of the Russian national wealth that was supposed to be building the Russian state in the 2000s and 2010s found its way to shell corporations in offshore havens. Many of these were in America' (Snyder 2018, p. 261). Like the US, other liberal democracies were and are comparably susceptible to laundering of money, such as in the City of London in the United Kingdom, real

estate in France, and intellectual property-related aspects in the Netherlands (cf. Bullough 2018; Burgis 2020; Reich 2021; Tjeenk Willink 2021).

Consequently, it is not surprising that there is an interaction between internal and external corrosion, because a general distrust of the public in liberal democracy helps to promote autocracy at home by claiming, in short, that liberal democracy is 'a sham' (Snyder 2018, p. 262).[79] Worse yet, it is no longer as easy to distinguish between external and internal corrosion of liberal democracies because of the involvement of Western enablers (e.g., Guriev and Treisman 2022; Vogl 2022). While Western tax advisors, lawyers and consultants may not even always commit crimes when they ensure the existence of more networked, complex structures of human trafficking, the spreading about conspiracies about this phenomenon and other such crime, the effect may be corrosive of liberal democracy nonetheless. This thus leaves us with many questions about how best to detect the interaction between internal and external erosion.

Seeking to find answers to such questions by focusing on the monetary aspects of both forms of internal and external erosion, it helps to use the following simplification. Simply put, whereas crimes like tax evasion take money out of a country, money laundering brings money in. If one does not (pro)actively examine the origins of laundered money, the inflows of dirty or dark money from domestic or foreign human traffickers or spreaders of conspiracies about this phenomenon can appear just another source of investment in liberal democracies. Consequently, it is insufficient to examine money laundering 'alone' without examining how that laundered money was 'earned'. Money laundering is a derivative offense, meaning there must be a predicate offense such as the lucrative crime of human trafficking. When potentially laundered money comes from abroad, whether from another liberal democracy or an autocracy whose economy rests heavily on human trafficking[80], such money laundering schemes will oftentimes require the involvement of enablers in liberal democracies like tax advisors, lawyers and consultants who know the domestic law and practice (e.g., Guriev and Treisman 2022; Vogl 2022). Without such enablers, generally speaking, the more networked, complex structures of human trafficking and related money laundering and corruption would hardly exist (Cf., on corruption, Slingerland 2018). Hence, this also means that 'following the money' will have to focus on detecting potential victims and perpetrators as well as enablers who facilitate the crime, so that, to focus on one example of Western enablers, financial institutions can comply with their moral, societal and legal obligation to safeguard the integrity of the global financial system against —the corrosive effects of— human trafficking.

*4.3. Concluding Remarks*

To provide concluding remarks, negatively put, if human traffickers like Mr. Epstein, oligarchs from autocratic states whose economies rest significantly on human trafficking, and their (Western) enablers can commit their crime with near impunity and are not hurt in their wallets so that they maintain their (corresponding) influence, the risk is that they grow more wealthy and powerful and are consequently able to erode democratic and rule-of-law institutions and their representatives further. This downwards spiral is disturbing, because the above-provided examples demonstrate how human trafficking operations as seen in the Epstein case already resulted in a decrease of public trust in democratic and rule-of-law institutions, thus damaging the social contract with the general population that is essential for a liberal democracy to exist. This raises many questions as to whether or not those living in liberal democracies want to ensure they contribute to democracy and the rule of law or rather to autocratic and oligarchical rule.[81] So, how should those in a liberal democracy respond, both in law and through research in academia?

## 5. Carefully Finding Solutions for Countering Organized Subversive Crime

Having explained the four interlinked and multi-faceted problems in the previous three sections, the remainder of this article tries to cautiously find solutions. One of the most important solutions will have to involve the countering of the –most damaging aspects of

the– corrosive effect of human trafficking and other such crime. Therefore, this section tries to provide answers to the conceptual, legal and research questions raised in the previous three sections (See Sections 2–4).

*5.1. The New Dutch Concept of Organized Subversive Crime: An Explanation*

In the Netherlands, Dutch scholars and policy-makers have started to call for specifically addressing some crimes that have a corrosive effect on its established liberal democracy as a unique category or aggravating circumstance. The most recent demand for this was made in relation to the July 2022 trials concerning the openly violent murder on 15 July 2021 of a journalist who investigated organized crime, Mr. Peter R. de Vries (Meeus 2019; Laumans and Schrijver 2021). Before his murder in broad daylight on the streets of Amsterdam, the nation's capital, two other such assassinations took place: of defense counsel Mr. Derk Wiersum and, before that, of the brother of an insider witness in a large-scale and international drug-trafficking case.[82] While these openly violent crimes happened 'out in the open', Dutch scholars and policy-makers assume that, because of the aforementioned *dark figure*, most –connected– organized subversive crime and its enabling remains off-the-radar and insidiously 'eats away' at state structures and public trust (Tjeenk Willink 2021; Boutellier et al. 2019; Korf et al. 2018; Meeus 2019; Laumans and Schrijver 2021).

Against this backdrop, Dutch scholars are still developing this new concept so as to fundamentally understand a potentially novel phenomenon of organized crime that oftentimes includes but is not limited to human trafficking (Huisman and Kleemans 2017; Tops and Tromp 2017; Eski et al. 2021). For this, they have converted the verb 'to undermine' into a noun: 'crime that undermines liberal democracies'.[83] They deem such crime deserving of special attention, because it not only harms victims and, by extension, their families and communities, but, in addition, subverts liberal democracies or at least has the potential to do so. The assumption is that some crime is thus not limited to 'only' exploiting state structures for financial or other gain, but rather erodes the integrity of democratic and rule-of-law institutions and their representatives. Alternatively described as 'the interwovenness of upper- and underworld' or 'organized subversive crime', such criminality contributes to, or even underlies, an overall sense of injustice and loss of trust from citizens in the state (Tjeenk Willink 2021). Some even deem this potentially newer type of criminality a *crisis*, arguably on a par with 'climate change, migration questions, and increased societal divisions' (Ibid.; cf. about the US Reich 2021; cf. on US oligarchy[84] Bernstein 2020; Giridharadas 2019; Mayer 2017; Snyder 2018). This crisis stems, according to them, from the resulting anger, frustration, and cynicism of the general public about democracy and the rule of law, which has the potential effect of corroding the moral and social foundation of liberal democracies (Ibid.) This can even result in a negative spiral: the sense that a liberal democracy is out of the public's control creates frustration, spurs anger and resentment, and drives polarization in the citizenry which can, in turn, result in even less trust in the state. Those effects hardly ever stay within the realm of one country in our interconnected world with a near global financial sector and technology realm (Cf. Guriev and Treisman 2022, p. 7).

Before exploring further the contours of this new concept and its potential relevance beyond the Netherlands, it is important to define one of its essential elements: liberal democracy. Here, a liberal democracy is defined as a state that combines free and fair elections with the rule of law, fundamental rights, and institutional checks and balances. Democracy and the rule of law are thus deemed to be of equal significance (the German concept of '*demokratischer Rechtsstaat*'). Both democratic processes and the rule of law, rather than rule by men, are necessary but insufficient conditions for a liberal democracy. This understanding is in keeping with the definition used by the United Nations:

> 'the rule of law refers to a principle of governance in which all persons, institutions and entities, public and private, including the State itself, are accountable to laws that are publicly promulgated, equally enforced and independently adjudicated, and which are consistent with international human rights norms and

standards. It requires, as well, measures to ensure adherence to the principles of supremacy of law, equality before the law, accountability to the law, fairness in the application of the law, separation of powers, participation in decision-making, legal certainty, avoidance of arbitrariness and procedural and legal transparency'.

(UN SG report 2004).

Seeing that democracy and the rule of law are intricately linked and mutually reinforcing in a liberal democracy[85], it is important to stress that the above-cited political scientific, historical and social scientific literature predominantly, oftentimes as a short-hand, refers to democracy 'only' (e.g., Levitsky and Ziblatt 2018; Stanley 2018; Snyder 2018; Miller-Idriss 2022). Damage to the rule of law is usually implied. However, given that the remainder of this article examines solutions from a criminal law perspective, here, the impact on both democracy and the rule of law as mutually reinforcing governance principles is made explicit. As a related point, in more democracy-focused scholarly work, the democratic form of governance is often juxtaposed with authoritarianism in a manner that is oversimplified, as scholars like Foa and Mounk have criticized (Foa and Mounk 2021). Consequently, this article seeks to prevent as much as possible taking a binary democracy versus authoritarian lens. Such care is important because it is also recognized here, as mentioned above, that liberal democracies 'erode slowly, in barely visible steps' (Levitsky and Ziblatt 2018, p. 3). Therefore, if we seek solutions to better study organized subversive crime in law and research, the effects on the interplay between the rule of law and democracy in all its complexity must be highlighted. Consequently, emphasis is put on the importance of human rights for a liberal democracy because, as the philosopher J.S. Mill once feared, '(w)ithout human rights, democracy runs the risk of becoming a 'tyranny of the masses' (Cf. Alegre 2022, p. 6). Human rights laws place the rule of law at the heart of government[86], ensuring that 'any limitations on freedom must be set down in law and could only be justified to protect the good of others or to prohibit actions that were harmful to society'. (Ibid., p. 11). This also has a historical connotation because, '(t)hroughout much of history, large swathes of humanity had no rights at all in law. Slavery is so appalling because it is the antithesis of human freedom' and dignity, both of which are the essential feature of human rights. (Ibid., p. 14) The continued existence of human trafficking, also known as modern-day slavery, proves that 'making laws is not enough to protect our rights; those laws need to be respected and enforced and we can never afford to be complacent about the rights we enjoy'. (Cf. Ibid., p. 15).[87] As a final point on the issue of a liberal democracy in the world, as mentioned above, this article considers countries around the globe as mostly interconnected, especially insofar as the global economy and financial flows are concerned. Because of globalization and digitization, most countries have become part of an interwoven, networked web. This also means that negative impact in one state is bound to have an effect on other countries, and that, for example, the transfer of criminally obtained wealth from one nation is bound to influence others.

Revisiting the contours of the novel concept of 'crime that undermines liberal democracy', it is important to emphasize that it is not fully clear yet what it does and does not entail. This conceptual flexibility allows for social, technological and other changes to be captured in our understanding of the phenomenon as well as for specific local variations within a country or between countries. The same holds true for organized crime. In other words, there is no definitive list of offenses that constitute the concept of organized crime, either, for this same reason of conceptual flexibility. Rather, internationally legal definitions 'merely' comprise notions like an organized criminal group, serious crime, structured group, and criminal proceeds (Article 2 of the United Nations Convention against Transnational Organized Crime (UNTOC)).[88] Nonetheless, the UNTOC does include crimes like money laundering and corruption, while human trafficking is included in one of its Protocols. So, just as organized crime as a research theme that has existed for more than a century now, the currently proposed Dutch concept of 'crime that undermines liberal democracy' still has to be delineated, especially if it is to have meaning beyond the bounds of the Dutch state.

The most important debates about this concept go beyond generally agreed-upon tenets of organized subversive crime that, at worst, such criminality erodes public and private institutions and their representatives' integrity, sometimes even irreparably so (Huisman and Kleemans 2017; Tops and Tromp 2017; Boutellier et al. 2019; Spapens 2019; Kruisbergen et al. 2021; Tjeenk Willink 2021). For example, some scholars argue that the criminal *intent* is determining, whereas others rather focus on the crime's corrosive *effects* (Boutellier et al. 2019; Eski et al. 2021). As a downside, this lacking clarity arguably already results in both its over- and under-use (Ibid). For example, critics put forward that this undefined concept is oftentimes abused to disguise the oftentimes failed 'war on drugs' (e.g., Bruijn 2020). This dearth of conceptual clarity requires critical reflection on what does and does not constitute organized subversive crime and how to use research findings to determine priorities more generally. However, this larger ambition goes beyond the scope of this article; here, the examination is limited to the corrosive effect of human trafficking. As for the latter, the previous sections have established how some complex, networked forms of human trafficking and conspiracies about this phenomenon that result in the incitement of (mob) violence already seem to have this corrosive effect.[89] Crime that undermines democracy and the rule of law thus does not appear to be just a problem in the Netherlands, as more liberal democracies around the world seem to experience comparable detected crime as 'the tip of the iceberg' while assuming that there must be much more off-the-radar criminality 'whose bulk lies under water'. Not only the aforementioned recent storming of the US Capitol by criminal groups like the Oath Keepers and Proud Boys, but also Russia's interference in the US election, the mafia's influence over Italy, and the current political situations in Hungary and Poland indicate the range of recent examples of attacks on democracy and the rule of law around the world (Levitsky and Ziblatt 2018). All in all, it is thus fair to conclude that several liberal democracies are struggling to deal with these new challenges, while leaders of backsliding democracies like President Orbán of Hungary propose the alternative of an 'illiberal democracy' for its own country and as an improved model that should be exported to states like the US and the Netherlands (Rachman 2022, pp. 89–101).

*5.2. The Usefulness of the New Dutch Concept of Organized Subversive Crime for Criminal Law*

This article, as explained in the introduction, emphasizes a criminal law perspective. Criminal law should, because of its usually most significant impact on human rights norms and standards, be used as a last resort (*ultimum remedium*). Nonetheless, the preceding sections gave an initial indication as to how important it can be to specifically charge crimes that have the intent to, or effect of, subverting liberal democracies in fiction and fact, from within and without.[90]

The examples mentioned above demonstrate how some forms of complex, networked human trafficking can go beyond 'only' exploiting state structures for financial or other material gain but rather undermine them. For example, some organized subversive crime results in, damage, in addition to victims, and by extension their families and communities, to democracy and the rule of law as well. Revisiting the recent most outright attack on democracy and the rule of law, there have been calls in the US to respond to January 6th with a to-be-introduced domestic terrorism charge. This new charge will still have to be developed, because the US—unlike the Netherlands—has not ensured enforcement of domestic terrorism yet. However, it does not seem that all those who would need to be held accountable sought to terrorize an entire population or impose their will on the US government to specifically forcibly obstruct the execution of the Electoral Count Act and the Twelfth Amendment of the US Constitution, which address the counting of electoral votes.[91] Instead, at least some of the most responsible actors did more subtle and insidious damage to democracy and the rule of law in the background 'in barely, visible steps' for many years before January 6th, thereby readying and ultimately inciting the 'foot soldiers' to commit the (mob) violence (Levitsky and Ziblatt 2018, p. 3; cf. Snyder 2018). These 'masterminds' are most responsible, but also furthest removed from the violence on January 6th itself.

Therefore, it is important that, first, in criminal law, a further distinction with terrorism—whether international or domestic terrorism—is made.[92] To explain this position on terrorism further, it is important to note that there is no universally agreed upon definition of this crime. However, the only treaty that does define terrorism helps to further explain both the acts and intent required: an unlawful act 'intended to cause death or serious bodily injury to a civilian, or to any other person not taking an active part in the hostilities in a situation of armed conflict, when the purpose of such act, by its nature or context, is to intimidate a population, or to compel a government or an international organization to do or to abstain from doing any act'.[93] This definition fits poorly with those who leading up to January 6th did not seek to intimidate the general population in order to coerce the government into an act or omission through outright violence but rather seem to gradually erode trust in public and private institutions and their representatives. They seem to stealthily, through wealth and power, as has been explained for the Epstein case, motivate others to conduct the violence to achieve that aim. Whereas some may thus have attacked the US Capitol for the purpose of halting the certification of the election while publicly intimidating the general population moreover, others in the background rather seem to have created the overall climate through funding, planning and incitement so that the 'foot soldiers' would do the outright harm needed to corrode public trust in democracy and the rule of law further.[94]

So, it seems that it would be useful to investigate criminals for the specific conduct of intending to, or acting in a way that results in, erosion of democracy and the rule of law. Especially for criminals who commit crimes that are also fundamental rights violations, like human trafficking or other offenses like crimes against life (murder and manslaughter) or against physical integrity (torture and inhumane and degrading treatment), it may thus be helpful to introduce a new criminal charge or aggravated circumstance: organized subversive crime.[95]

As a first example, a criminal who ordered or executed the murder of a well-known member of the free press who investigates organized crime like the aforementioned Mr. Peter R. de Vries in the Netherlands causes the death of someone in this profession that is essential for a democratic society. This can have a chilling effect on others in this same profession and thus erode democracy and the rule of law. The same holds true for the above-mentioned assassinations of defense lawyer Mr. Derk Wiersum and potentially the brother of the insider witness if committed out of retaliation for his brother testifying in an international drug trial. This may have a chilling effect on lawyers who are essential for a fair trial and who will thus no longer be willing to render their services or for other witnesses coming forward to give testimony in order to ensure that justice can be done. Equally, it may be helpful to specifically label trafficked 'hitters' or 'spotters' on these murders as engaged in organized subversive crime, while recognizing the complexity that one crime may be hidden in another and thus also possible victim-perpetrator duality. Of course, it can be argued, since these crimes happened through outright violence in the open, that a domestic terrorism charge is (also) relevant in these high-impact murders. For those who intended to commit these crimes to, in short, terrorize the general population so as to weaken the state, that charge is reasonable. However, their activities do not all seem to have been based on this more public-facing terrorization, at least not in the years leading up to the openly violent murders. Some of the higher level actors, who appear to have funded, incited or perhaps even ordered the murderers so that 'foot soldiers' would commit them, already more insidiously in the background seem to have weakened democracy and the rule of law for a long time. Comparably, complex, networked forms of human trafficking including in sex and labor sectors or for the purpose of organ removal[96] can constitute organized subversive crime as well, as we have seen thanks to the few examples that did come to light such as the Epstein case.[97] Similarly to those who commit these actual crimes, those who –ensure the funding and infrastructure to— spread fictions in the form of conspiracies about human trafficking who thereby incite (mob) violence with the intent or effect of damaging democracy and rule of law, could also fall in this category of organized

subversive crime.[98] To clarify this point, in liberal democracies, some degree of spreading of conspiracies will have to be permitted out of respect for free speech. However, where such conspiracies spill over in criminal conduct in the form of inciting (mob) violence that line between legal and illegal conduct can be drawn.[99] Differently put, on a case-by-case basis, it will have to be determined whether the line is not crossed from criminalizing conduct or words used, on the one hand, into criminalizing thoughts, on the other.[100] The latter is more a cautionary tale about the limits of the prevention of crimes like terrorism than an inability to charge funding of and incitement to a crime involving violence.

As a final example in this non-exhaustive list, those involved in the –(inter)national funding, planning or incitement of the– storming of the US Capitol could perhaps better be charged with a to-be-introduced organized subversive crime of conspiring[101] to damage democracy and the rule of law than a new charge of domestic terrorism. This seems to better fit some of the crimes leading up to or committed on January 6th that do not constitute the violent overthrow but rather the 'ero[sion] slowly, in barely visible steps' of the liberal democracy of the US (Levitsky and Ziblatt 2018, p. 3).[102] This is especially important since reportedly of the more than 38,000 Oath Keepers members in the United States, more than 470 of them work in law enforcement or are members of the military.[103] Their involvement may already indicate a degree of erosion of democracy and the rule of law, through capture of public institutions (Cf. Whitehouse 2019, 2022). A major question in this regard remains who funded, planned and incited the storming of the Capitol and how one can charge the presumably larger network behind the 'foot soldiers' in the yet indicted seditious conspiracy: the funders, profiteers, and planners, for example. Finding evidence for the involvement of the latter requires an in-depth analysis of money transfers and coordination regarding communication and planning within the US or potentially even internationally, as at least the apparently global spreading of its motivating QAnon 'philosophy' warrants (Cf. Snyder 2018; Riggleman and Walker 2022; Rachman 2022).

As will have been noted, this new concept of organized subversive crime requires that crimes are committed with the intent to, or effect of, damage to democracy and the rule of law. On a case-by-case basis, both the subjective and objective elements of this crime as well as the threshold of what constitutes damage that erodes democratic and rule-of-law institutions or their representatives' integrity will have to be determined. Nonetheless, in this manner, criminalizing organized subversive crime does help drawing the historical record on all also higher-level actors who sought to subvert democracy and the rule of law; centralize truth finding which has become increasingly relevant in our day and age with all these attacks on the truth; and form the accurate basis for restoration including redress for victims among whom their next of kin and society at large.

Of course, a charge of organized subversive crime could be brought alongside or in addition to terrorism crimes or crimes committed with terrorist intent. But the latter two crimes are specific in their aim at wanting to terrorize the general population for a political motive or to compel a government or an international organization to do or to abstain from doing any act, on the one hand, or seeking the much more subtle and hidden creeping poison of organized subversive crime, on the other. Differently put, in this complicated interplay between fiction and fact as well as corrosion from within and without, it gets increasingly more difficult to draw the dividing line between organized crime for profit and political crime for purposes of undermining democracy and the rule of law, especially in relation to lucrative crimes like human trafficking that 'earn' the funding needed for the erosion of liberal democracies.

### 5.3. The Usefulness of the New Dutch Concept of Organized Subversive Crime for Research

Even if legislators do not (yet) introduce this new crime or aggravating circumstance, it remains relevant to conduct research into crimes that undermine democratic societies governed by the rule of law.[104] One of the most effective ways to conduct research into human trafficking and other organized subversive crime is by identifying especially the larger, more hidden criminal networks and take the profit out of this criminality.[105] In other

words, it helps to follow the financial trails criminals leave behind (Coster van Voorhout 2020; TIP report 2022). By following the money, it becomes possible to examine erosion of democracy and the rule of law.[106] Focusing on erosion helps, because, as Snyder mentioned, '(e)rosion reveals what resists, what can be reinforced, what can be reconstructed, and what must be reconceived'. (Snyder 2018, p. 13). This also means introducing greater transparency so that the exploiters of banking secrecy, tax havens, intelligence networks and organized crime in liberal democracies can be detected (Cf. Guriev and Treisman 2022 and Vogl 2022). This may even disclose the enablers such as banks, law firms, accountancy organizations and tax advisories whose acts are generally not criminalized but without whose activities the aforementioned complex, networked forms of organized subversive crime could hardly be committed. Such research into what public[107] and private[108] institutions get eroded through organized subversive crime should be done with due regard for history.[109] Historically, human trafficking is relevant in a twofold manner. As long as education does not correctly address the history of slavery and indeed the imperial histories or slave-based histories of most Western states, it allows for the twofold problems of racism and human trafficking. As Mr. Eric Williams explains: '[slavery was] basically an economic phenomenon. Slavery was not born of racism; rather, racism was the consequence of slavery'. (Williams [1944] 2000, pp. xi–xii).[110] For example, we still notice the morphing of justifications of slavery into language that hides that connotation such as, in the context of the United States, the reference to state rights as well as the perpetuation of a nondemocratic system with an electoral college and with giving two senators to each state (Lepore 2018). In Europe, although states differ, Germany, France, Britain, Italy, the Netherlands, Spain, and Portugal have all had significant slave trade and slavery pasts before integrating their empires into the European Union, and in all these countries populist antimigration rhetoric appears on the rise. Plus, this lack of a real historical understanding is the reason why many argue that it is better to refer not to human trafficking but rather to modern slavery. It is a reasonable question whether slavery and the slave trade have morphed into a new form[111] given the increasing world population and social and economic inequality which makes more people vulnerable to being trafficked or, in the words of Kevin Bales, 'disposable' (Bales 2012).[112] Considering all these heightened complexities, it is only fair to conclude that a good response to human trafficking in all its facets will require the whole-of-society by many liberal democracies acting in concert.[113] Revisiting the previous example of Russia, seeing the significant benefits to the Russian labor economy that human trafficking provides, President Putin is unlikely to address the problem in the near future. So this also means that human trafficking is another fault line for liberal democracies, on the one hand, and autocracies with a policy or pattern of human trafficking, on the other. This leads to the following conclusion that research into crimes that undermine democratic societies governed by the rule of law is indeed urgently required, as follows.

## 6. Conclusions and Recommendations

This article sought to answer the central research question: Should some forms of human trafficking be dealt with as a new and unique category of crime or accompanied by an aggravating circumstance that emphasizes its corrosive effect on liberal democracy and, if so, under what conditions would that be useful in law and in research? To formulate an answer to this question, the preceding sections have demonstrated that, much like conspiracies about this crime that incite (mob) violence, some specific forms of the actual offense of human trafficking can qualify as organized subversive crime as well. Also, this article has shown that normative and, in future, empirical research, including examinations that improve our historical understanding of disintegration of liberal democracies, can serve as a guide to repair and, hopefully, more proactively as an agenda for its prevention. Therefore, it indeed seems helpful that the Dutch have pioneered the concept of organized subversive crime so as to examine how an offense like human trafficking advances the erosion of liberal democracy or, in this interconnected world, several liberal democracies. For that, human trafficking first has to be detected, and fact has to be distinguished from

fiction. After detection, we can build resilience and ensure prevention by especially focusing on how to make liberal democracies more democratic and strengthen the rule of law. For the rule of law, this means the interplay of civil, administrative and criminal law with due consideration to international human rights norms and standards. In this dynamic, criminal law has to particularly focus on holding accountable those who subvert liberal democracies, in fiction and fact, and from within and without (or a combination of all such factors).

As a first recommendation for further research, it is helpful to examine the systemic issues of organized subversive crime with the use of the following metaphor. Just as regular business, organized subversive crime requires people, money and infrastructure. In criminal law terms, this means the crimes of human trafficking (people), money laundering (money) and corruption (infrastructure ). Using this metaphor helps focusing research on the systemic issues of organized subversive crime. If research can find the people, money and infrastructure of a crime like drug or weapon trafficking, for example in a source of data that thus far is not proactively being queried such as in banking records, it helps our fundamental understanding of witting and unwitting perpetrators, enablers and victims. This shows the points of weakness in this complex, oftentimes networked interplay between criminals and public and private institutions as well as their representatives which, in turn, can show where to build resilience and take preventative action.

As a further, related recommendation, it helps to conduct this type of research in a public-private partnership, and to pay particular attention to detecting the otherwise often undetected victims of human trafficking in data sources that thus far have been used less regularly. For example, in a consortium involving banks, ministries, non-governmental organizations, and academics, it is possible to find otherwise undetected victims, follow the financial flows of human trafficking, and thereupon also trace where the crime intersects with the 'upperworld' (Coster van Voorhout 2020; TIP report 2022). All of this helps us gain a better understanding of whether or not and, if so to what extent, organized subversive crime indeed contributes to the erosion of liberal democracy locally or, in our interconnected world, globally, in both an empirical and normative manner.

On the one hand, we live in challenging times of rising social and economic inequality in and between countries, war, and climate change which will encourage more migration. On the other hand, we can now conduct data-driven research; explore downsides of modernization and globalization; create new options for corporate social responsibility; and learn from history how slavery and slave trade has contributed to both racism and discrimination, on the one hand, and human trafficking, on the other. Most of all, we can really develop our responsibility as academia to respond to the subversion of liberal democracies, by seeking to build resilience and prevent organized subversive crime in and between states through a combination of empirical and normative research.

**Funding:** This research received no external funding.

**Institutional Review Board Statement:** Not applicable.

**Informed Consent Statement:** Not applicable.

**Conflicts of Interest:** The author declares no conflict of interest.

## Notes

[1]　Lincoln's Key Letter Foreshadows Second Inaugural 4 April 1864.

[2]　For an overview of data used to undergird this thesis and the conclusion itself, see the report by the Economist Intelligence Unit from 2021, available at: https://www.economist.com/graphic-detail/2021/02/02/global-democracy-has-a-very-bad-year (accessed on 22 November 2022). Cf. under the term 'democratic deconsolidation', e.g. Yascha Mounk and Larry Diamond (Mounk 2019; Diamond 2019). Some argue that these concerns about the decline of democracies worldwide are somewhat exaggerated but even they express concern about the degree to which what they call 'spin dictators' damage liberal democracies, oftentimes through erosion from without (See Guriev and Treisman 2022). Similarly, on the trend of the decline of liberal democracies itself, see for example Levitsky and Ziblatt (2018). Rather than decline through an outright coup, they argue that currently democracies rather 'erode slowly, in barely visible steps' (Levitsky and Ziblatt 2018, p. 3). More positive is a scholar

like Steven Pinker (See Pinker 2018). However, the author did not find any writings on this topic from Pinker after 2018, who may also have changed his opinion at least somewhat after 6 January 2021. Pinker's language expertise was used by the Epstein defense (Brown 2021, p. 156).

3    *Supra* note 2.

4    At the time of writing, the Dutch public prosecutor has charged the Mr. De Vries murder—and may in the future do the same for the person or persons who ensured the funding and/or gave the order—as a crime committed with 'terrorist intent'. The latter is, unlike in the United States, in the Netherlands not reserved for foreign, international terrorism 'only'. The Dutch courts have not decided yet. In the US, prosecutors have opted for charging the head of the Oath Keepers, Mr. (Elmer) Stewart Rhodes, and four of its members who stormed the Capitol (the four members) or planned it (Rhodes) with 'seditious conspiracy' ex 18 U.S. Code Chapter 115-Treason, Sedition, and Subversive Activities, more specifically 18 U.S. Code § 2384-Seditious conspiracy. This charge requires the prosecution to prove that two or more people conspired to 'overthrow, put down or to destroy by force' the US government or bring war against it, or that they plotted to use force to oppose the authority of the government or to block the execution of a law. This Civil War-era charge has been explained to mean that these Oath Keepers conspired to forcibly oppose the authority of the federal government and forcibly block the execution of laws governing the transfer of presidential power. Specifically, they are accused of conspiring to forcibly obstruct the execution of the Electoral Count Act and the Twelfth Amendment of the Constitution, which address the counting of electoral votes. It remains to be seen whether the (criminal) conduct of funders, planners and (other) inciters can also be indicted under this relatively specific seditious conspiracy charge. While it is understood that conspirators can perform a range of acts or omissions that are further removed from these specifically violent election-undermining acts and still be held accountable for their role in the conspiracy, it may prove difficult to charge the (inter)national funders, planners, and profiteers of all conduct leading up to January 6th and following this storming of the US Capitol under this specific charge, just as the Dutch are struggling to have the domestic terrorism charge fit the murder offense committed.

5    There is an important debate to be had about the degree to which the United States should fulfil this role of 'policeman of the world' regarding human trafficking by drafting these Trafficking in Persons (TIP) reports, but this discussion falls outside the scope of this article. Here, the TIP report will only be used where findings have also been confirmed by other sources such as the ILO or UNODC.

6    This means that this article cannot delve into the entire range of democracies including its most populated, India, and the many others in North America (Canada), western Europe, Central and Eastern Europe, Latin America and the Caribbean, Asia and Australasia, the Middle East and North Africa, and Sub-Saharan Africa. See Section 6.

7    As a consequence, this may also result in polarization which according to Levitsky and Ziblatt is a sure way to end democracy.

8    Cf, though in relation to education rather than reporting research findings and conceptualizing the intent or effect of subverting liberal democracies (Benton and Peterka-Benton 2021; Cook 2020).

9    United States District Court, Western District of Virginia, Charlottesville Division, Sines v. Kessler, No. 17-cv-00072.

10   Dutch news service NOS, American conspiracy QAnon also on the rise in the Netherlands, 2020. Available at https://nos.nl/nieuwsuur/artikel/2349814-amerikaanse-complottheorie-qanon-ook-in-nederland-in-opkomst (accessed on 22 November 2022).

11   Hitler did the same in *Mein Kampf*: 'The Jews were largely responsible, he [Hitler] says he found, for prostitution and the white-slavery traffic' (Shirer [1956] 1990, p. 26).

12   'The operative concept in the Russian language today is *bespredel*, boundary-less-ness, the absence of limits, the ability of a leader to do anything. The word itself arose from criminal jargon' (Snyder 2018, p. 80).

13   The 2021 Trafficking in Persons Report includes Russia as one of 11 governments with a documented "policy or pattern" of human trafficking, trafficking in government-funded programs, forced labor in government-affiliated medical services or other sectors, sexual slavery in government camps, or the employment or recruitment of child soldiers: Afghanistan, Burma, China, Cuba, Eritrea, North Korea, Iran, Russia, South Sudan, Syria, and Turkmenistan. Sadly, the first time the European Court of Human Rights had to decide a case of sex trafficking concerned a Russian girl who was found dead on Cyprus (Rantsev v. Cyprus and Russia—no. 25965/04). Russia withdrew from this Court on 11 June 2022. President Vladimir Putin signed two bills into law that unilaterally removed Russia from the jurisdiction of the European Court of Human Rights (ECtHR) as of 15 March 2022. Therefore, all ECtHR judgments from 15 March 2022 onward are no longer legally enforceable in Russia.

14   Cf. Alegre 2022 who argues: 'Despite its spiritual and political origins, attention capture has never really been a purely ideological endeavour; it has always been intertwined with money. The Catholic Church may have provided a spiritual home for billions through the centuries, but its success has rested equally on its ability to split from the humble roots of early Christianity to accrue vast amounts of wealth and power through the devotion of its adherents. And in democratic societies, political power and financial backing are inextricably connected to the ability to persuade people to vote for you, a trick that is increasingly reliant on the science of marketing rather than the art of political ideology. There is big money in mind control. Technologists and ethicists have begun to sound alarm bells about the potency of distraction in the digital world' (Alegre 2022, p. 99).

15   See further about this Epstein case Section 3. Cf, on how QAnon appears to impact even the wife of a US Supreme Court justice, Ms. Ginni Thomas, 'The strongest conspiracy belief systems spring from a grain of truth' (Riggleman and Walker 2022, p. 213).

[16] Such a distinction is oftentimes complicated, inter alia, because of the many 'gray zones' in criminal law itself. That is, the notion of criminality is itself not always clear-cut (Reich 2021; Korf et al. 2018; Passas 2005; Boister 2018). Some government actions, business practices or individual's activities can be 'lawful but awful', as will be explained further in Section 5 about Passas's concept, in that the interfaces between mostly-legal and mostly-illegal actors become fundamentally questionable (Passas 2005). Criminal networks intersect at least with legitimate public and private institutions. Sometimes that is because criminal networks (ab)use the legitimate infrastructure for their illegitimate goals (Tops and Tromp 2017; Huisman and Kleemans 2017). However, oftentimes it is because those same legitimate public and private institutions are not 'infiltrated' but rather themselves develop improper dependencies (Tjeenk Willink 2021; Thompson 2018; Lessig 2015; Passas 2005; Reich 2021). As an effect, respected and legitimate actors do not only have to be crime victims or become 'intruded' or 'corrupted', but rather can and often are corrupting themselves (Passas 2005). This means that a clear line cannot always be drawn between unethical or unknowingly enablers, on the one hand, and organized crime, on the other.

[17] The latter are redolent of the "scare" or "threat" of white slavery (Allain 2017). Compared to the explanation of Riggleman, *supra* note 15, that emphasis on the realistic and actually needed protection of women and children is reflected in the name of the Protocol to Prevent, Suppress and Punish Trafficking in Persons to the United Nations Convention on Organized Crime with its reference to 'Especially Women and Children'.

[18] As the latter was mob violence, it is important to also reference Alegre 2022 who explains: 'The social psychologist Jonathan Haidt calls it 'the hive switch', the collective psychology that takes over when we gather together for a joint purpose. It is what Aldous Huxley called 'herd poisoning', which makes the individual escape 'from responsibility, intelligence and morality into a kind of frantic, animal mindlessness' (Alegre 2022, p. 92).

[19] After, on 18 July 2016, a teenage refugee attacked passengers on a train near Würzburg with an ax and a knife, on 21 July 2016, an 18-year-old German shot and killed nine people in Munich. The Munich killer said he wanted to copy the Norwegian right-wing extremist and mass murderer Mr. Anders Breivik (Cf. Sauerbrey 2016).

[20] This also goes to its pervasiveness, cf. Alegre 2022 who cites Albert Speer who stated: 'Through technical devices such as radio and loudspeaker 80 million people were deprived of independent thought' (Alegre 2022, p. 95).

[21] Cf. Alegre 2022 who elucidates: 'These [propaganda and brainwashing techniques] were the precursors to the memes, viral clips and fake news of the twenty-first-century authoritarian. And he [Huxley] warned that the dictators of the future would combine these techniques with the non-stop distractions that in the West were already drowning out rational discussions about liberty and democracy in a sea of irrelevance' (Alegre 2022, p. 98).

[22] Letter from US Senators Ben Sasse, Richard Blumenthal, Ted Cruz and Marsha Blackburn to the Inspector General of the US Justice Department (2 December 2019), available at: https://www.sasse.senate.gov/public/index.cfm/2019/12/ben-sasse-to-doj-inspector-general-finish-the-epstein-investigation (accessed on 22 November 2022).

[23] e.g., This complexity is exemplified by inter alia the fact that human trafficking has the longest definition in the Dutch criminal code (e.g., Article 273f Sr; cf. Article 3a of the Palermo Protocol on human trafficking).

[24] e.g., As the modern-day interpretations of the articles regarding slavery, enslavement, servitude and related phenomena, because the term of human trafficking was not common at the time, in Article 4 of the Universal Declaration of Human Rights (UDHR), Article 8 of the International Covenant on Civil and Political Rights (ICCPR) and Article 4 of the European Convention on Human Rights (ECHR). Later also referred to in Conventions like the one on the rights of the child and the prevention of discrimination against women.

[25] e.g., police statistics are usually based on filed police reports and not every victim reports a crime they were victimized by, while global data, such as from UNODC or as gathered in the US TIP reports, are taken from reports by governments based on contact of victims with authorities.

[26] Less than 1% of illegal proceeds derived from human trafficking are confiscated or frozen, according to the United Nations Office on Drugs and Crimes (UNODC 2011a). I use this estimation, while I realize its limitations to which I also want to bring your attention. Even within countries there is a lot of uncertainty about even the definition of crime, so that a global estimation of profit is inevitably hindered by many assumptions. But nonetheless it gives us a—though imperfect—indication of the money potentially made through this crime. In 'the European Union and several other developed countries' the income of sex trafficking has been estimated to be 23.5 billion euros (ILO 2014).

[27] To make the overall sum of money 'earned' by human traffickers of $150 billion worldwide more tangible, this is more money than the combined 2021 profits of four large companies: Meta (formerly known as Facebook), Disney, Starbucks, and Nike. This amount is also comparable to the economy of Morocco in 2022 and Kuwait in 2021. Also, it is some $11 billion more than the estimated wealth of Mr. Bill Gates in 2021.

[28] In 2020, according to UNODC, worldwide 6% of victims were subjected to forced criminal activity (UNODC 2011b). Cf, on one such forcibly committed crime, i.e. the forced couriering of drugs, ECtHR 16 February 2021, *V.C.L. and A.N. v. the United Kingdom -* Applications nos. 77587/12 and 74603/12, 16 February 2021.

[29] This critique can be added to the fact that the $300 billion made through drug crime is based on street value rather than raw materials which is remarkable because this is the only crime for which calculations are based on price inflation; cf. Wainwright (2017).

30    This is not to say that globally including in Europe these principles like the non-prosecution and non-puishment principles of victims who are mistaken for perpetrators ('only') are always complied with in practice, as UNODC has explained, for example in its report, UNODC, ICAT, The Inter-Agency Coordination Group against Trafficking in Persons, Special Issue 8, available at: https://www.unodc.org/documents/human-trafficking/ICAT/19-10800_ICAT_Issue_Brief_8_Ebook.pdf (accessed on 22 November 2022).

31    The US State Department's annual TIP Report reviews responses by governments to combat human trafficking worldwide.

32    The description of the Epstein case with a view to explaining its corrosive effect is especially important because such human trafficking cases evidently stay hidden for a long time, at least insofar as their full scope and networked form are concerned. Due to the limited scope of this article, however, not all aspects of this large-scale sex trafficking case can be examined in full. Also, this would be impossible at present, given that, despite Mr. Epstein's death, some elements are still being investigated in several jurisdictions around the world. The below description therefore focuses on the legal facts and procedure followed in the Epstein case that indicate how human trafficking can have a corrosive effect on democracy and the rule of law. Case details have been derived from, inter alia, Unites States Court of Appeals for the eleventh Circuit, case No. 19-13843, re: Courtney Wild, D.C. Docket No. 9:08-cv-80736-KAM, p. 5, available at: https://www.courthousenews.com/wp-content/uploads/2021/04/wild-epstein-ca11.pdf (accessed on 22 November 2022); United States District Court Southern District of New York, United States v. Ghislaine Maxwell, Defendant, Sealed Indictment, 20 Cr. 330, available at https://www.justice.gov/usao-sdny/press-release/file/1291491/download (accessed on 22 November 2022). Cf. https://www.justice.gov/usao-sdny/pr/ghislaine-maxwell-sentenced-20-years-prison-conspiring-jeffrey-epstein-sexually-abuse (accessed on 22 November 2022). Cf. Brown (2021).

33    Nine years before the Florida investigation, two sisters appear to have already reported Epstein's crime or crimes that was/were committed in New York, cf. Baker (2019).

34    By then the state prosecutor, according to the police, had already changed his position of seeking to prosecute Mr. Epstein. e.g., 'Privately, Recarey and Reiter were baffled by the change in Krischer's [the Palm Beach Florida state attorney's] approach to the case. He went from 'let's get him' to 'why do you want to subpoena those records?', Recarey recalled'. (Brown 2021, p. 83). Cf. Patterson et al argue that Krischer had been accused of sexual misconduct in 1992 (Patterson et al. 2020, p. 160).

35    This non-prosecution agreement listed the following federal crimes: (1) using and conspiring to use a facility of interstate commerce to persuade, induce, or entice minors to engage in prostitution, in violation of 18 U.S.C. §§ 2422(b), 371, and 2; (2) traveling and conspiring to travel in interstate commerce for the purpose of engaging in illicit sexual conduct with minors, in violation of 18 U.S.C. § 2423(b) and (e); and (3) recruiting, enticing, and obtaining a minor to engage in a commercial sex act, in violation of 18 U.S.C. §§ 1591(a)(1) and 2. The Agreement extended immunity to Epstein's named co-conspirators, 'Sarah Kellen, Adriana Ross, Lesley Groff, [and] Nadia Marcinkova', as well as 'any potential co-conspirators' of Epstein's. The latter are thus unnamed possible co-conspirators.

36    'Acosta would later contend that he agreed to give Epstein federal immunity from sex trafficking charges based on the unlikely success that prosecutors felt they would have at trial. Even with the little bit that I [Brown] knew about the case in 2016, this never made sense to me. After all, immunity is a benefit granted in exchange for something else of value to prosecutors. What, if anything, did federal authorities get for giving Epstein and his co-conspirators — both named and unnamed— immunity?' (Brown 2021, p. xiii).

37    The introduction of prostitution charges —rather than trafficking charges—, may have put additional pressure on victims to not press for prosecution of Mr. Epstein and his co-conspirators because prostitution was at the time an illegal offense in Florida. Cf. 'Belohlavek [the female prosecutor] was also concerned that, under state law at the time, minors as young as fourteen could be prosecuted for prostitution, meaning the girls could be charged'. (Brown 2021, p. 83).

38    Cf. Berman: 'We agreed that our case could not be a rehashing of the Florida investigation. It should focus on assaults committed at the New York mansion, and we should try to find New York victims who were not interviewed in the prior investigation. While the non-prosecution agreement did not bind SDNY, because we did not sign it, the Miami US Attorney's Office was bound by it. If our case was somehow perceived as a handoff from the Florida prosecutors or a carbon copy of that investigation, it might be thrown out. The last thing we wanted was to make a mistake and give Epstein another get-out-of-jail-free card. Thus, we agreed there would be no coordination with the Florida prosecutors'. (Berman 2022, pp. 153–54).

39    Brown (2021, pp. xi–xii). Mr. Epstein also owned an apartment in Paris, France, and was there before his arrest in the US (Berman 2022, p. 157). The SDNY was worried about Epstein staying in France, because 'In contrast to other countries in western Europe, France is problematic when it comes to extraditing criminal defendants to the United States. The most famous example involves the film director Roman Polanski, who fled there in the late 1970s after being charged with raping a thirteen-year-old girl in Los Angeles and has been out of reach of American law enforcement ever since'. (Berman 2022, pp. 157–58).

40    According to the Unites States Court of Appeals for the eleventh Circuit 'Following a tip in 2005, the Palm Beach Police Department and the FBI conducted a two-year investigation of Epstein's conduct' (p. 3). The Dissenting opinion of Judge Branch, who is joined by Martin, Jill Pryor and Hull, makes this even more explicit 'following a 2005 report by the parents of a 14-year-old girl that then 52-year-old billionaire Jeffrey Epstein sexually abused their daughter, local Florida authorities—and later the FBI—began investigating Epstein'. (p. 100). 'In June 2008, Epstein pleaded guilty to the state crimes as agreed and was sentenced to 18 months' imprisonment, 12 months' home confinement, and lifetime sex-offender status'. (p. 5), in case No. 19-13843, re:

Courtney Wild, D.C. Docket No. 9:08-cv-80736-KAM, available at: https://www.courthousenews.com/wp-content/uploads/2021/04/wild-epstein-ca11.pdf (accessed on 22 November 2022).

[41] Formerly, Ms. Virginia Roberts who settled with Mr. Epstein in 2009. Earlier, in 2008, Mr. Epstein had also already settled with victims' representative Mr. Brad Edwards.

[42] e.g., Case 19 Civ. 3377 (LAP), Virginia L. Giuffre, Plaintiff, v. Alan Dershowitz, Defendant, 10-16-2019, available at: https://casetext.com/case/giuffre-v-dershowitz (accessed on 22 November 2022). Mr. Dershowitz countersued. Also, on 15 November 2022, two anonymous women who accuse the late Mr. Jeffrey Epstein of sexual abuse have filed separate civil lawsuits against JP Morgan Chase & Co. and Deutsche Bank AG, claiming the big banks enabled and benefitted financially from Epstein's alleged sex trafficking operation, available at: https://edition.cnn.com/2022/11/25/business/epstein-accusers-sue-jp-morgan-deutsche-bank-alleged-sex-trafficking/index.html (accessed on 22 November 2022). (Beech 2022).

[43] Both her series in the Miami Herald and her book, from which citations are provided, are entitled *Perversion of Justice*. For example, Brown argues 'The Jeffrey Epstein story epitomizes our nation's lopsided system of justice, and how victims of sexual assault, especially those who are young and poor, are discarded, shamed, and mistreated by the very people who were supposed to protect them.' (Brown 2021, p. xiv). She also compares the regular course of justice with the one in the Epstein case: 'Recarey [the lead detective on this case in South Florida], in a fatherly tone, tried to reassure them [the victims] that they would indeed be safe and that Epstein would be arrested, telling them: "It doesn't matter how much money you have or how many connections you have, if you commit a crime then you will be punished. That's the way our justice system works".'. (Brown 2021, p. 16). She adds: 'One of the many mysteries of the Epstein case is how he got away with such flagrant sex crimes at a time when the FBI was cracking down on child exploitation and putting away men for decades for far lesser sex crimes. In 2006, the Justice Department under President George W. Bush had launched a task force focused on sex crimes against children. Hundreds of arrests and prosecutions happened during these years. Although the effort focused largely on child pornography, combating human trafficking was also one of its aims, even though I [Brown] later came to learn that trafficking, at least back then, was seen by law enforcement as a largely foreign phenomenon perpetrated mostly by black and brown people who came from other countries. The Justice Department didn't seem to fathom that sex trafficking could be a pervasive crime committed by well-to-do and powerful people in the United States. Or that pornography—especially child pornography—was fast becoming a multibillion-dollar worldwide industry'. (Brown 2021, pp. 17–18).

[44] On the issue of celebrity, former R&B star R. Kelly was sentenced to 30 years in prison on 28 June 2022 (the same day when Ms. Ghislaine Maxwell was sentenced to 20 years imprisonment, and the sixth January 6th hearing took place), for racketeering and sex trafficking, charges stemming from nearly 30 years of allegations that he physically and sexually abused women and minors. Available at: https://www.justice.gov/usao-edny/pr/r-kelly-convicted-all-counts-federal-jury-brooklyn (accessed on 22 November 2022).

[45] Letter from Sen. Sasse to Attorney General Barr (13 August 2019), available at: https://www.sasse.senate.gov/public/_cache/files/1b35b35b-5b79-4447-9cab-5c176cb8fa50/08-13-19-lettertowilliambarr.pdf (accessed on 22 November 2022).

[46] United States District Court Southern District of New York, United States v. Ghislaine Maxwell, Defendant, Sealed Indictment, 20 Cr. 330, available at https://www.justice.gov/usao-sdny/press-release/file/1291491/download (accessed on 22 November 2022). Cf. https://www.justice.gov/usao-sdny/pr/ghislaine-maxwell-sentenced-20-years-prison-conspiring-jeffrey-epstein-sexually-abuse (accessed on 22 November 2022).

[47] e.g., Robertson et al. (2019). Cf. "Jean-Luc Brunel: three former models say they were sexually assaulted by Jeffrey Epstein friend". The Guardian. ISSN 0261-3077. Retrieved 21 March 2020.

[48] New York State, Department of Financial Services, in the matter of Deutsche Bank AG, Deutsche Bank AG New York Branch and Deutsche Bank AG Trust Company of the Americas, consent order under New York Banking Law §§ 39 and 4, available at: https://www.dfs.ny.gov/system/files/documents/2020/07/ea20200706_deutsche_bank_consent_order.pdf (accessed on 22 November 2022). Cf. 'The story also alarmed Epstein's financial institution, Deutsche Bank, where some employees had already been raising flags about Epstein's accounts'. (Brown 2021, p. 282). Cf. Stewart (2020).

[49] Patterson et al. (2020), who also includes original transcripts of interviews by, seemingly contemporaneous notes taken, or a letter written to the state prosecutor by criminal investigators like Reiter and interviews with other investigative journalists like Vicky Ward who profiled Mr. Epstein and (persons who know the) women who consider themselves friends with Mr. Epstein like Eva Dugin, Ana Obregón and Ghislaine Maxwell. One of the victims already identified others as "sex slaves" even though that same approach was ultimately not take by at least federal prosecutor Mr. Acosta (Patterson et al. 2020, p. xvii and p. 37).

[50] This literature and case law review covered all the English-language major university library and google scholar databases without time limitation using key words of "effect(/s) of" "damage done by" "the Epstein case" or "Epstein (sex) trafficking case" on "the rule of law", "(US) (U.S.) (United States) (United States of America) (American) (criminal) justice system", "the United Kingdom", UK, U.K." and "France", "Deutsche Bank", as well as "abuse of justice", "perversion of justice", "abuse of (criminal) procedure", "(public) corruption", "Crime Victims' Right Act" as well as "(Ghislaine) Maxwell", "(Jeffrey or Jeff) Epstein", "(Jean-Luc) Brunel", "Palm Beach Police Department", "U.S. Attorney's Office in South Florida", "Palm Beach Sheriff", "Special Victims Unit", "U.S. Attorney's Office in the Southern District of New York", "SDNY", "Public Corruption Unit", "St. James", "St. Thomas", and based on Brown's 2021 book: "(Virginia) Giuffre/Roberts", "(Courtney) Wild", "(Sarah) Ransome", "(Maria and/or Annie) Farmer", "(Michelle) Licata", "(Alfredo) Rodriguez", "Sarah Kelsen", "Nadia Marcinkova/Marcinko"

"(Lesley) Groff", "(Adriana) Ross", "(Alex) Acosta", "(Juan) Alessi", "(Ann Marie/ Maria) Villafaña", "(Lanna) Belohlavek", "(Carolyn) Bell", "(Ric) Bradshaw", "(E. Nesbitt) Kuyrkendall, "(Barry) Krischer, "(Joseph or Joe) Recarey, "(Michael) Reiter, "(Daliah) Weiss", "(Jason) Weiss", "(Matthew) Menchel", "(Andrew) Lourie", "(William) Harvey", "(Preet) Bharara", "(Geoffrey) Berman", "(Jeffrey) Sloman", "(Jack) Goldberger", "(Cy) Vance", "(Alan) Dershowitz", "(Mike) Cernovich" "(Ken/Kenneth) Starr", "(Martin) Weinberg", "(Lilly Ann) Sanchez", "(Guy) Fronstin", "(Gerald) Lefcourt", "(Jay) Lefkowitz, "(Paul) Cassell", "(Brad) Edwards", "(Spencer T.) Kuvin", "(David) Boies", "(Sigrid) McCawley", "(Jeff) Hermann", "(Luke) McSorley", "(Bruce) Reinhart", "(Jan) Ford)", "(John) de Jongh", "Prince Andrew". Cf. Academic research does cover the Epstein case in relation to how white collar investigative techniques could have better examined its networked form as well as how money and inequity disturbs the balance of power in the US criminal justice system while recognizing some influence of autocratic actors, but does not yet extend to researching its potential corrosive effect, neither from within nor outside of liberal democracies. Cf.. (Binder 2022; Jackson 2019).

51  For example, in the sentencing memo of the prosecutor in the case of the co-conspirator Ms. Maxwell no reference has been made to the impact of this case on the rule of law, while the prosecution did take into account the aspect of the need to promote respect for the law. Cf. Berman adds about Maxwell: 'The evidence at trial proved that Epstein had given Maxwell €30 million. It wasn't credible that he paid that simply to compensate her for managing his houses'. (Berman 2022, p. 180). That aspect has been used to sustain the reason for asking the court to impose a fine, which Ms. Maxwell argued she would not be able to pay due to lacking funds. Cf. https://www.courthousenews.com/wp-content/uploads/2022/06/maxwell-government-sentencing-memo.pdf (accessed on 22 November 2022).

52  Cf. Even only for the SDNY that did prosecute the Epstein case, during Mr. Berman's term alone, there was no priority given to human trafficking beyond sex trafficking. Berman relates the following convictions: cases involved a sex-trafficking ring where girls, some as young as thirteen years old, were recruited from the child welfare system at Hawthorne Cedar Knolls, a now-shuttered residence and treatment center for at-risk children in Mount Pleasant, Westchester County; the nine trey gang; Larry May; and Peter Nygard. While the SDNY prioritized (or perhaps still prioritizes) sex trafficking but not yet other forms of human trafficking, in the aforementioned Larry May case also human trafficking for the purpose of labor exploitation was included in the conviction because boys were required to conduct forced labor for the convicted person (Berman 2022, pp. 125–81).

53  Cf. Section 2. With the notable exception of Shelley (1995), although even this research does not deal with human trafficking specifically—or the Epstein case for that matter—but rather refers to the corrosive effects on states caused by transnational organized crime more generally. Another exception is Snyder (2018) who does not refer to human trafficking or any other specific crimes either, but does mention the effect of illicit financial flows more broadly, as is explained in Section 4.2.

54  The description of these corrosive factors is inevitably hampered by the fact that it will take years –if not decades– to fully understand the Epstein case in its networked form, in part because the integrity of public and private institutions as well as their representatives who should have investigated these crimes in full seem to have been, at best, manipulated or, at worst, have gotten eroded themselves. However, it is possible to trace some of the most relevant corrosive factors from the Epstein case based on literature that seeks to explain democratic deconsolidation worldwide, as done here in line also with the UNODC's practice as explained in footnote 55 below (e.g., Diamond 2019; Mounk 2019; Levitsky and Ziblatt 2018; Guriev and Treisman 2022).

55  The UNODC which examined human trafficking-related corruption examined consequences such as fraudulent travel or identity documentation made by bribed customs officials or actual illegal border crossings facilitated by agents who turned a blind eye (UNODC 2011a). From such consequences, the potential corruption can be inferred. The same will be done here. Cf. *supra* note 54.

56  'Epstein didn't need to use flamboyant charisma or to showboat his smarts to build his international network of influential people; he did that the old-fashioned way—with his money'. (Brown 2021, p. 128).

57  Cf. Patterson et al. who argue that this approach changed in or around 2002: 'Another source, one who had worked with Epstein, said, "He's reckless, and he's gotten more so. Money does that to you. He's breaking the oath he made to himself—that he would never do anything that would expose him in the media. Right now, in the wake of the publicity following his trip with Clinton, he must be in a very difficult place".'. (Patterson et al. 2020, p. 144).

58  'I [Brown] also learned that Epstein likely conducted video surveillance in every home he owned. As insurance, he probably had tapes and photographs of important visitors—mainly men—in compromising situations. Whether that was true or not, even the possibility that he had blackmail material was enough motive for many powerful people to do everything possible to cover up Epstein's crimes'. (Brown 2021, p. xiv). Cf. Patterson et al. who explais that two surveillance cameras in the West Palm Beach house were installed by the local police itself, though the images could only be seen by Mr. Epstein (Patterson et al. 2020, pp. 76–77).

59  'One of his specialties was helping the super wealthy—as well as himself—to avoid paying taxes. Yet he was never in the Forbes 400 list of the wealthiest Americans, largely because the magazine was never able to determine the true size of his fortune'. (Brown 2021, p. 128).

60  e.g., 'This seemed suspicious to Michelle [Licata, one of Epstein's named victims]. So she hired another lawyer, not paid by Epstein. As a mediation, Epstein's lawyers gave her an offer she would have to accept or reject immediately. She was troubled when her attorney informed her she couldn't talk to her parents about the settlement first. By then, Epstein's lawyers had already sent a message that they intended to destroy her and her family'. (Brown 2021, pp. 111–12).

61    Cf. Berman argues: 'Even now, it is mind-boggling—and enraging—to look back on the seven-page document [the non-prosecution agreement]. If I had to pick out the most astonishing passage, it might be this one: "The parties anticipate that this agreement will not be made part of any public record. If the United States receives a Freedom of Information Act request or any compulsory process commanding disclosure of the agreement, it will provide notice to Epstein before making that disclosure".' (Berman 2022, pp. 151–52). About the impossibility to investigate any prosecutorial or other corruption in this regard, Berman states: 'What we found out, rather quickly, was that the statute of limitations on any potential corruption charges related to the non-prosecution agreement had long since expired' (Berman 2022, p. 154).

62    Citation from Unites States Court of Appeals for the eleventh Circuit, case No. 19-13843, re: Courtney Wild, D.C. Docket No. 9:08-cv-80736-KAM, p. 5, available at: https://www.courthousenews.com/wp-content/uploads/2021/04/wild-epstein-ca11.pdf (accessed on 22 November 2022).

63    Initially Judge Marra ruled that this lack of informing the victims about the NPA had violated the rights of the petitioning victim Ms. Courtney Wild under the Crime Victims' Rights Act, 18 U.S.C. § 3771 (VCRA), but the cited court's majority did not reach that conclusion. Ibid cited in *supra* note 62. Berman argues about the impossibility of granting monetary compensation for the victims: 'Justice was imperfect in other ways as well. We had sought forfeiture of Epstein's Upper East Side mansion, the scene of so much of his abuse, as part of the indictment. The money could have gone out to victims. But without a conviction, ownership reverted to his preferred heirs. The town house was later sold to a Wallstreet executive for €51 million' (Berman 2022, p. 173).

64    'Two weeks later, Epstein donated money for the firearms simulator. Reiter [the local police chief] assigned someone to start looking into purchasing the equipment. But shortly thereafter, he learned about the investigation into Epstein and put a hold on the purchase. As time went on with the case, Reiter began to suspect that Epstein's altruistic endeavors were aimed more at influencing the police than they were at helping them. Reiter was smart enough not to return the money immediately. He didn't want to alert Epstein or jeopardize their investigation' (Brown 2021, p. 125). '"His enthusiasm in making contact and in finalizing the donation was somewhat suspicious, different in manner than when he made previous donations and suggested that he may have become aware of the investigation at that time", Reiter wrote in a later report' (Brown 2021, p. 126).

65    This reference to *at least* a decade has been made, because it is hard to know when the Epstein human trafficking operation began. As Brown states: 'It's difficult to know how and when Epstein's scheme began. What is known is that in 1998, Epstein's then girlfriend, the British socialite Ghislaine Maxwell, began visiting colleges, art schools, spas, fitness centers, and resorts in and around Palm Beach County [Florida, the United States], under the guise that she wanted to hire young and pretty masseuses or "assistants" to come to Epstein's home and work for him' (Brown 2021, p. 10). Also, 'Epstein promised to rescue them, but at a cost: not only were they expected to perform for him sexually, but in some cases, they were pressured to have sex with other men old enough to be their grandfathers' (Brown 2021, p. 10). 'Epstein's houseman, Juan Alessi, was ordered to drive Maxwell from resort to resort for her to hand out business cards and recruit massage therapists for Epstein. Alessi was skeptical of her motives, especially when the first who began coming to the house looked as young as Alessi's daughter' (Brown 2021, p. 10).

66    Some of these influences of money on politics and from politics on the criminal justice system in the US may not be felt –at least not to that extent– in other liberal democracies that do not elect their sheriffs, prosecutors and/or judges, for example, such as in the Netherlands. However, dirty and dark money does not have to negatively impact elections 'only', of course, as it can be used in other corrupting practices such as (threats of) bribes, campaign financing so as to exert pressure on the legislative branch and (threats to) use violence against criminal justice actors. Cf. Brown who argues that some such influence was at least perceived in Palm Beach: 'Recarey [the investigating agent] knew full well that the grand jury schedule set by prosecutors—and Epstein's lawyers—was designed to fail, and, in an interview, he told me how and why he believed that they were throwing the case. It was part political, he said, because Epstein was a million-dollar donor to the Democratic Party, which controls Palm Beach. Krischer was reminded that Clinton was friends with Epstein, and there were a lot of other political heavyweights also tied to Epstein, including George Mitchell. If Epstein's secrets got out in a big way, it would hurt the party. Krischer was a powerful force in Palm Beach politics, and it was up to him to contain the case' (Brown 2021, p. 90).

67    e.g., then Former President Mr. Bill Clinton whose former top advisor attests he had been on the island St. James where some of the abuse took place and now Former President Mr. Donald Trump who is also alleged to have sexually assaulted some of the victims). In New York Magazine: '"I've known Jeff for fifteen years. Terrific guy", said Donald Trump, fifteen years before he would be elected president. 'He's a lot of fun to be with. It is even said that he likes beautiful women almost as much as I do, and many of them are on the younger side. No doubt about it—Jeffrey enjoys his social life".' (Brown 2021, p. 42). For example, Brown details: 'The truth is there were powerful people on both sides of the political rails—as well as people in the worlds of finance, academia, and science—who were involved with Epstein or, at the very least, complicit with what he was doing' (Brown 2021, p. xiv).

68    e.g., 'The message pads [found in his West Palm Beach house during its search] also contained a who's who of influential people calling Epstein: Donald Trump, Les Wexner, former J.P. Morgan banker Jes Staley, real estate mogul Mort Zuckerman, former Maine senator George Mitchell, and Hollywood producer Harvey Weinstein'. (Brown 2021, p. 81). Cf. 'Maybe they [Epstein's victims, after seeing Acosta's confirmation hearing in which Epstein's name barely came up] had something to say about Acosta running a massive government agency with oversight of human trafficking and child labor laws.' (*Ibid.*, p. 6). Cf. 'William Barr, the attorney general, recused himself from the case because of his ties to the Kirkland firm [apparently not because of his father's hiring of Mr. Epstein for Dalton College and/or potentially knowing him as they were the same age at that time].' (*Ibid.*,

p. 332). Cf. 'What is noteworthy, besides Epstein's unrelenting endeavor to game the system, was that he managed to enlist New York District Attorney Cy Vance's office to help him. Piuckholtz noted that in all her years on the bench she had never seen a prosecutor try to ease the registration burden for a convicted sex offender.' (*Ibid.*, p. 345). Cf. 'Then Assistant U.S. Attorney Alex Rossmiller dropped a bomb. "Any doubt that the defendant is unrepented and unreformed was eliminated when law enforcement agents discovered hundreds of thousands of seminude photographs of young females in his Manhattan mansion on the night of his arrest, more than a decade after he was convicted of a sex crime involving a juvenile," he wrote. Rossmiller also revealed that a phony Austrian passport and millions of dollars in diamonds, some as large as 2.38 carats, together with a pile of cash were found in Epstein's safe. Epstein had used the now expired passport with a fake name to enter countries in the 1980s, including France, Saudi Arabia, and Spain, prosecutors said. Rossmiller also raised the specter of witness tampering, pointing out that Epstein had recently sent payments to two of his 2008 co-conspirators. Rossmiller said it was clear that Epstein was trying to send a message to the two women that they should keep their mouths shut.' (*Ibid.*, p. 346). Cf. 'At the same time that the New York Times was poking into Epstein's science ties, the Wall Street Journal was dissecting Epstein's financial and personal relationships with some of the wealthiest people in the country, including Leslie Wexner; Leon Black; Jes Staley, then a top executive at J.P. Morgan; and Glenn Dubin, the cofounder of Highbridge Capital Management, one of the fastest-growing hedge fund firms in the 2000s.' (*Ibid.*, p. 376)

[69] Cf. 'I [Brown] had been trying for months to carve out some time to visit St. Thomas and take a trip out to Epstein's "Pedophile Island," which was also sometimes called "Orgy Island".' (Brown 2021, p. 355). 'Both Chef James and Island Mike claimed, without proof, that Epstein had the fix in with the former governor of the U.S. Virgin Islands, John de Jongh Jr., and Epstein had even hired the governor's wife, Cecile, to work for him at his St. Thomas-based company, Southern Trust, which was purportedly a data-mining venture. St. Thomas is a poor island, and it wouldn't take a lot to get the local politicians to look the other way when it came to doing what Epstein wanted' (Brown 2021, p. 356). 'By the time Emily and I [Brown] arrived on St. Thomas, the feds had already raided Epstein's island. I was talking to a woman who worked on the compound briefly, and whose boyfriend was still employed by Epstein at the time of his arrest. The couple told me that, immediately upon Epstein's July 6 arrest, one of his employees, Lesley Groff, arrived from New York and began dismantling the camera system on Epstein's island. His computers were moved, as well as boxes of unknown items. They also said a giant steel safe in his office was carted away. (Groff's spokeswoman said Groff was not on the island after Epstein's arrest, and Groff was not aware of any cameras on the island). The couple was afraid, however, and wouldn't talk on the record. They claimed they had been required to sign nondisclosure agreements with a one-million-dollar penalty. The male employee said when the FBI agents arrived, all the employees were asked to leave. "By then, all the cameras were already gone," the employee said. "We were surprised that they waited so long to raid the island".' (Brown 2021, pp. 356–57).

[70] The Frederick Douglass Trafficking Victims Prevention and Reauthorization Act was passed on 28 July 2022 in a 401-20 vote. Also, the other way around, the Epstein case still translates into many (QAnon) conspiracies including about not only his suicide—also because Mr. Brunel's suicide similarly happened by way of hanging with bed linen as Mr. Epstein's—but also how the Clintons—rather than former president Trump or others—are supposedly connected thereto.

[71] Cf. Berman who states: 'I'm aware of all the conspiracy theories and have looked at them, searching for even a trace of plausibility. There isn't any' (Berman 2022, pp. 164–65).

[72] Cf. 'Her [Ginni Thomas's, wife to US Supreme Court justice Clarence Thomas's] text went on to suggest the issues with voting machines were somehow connected to human trafficking, a central fear for believers in the QAnon conspiracy theory' (Riggleman and Walker 2022, p. 18).

[73] The Australian bank Westpac has agreed to conclude a record transaction of $1.3 billion to settle legal action over money laundering and human trafficking for the purpose of child exploitation allegations levelled against it by the financial intelligence agency, Austrac. Westpac is also one of the largest banks in New Zealand. Available at: https://www.austrac.gov.au/news-and-media/media-release/austrac-and-westpac-agree-penalty (accessed on 22 November 2022). This literature and case law review covered all the English-language major university library and google scholar databases without time limitation using key words of key words of "effect(/s) of" "damage done by" "(sex/human) trafficking" on "the rule of law", "Australia", "Philippines", "Westpac", "Austrac", "live streaming of child sex shows and offering children for sex", "conviction for child exploitation", "travel", "flights", "Peter Scully", "Brian Hartzer", "Customer 1" paid $136,000 over five and a half years", "Customer 5", paid $75,000 over four years and repeatedly travelled to south-east Asia". Cf. (Butler 2019).

[74] Although the first case before the European Court of Human Rights concerning sex trafficking involved a Russian national who was found dead on Cyprus (ECtHR, Rantsev v. Cyprus and Russia, Application no. 25965/04, 10 October 2010), Russia has now left the Council of Europe, thereby leaving its nationals without protection from this Court.

[75] This lack of research can be explained somewhat because of the surrounding secrecy because, while it is well known that Russia's current and former officials and oligarchs are known to store their illegal gains in established democracies, the vast majority of Russian-owned foreign assets are shrouded in secrecy (e.g., Snyder 2018; Bullough 2018; Burgis 2020; Vogl 2022). The sanctions on Russian companies and individuals by the United States and the European Union to which the Netherlands is a member pierce that veil of secrecy to some extent. These sanctions may have an effect on human trafficking given that by estimation 90% of human trafficking happens in the private sector but benefits the Russian government (e.g., Investigative journalists explore leaks like the Panama and Paradise Papers as well as the effects of US and EU sanctions on

the Russian state and its oligarchs so as to examine what is known about secret finances of oligarchs in the West, available at: https://www.icij.org/investigations/pandora-papers/as-the-west-takes-aim-with-russian-sanctions-heres-what-we-know-about-oligarchs-secret-finances/) (accessed on 22 November 2022). Nonetheless, (foreign) human traffickers or oligarchs can still often hide that amount of money through shell companies and anonymous ownership. By estimation, the amount of current offshore money is $7 trillion, with 10% of world GDP held offshore (UNDESA 2020). Hence, dirty and dark money has its tentacles in a network of liberal democracies and other states around the globe in all the aforementioned ways. This is why further initiatives including by the Biden Administration to create a global register for ultimate beneficial ownership intends to prevent financial secrecy and promote financial integrity in the future can also assist such much needed research (e.g., Memorandum on Establishing the Fight Against Corruption as a Core United States National Security Interest, 3 June 2021, Presidential actions, available at: https://www.whitehouse.gov/briefing-room/presidential-actions/2021/06/03/memorandum-on-establishing-the-fight-against-corruption-as-a-core-united-states-national-security-interest/) (accessed on 22 November 2022). Such research will then also have to extend to the (corrupted) enablers in Western democracies who assist organized crime (cf. Vogl 2022; Guriev and Treisman 2022). All of this is important, without losing sight of the fact that also some of the actions of Western democracies themselves have a damaging effect on democracy and the rule of law including on the international rule of law such as the invasion of Iraq by the United States.

76    Justice Breyer's dissenting opinion in *McCutcheon et al v. Federal Election Commission*, available at: https://www.supremecourt.gov/opinions/13pdf/12-536_e1pf.pdf (accessed on 22 November 2022).

77    Cf. Frank Vogl, the founder of Transparency International, in his 2022 book: 'Across the world, leaders of authoritarian governments, and their cronies, are robbing their people. These leaders are kleptocrats and they are pocketing staggering sums of cash, which they move through the world's financial system into investments in the wealthiest Western nations. These crimes perpetrated by kleptocrats governing Russia, China, Iran, Egypt, Hungary, Nigeria, and many more nations not only impoverish their own citizens but all of us. More gallingly, we are assisting them in their greed and their grand corruption. Even more worrying, we are complicit in their quest for ever greater power. Central to Western complicity with kleptocrats and their associates across the globe are the armies of financial and legal advisors, real estate and luxury yacht brokers, art dealers and auction house managers, diamond and gold traders, auditors, and consulting firms, based in London and New York and in other important global business centers, who aid and abet the kleptocrats in return for handsome fees– these are the enablers. They are motivated not only by the widespread failures of law enforcement across the Western democracies to impose punishments that are sufficient to serve as meaningful disincentives. At the major banks, for example, who have been prosecuted at times for multi-billion dollar laundering of dirty cash, not a single chairman or chief executive officer has personally faced criminal charges for such activities, while the fines that are agreed to settle legal actions appear, quite simply, to be viewed by bankers as just the costs of doing business' (Vogl 2022, pp. 1–2). Vogl's book, however, refers mostly to the crime of corruption but does not examine what underlying crimes—such as human trafficking—provide the funding for such bribes or other means to ensure abuse of power. Oftentimes there is only a reference to organized crime at the aggregate level, which is oftentimes followed by an overemphasis on drug crimes (Vogl 2022).

78    Cf. Guriev and Treisman who explain how the new class of autocrats which they call 'spin dictators' because, just as conspiracies, they have changed into a more subtle and insidious group of dictators who use spin rather than the less sophisticated propaganda and force from previous centuries, 'recruit and corrupt Western elites much as they co-opt their own educated class.' (Guriev and Treisman 2022, p. 147). 'Today's spin dictators turn Tito's double game into an art. They participate in Western institutions in order to extract benefits, exploiting the design flaws and weaknesses of these bodies. They trade with Western countries, while denouncing them. They recruit networks of corrupt partners in the West, simultaneously pursuing concrete goals and eroding Western cohesion. At the same time, they make hypocritical speeches about the West's hypocrisy' (Guriev and Treisman 2022, p. 152). Cf. 'Leaders of Western democracies have done little to guard against the backflow of 'freed capital flows, deregulated business, and opened trade with their former adversaries' (Guriev and Treisman 2022, p. 209).

79    Cf. 'Russia in the 2010s was a kleptocratic regime that sought to export the politics of eternity: to demolish factuality, to preserve inequality, and to accelerate similar trends in Europe and the United States. This is well seen from Ukraine, where Russia fought a regular war while it amplified campaigns to undo the European Union and the United States' (Snyder 2018, p. 11). 'In this particular way, the American politics of inevitability, the idea that unregulated capitalism could only bring democracy, supported the Russian politics of eternity, the certainty that democracy was a sham' (Snyder 2018, p. 262).

80    Cf. Guriev and Treisman who explain 'Today's autocracies pose new challenges to the democracies of the West' (Guriev and Treisman 2022, p. 205). Several of these autocracies have a policy or pattern of human trafficking (TIP report 2022).

81    Cf. Reich, who argues: 'History shows that oligarchies cannot hold on to power forever. Oligarchies are inherently unstable. This was as true in ancient Rome as it was in America's antebellum South, where fewer than four thousand families owned about a quarter of America's capital in the form of enslaved human beings' (Reich 2021, p. 166).

82    Previous examples are the so-called accidental murders outside the 'criminal milieu' (*vergismoorden*) and the detection of a full-fledged torture chamber in the province of Brabant in the Netherlands.

83    In Dutch: derived from the noun of 'to undermine': *ondermijnen*, the noun *ondermijning*, which is a shorthand for *rechtsstaat-ondermijnende criminaliteit*). This is not to say that there is not some debate between Dutch scholars as to what defines this as of yet, as is the case for organized crime itself, of undefined category of crimes. See below in this section (Section 5.1).

84    In keeping with Snyder's explanation, this article follows the meaning given to 'oligarchy' as used by Aristotle, meaning rule by the wealthy few; the word in this sense was revived in the Russian language in the 1990s, and then, with good reason, in English in the 2010s. (Snyder 2018, p. 11).

85    The indices measuring the quality of democracy—such as those of the Economist Intelligence Unit, Freedom House, Varieties of Democracy, and International IDEA—includes disrespect for the rule of law in addition to crackdowns on civil liberties, populism's rise, declines in popular trust in politics and political parties, and falling democratic participation.

86    Cf. Alegre, who states: 'Democracy may not have done much for Socrates' freedom of opinion, but throughout the twentieth century, more and more countries adopted democracy and respect for human rights as the basis for their political and governance structures. To support these shifts, electoral laws developed to protect the democratic ideal of free and fair elections. Limitations on the levels or sources of funding for political campaigns in many countries reflect fears that either oligarchic elites or foreign powers could exert undue influence on the opinions of the electorate. In liberal democracies, there are commonly laws that prevent campaign tactics that might either entice voters to turn out or discourage others from going to vote. These safeguards are crucial to prevent corruption in the election process through votes being either bought or suppressed' (Alegre 2022, pp. 170–71).

87    Consequently, this article also juxtaposes the notions of liberal and illiberal democracies currently advocated by elected leaders who corrode their own democracies such as in Hungary, Turkey, and India.

88    Article 2 (a) 'Organized criminal group' shall mean a structured group of three or more persons, existing for a period of time and acting in concert with the aim of committing one or more serious crimes or offences established in accordance with this Convention, in order to obtain, directly or indirectly, a financial or other material benefit. Article 2 (b) 'Serious crime' shall mean conduct constituting an offence punish able by a maximum deprivation of liberty of at least four years or a more serious penalty. Article 2 (c) 'Structured group' shall mean a group that is not randomly formed for the immediate commission of an offence and that does not need to have formally defined roles for its members, continuity of its membership or a developed structure. There is no definitive list of organized crime, although human trafficking is included as defined in the Protocol to Prevent, Suppress and Punish Trafficking in Persons, Especially Women and Children to the United Nations Convention on Organized Crime (Article 3 (a) Palermo Protocol).

89    The author thanks the two anonymous reviewers for their remarks that 'hate speech' and corruption are more familiar to the audience of this journal.

90    The ideal is that transnational criminal law ensures that the corrosive effects of human trafficking, corruption and money laundering are reflected in laws that criminalize these offenses beyond borders of states, so that one cannot pick and choose the ethical and legal frameworks as a criminal changes country. However, in practice, the reality does not live up to that ideal in all three crime areas, as explained in Sections 2–4. Cf, for the idealized picture on corruption and human trafficking, Alegre (2022, p. 315).

91    The importance of prosecuting not only the proverbial foot soldiers such as those who stormed the Capitol on a whim on 6 January 2021, but rather its planners and those who funded and orchestrated this mob violence from afar also relates to the prevention of class justice (Cf. Rawls 1971). People notice that often lower rungs of criminal networks rather than masterminds are held accountable. This causes erosion of trust in the state because there is a flipside to this too. People do see how supposedly legitimate business, which are oftentimes more responsible for sustaining the (financial) structures of organized subversive crime, usually remain scot-free. Class justice even happens *within* criminal procedure because '(t)he more sophisticated and knowledgeable the criminal, the more valuable is his cooperation and the more benefit he can obtain and offset the punishment which might otherwise have been imposed' (Burgis 2020, citing Judge Glaser, p. 87). So low-level drug dealers, couriers, or money mules who have no information to give to the government, do suffer the sentence which the law requires (Cf. Sandel 1998).

92    In the absence of a universally agreed definition of the term, various terminology describing the notion of 'terrorism' can be found within its outputs. One notable exception though is the example discussed here of article 2 of the International Convention for the Suppression of the Financing of Terrorism of 1999.

93    Article 2 (b) of the International Convention for the Suppression of the Financing of Terrorism of 1999.

94    Currently debates are ongoing whether some aspects of the January 6th attack should be considered stochastic terrorism. While the exact definition has morphed over time, stochastic terrorism has commonly come to refer to a concept whereby consistently demonizing or dehumanizing a targeted group or individual results in violence that is statistically likely, but cannot be easily accurately predicted.

95    Cf. Alegre, but insofar as the freedom to think is concerned: 'No one can enslave another person for any reason. Torture and slavery are prohibited absolutely, as they are anathema to human dignity and an affront to humanity. So too is anything that interferes with our inner freedom' (Alegre 2022, p. 26).

96    Human trafficking can qualify as organized subversive crime and even other offenses like drug trafficking involving forced couriers who are victims of human trafficking can fit that bill. However, this should not be misread to mean that *all* organized subversive crime involves trafficked persons who are forced to work in (sex) labor or commit offenses. But some may be. As a related point, this also does not imply that organized subversive crime cannot be committed without any laundering of illegal proceeds derived from criminality. For example, some offenses will still be committed through violence 'only' and some of the illegal proceeds will exclusively circulate among criminals. However, oftentimes these organized crimes that are specifically committed because of their significant financial or other material gain can be better understood if the damage to

liberal democracies will also be taken into account. As a final qualifying remark, it does not mean either that all organized subversive crime necessarily involves 'upperworld' enablers like tax advisors, lawyers and consultants. Some of this criminality can fully stay within the realm of the criminal milieu. However, it does mean that sometimes human trafficking is coupled with money laundering or corruption so that it becomes necessary to fully understand its contribution to the erosion of democracy and the rule of law by also taking into account its corrosive effect on liberal democracy.

[97] This conclusion should not be misunderstood as if this article argues that perpetrators of human trafficking inevitably commit a crime that does not exploit but even undermines state structures and results in a sense of injustice and a loss of trust by the general public in democracy and the rule of law. For instance, there may be instances where a human trafficker abuses one victim without the intent to or larger effect of eroding liberal democracy more broadly. To give a potential example, usually a trafficker who exploits a single victim domestically in a one-on-one pretend loving relationship merely out of financial gain may not have the larger effect of hurting democracy or the rule of law to the extent needed to legitimately qualify as a corrosive effect on liberal democracies (the poor term developed by the Dutch: 'loverboy'; Bovenkerk and Pronk 2007). While the individual victim and their family or community may consequently lose trust in democracy and the rule of law, oftentimes it will require a larger scale human trafficking operation to have the wider more fundamental corrosive effect on liberal democracy more generally. However, this is not to say that the accumulation of many such cases of one-on-one exploitation cannot create the inadvertent permissive environment that results in erosion of liberal democracies.

[98] Cf. Alegre who bases a similar conclusion about spreading of disinformation on the case of the International Criminal Tribunal for Rwanda insofar as incitement to genocide is concerned: 'Those who control such media are accountable for its consequences' (RTLM media case ICTR)' (Alegre 2022, p. 111) 'The appeal court ruling provides some interesting insights into the way hate speech and incitement are considered in international criminal law. It found that direct and public incitement to genocide was a crime that could be established even if genocide itself did not follow from the incitement'. (Ibid., p. 111) 'But the message of the media case is clear—if you use your power and your platform to warp the minds and destroy the lives of others, you will be punished, and your right to freedom of expression will not protect you'. (Ibid., p. 112)

[99] As enablers, this may even require civil and perhaps criminal liability—when, like for banks, due diligence obligations are neglected—of (social) media companies for spreading of human trafficking conspiracies as well as facilitating the actual crime. Cf. Alegre (2022): 'As technologist turned philosopher James Williams puts it: 'The effect of the global attention economy—i.e., of our digital technologies doing precisely what they are designed to do—is to frustrate and even erode the human will at individual and collective levels, undermining the very assumptions of democracy. These are the distractions of a system that is not on our side' (Alegre 2022, p. 143).

[100] Cf. Alegre (2022, p. 291). Cf. 'The shift from preventing terrorism to preventing 'extremist ideology' pulls the focus from preventing actions and behaviour to truing to prevent thoughts, beliefs and opinions'. (Ibid., p. 292) 'The UN Special Rapporteur on human rights while countering terrorism has pointed out that 'the perception that a government can authorize or control the way that individuals think or what they believe through targeted ideological or theological interventions or what authorized thoughts or beliefs are has no place in societies governed by the rule of law and respect for human dignity'. (Ibid., p. 292). In his speech in September 2016, the then UN High Commissioner for Human Rights, the Jordanian Zeid Ra'ad al Hussein, made the direct link between fear, the appeal of the populist and the manipulative potential of new information delivery methods in the digital age: Populists use half-truths and oversimplification—the two scalpels of the arch propagandist, and here the internet and social media are a perfect rail for them, by reducing thought into the smallest packages: sound-bites; tweets. Paint half a picture in the mind of an anxious individual, exposed as they may be to economic hardship and through the media to the horrors of terrorism. Prop this picture up by some half-truth here and there and allow the natural prejudice of people to fill in the rest. Add drama, emphasizing it's all the fault of a clear-cut group, so the speakers lobbying this verbal artillery, and their followers, can feel somehow blameless. The formula is therefore simple: make people, already nervous, feel terrible, and then emphasize it's all because of a group, lying within, foreign and menacing. Then make your target audience feel good by offering up what is a fantasy to them, but a horrendous injustice to others. Inflame and quench, and repeat many times over, until anxiety has been hardened into hatred'. (Ibid., pp. 295–96). Cf. Miller-Idriss 2022 who similarly points out that one should no longer refer to conspiracy theories but rather conspiracies without the theory, the supposedly explanatory part.

[101] Intended in common law terms; in the Netherlands this would be membership of an organized crime group because, in short, civil law countries in these instances place more emphasis on conduct than intent (the objective element also known as *actus reus* rather than the subjective element also known as *mens rea*), whereas in common law countries generally the subjective element takes priority.

[102] The problem of charnging seditious conspiracy is, in short, that it does not yet capture the notion of 'erod[ing] slowly, in barely visible steps' (Levitsky and Ziblatt 2018, p. 3). Instead, the focus is still very much on the public-facing violent components of how elected governments got overthrown in the earlier days. While ultimately violent behavior did occur during the storming of the Capitol itself, many of the related conduct leading up to it was non-violent, such as the funding, planning and incitement of this seditous conspiracy. Since those acts are oftentimes internationalized now as well, it remains incredibly difficult to decide how to charge all relevant acts or omissions best. This also relates to the problems with a domestic terrorism charge, yet mentioned in relation to the Dutch prosecution of the Mr. de Vries murder case. Translating those problems to the United States, which has a criminal definition of domestic terrorism ex title 18, section 2331 of the United States Code, but no specific domestic terrorism

laws for its enforcement. Per that section, the term 'domestic terrorism' means activities that—(A) involve acts dangerous to human life that are a violation of the criminal laws of the United States or of any State; (B) appear to be intended—(i) to intimidate or coerce a civilian population; (ii) to influence the policy of a government by intimidation or coercion; or (iii) to affect the conduct of a government by mass destruction, assassination, or kidnapping; and (C) occur primarily within the territorial jurisdiction of the United States; and (6) the term 'military force' does not include any person that—(A) has been designated as a—(i) foreign terrorist organization by the Secretary of State under section 219 of the Immigration and Nationality Act (8 U.S.C. 1189); or (ii) specially designated global terrorist (as such term is defined in section 594.310 of title 31, Code of Federal Regulations) by the Secretary of State or the Secretary of the Treasury; or (B) has been determined by the court to not be a 'military force'. (Added Pub. L. 102–572, title X, § 1003(a)(3), Oct. 29, 1992, 106 Stat. 4521; amended Pub. L. 107–56, title VIII, § 802(a), Oct. 26, 2001, 115 Stat. 376; Pub. L. 115–253, § 2(a), Oct. 3, 2018, 132 Stat. 3183). As a related issue, the US may be especially sensitive to a new charge or aggravating circumstance focused on subversion of liberal democracy due to its history of McCarthyism. However, with the right focus on conduct rather than thoughts or words used, it may solve current problems with the newer forms of assaults on liberal democracies that are much more subtle and insidious as a consequence of which a focus on violent coups alone no longer works.

[103]   Anti-Defamation League, 'The Oath Keepers Data Leak: Unmasking Extremism in Public Life', 9 June 2022, available at: https://www.adl.org/resources/report/oath-keepers-data-leak-unmasking-extremism-public-life (accessed on 22 November 2022).

[104]   Because of all the complexities noted above, it is reasonable to conclude that human trafficking is, as a research notion, a wicked problem (e.g., Cels et al. 2017). A wicked problem 'lacks clarity in both its aims and solutions, and is subject to real-world constraints which hinder risk-free attempts to find a solution' (Ibid.). It is 'a social or cultural problem that is difficult or impossible to solve because of its complex and interconnected nature' (Ibid.). Consequently, research will have to pay special attention to the detection of human trafficking, particularly in spheres that are less known as susceptible to this crime such as labor sectors, domestic work, the forced commission of crimes, and organ removal. For this, it helps to use both empirical and normative findings to understand the systemic issues of organized subversive crime. It helps that financial data can be, as for example in epidemiological research such as into COVID-19, used for modelling. This has the added benefit of testing potential real-world effects, without having to test those in practice.

[105]   While research should not be involved in improving criminal investigations, research findings can be used to improve suspicious activity reports regarding human trafficking by banks and other reporting entities under anti-money laundering and anti-terrorism financing legislation. Cf. Berman who states: 'The traditional way for a US attorney's office to become involved in a case is that it is brought to them by the FBI or another federal agency. Law enforcement agents come across possible criminal conduct through data analysis, an informant, a victim, or various other means and are often already investigating when they approach prosecutors. We did not always have to wait for that. The Southern District's resources allowed us to be entrepreneurial, to proactively seek new investigations. I have already mentioned that our public corruption unit would scour complaints filed with the SEC. Similarly, our securities unit did not depend on the SEC referring us cases. We regularly reviewed SARs that banks are required to file and looked for what seemed like egregious violations. Or, through the use of sophisticated data analysis, we spotted patterns of possible white-collar criminal activity that invited more digging on our part. Keep in mind that financial crimes always have victims, and the worst of them ruin people's lives. The reason we pored over those SARs was to make sure we were doing everything possible to identify financial predators. They didn't have to be as big as Bernie Madoff to severely hurt people' (Berman 2022, p. 129). However, Berman thus argues for quite a siloed approach, with SARs being used for the investigation of financial crime rather than all sorts of crime including human trafficking. The question remains: what if the potential SARs from Deutsche Bank regarding Epstein would have been criminally investigated?

[106]   Even in the Mueller investigation into the interference with the US election by Russia no structural financial investigation took place; cf. one of its prosecutors Andrew Weissman: 'In this investigation, that tenacity was as much an asset as a curse: The inability to chase down all financial leads, or to examine all crimes, gnawed at me, and still does' (Weissmann 2020, p. 264).

[107]   It stands to reason that the corrosive effect of dirty and dark money on democratic and rule-of-law institutions is far greater where governments have placed decades-long austerity measures on the public sector including on criminal investigators and the prosecution (so-called 'more with less'-policies; Reich 2021; Tjeenk Willink 2021). In liberal democracies, the oftentimes squeezed public sector will have to investigate the crime of human trafficking and (related) money laundering and corruption.

[108]   Nowadays most corporations are also aware that they may (in)directly cause or facilitate human trafficking or use goods or services made with forced labor (e.g., Shelley 2010). Tech companies, hotels, recruitment agencies, and other legitimate companies appear to play a central role in facilitating human trafficking (e.g., The Texas Supreme Court on 27 June 2021 ruled that Facebook can be held liable for sex traffickers that use its platform to recruit and prey on child victims. Available at https://search.txcourts.gov/SearchMedia.aspx?MediaVersionID=a1c6da82-96ba-48b8-87d3-e5556420b8b4&coa=cossup&DT=OPINION&MediaID=2b5b90ae-2e7e-41ac-9ad6-aceaa1159fc0) (accessed on 22 November 2022). For instance, human trafficking is front and center in debates about how Meta and other Big Tech giants fail to ensure integrity on their platforms (e.g., A leaked SEV (or Site Event) report shows, referenced briefly by The Wall Street Journal's Facebook Files reporting, indicates that Apple threatened to pull Facebook and Instagram from iOS on 23 October 2019. Available at: https://www.theverge.com/22740969/facebook-files-papers-frances-haugen-whistleblower-civic-integrity) (accessed on 22 November 2022). The Facebook papers showed that, after a BBC investigation into domestic servitude, Facebook scrambled to address human trafficking content after

Apple threatened to kick its apps off the iOS App Store. Also, for sex trafficking, there is an intersection of the legitimate economy in the hotel sector and locales such as Airbnb or Vacation Rentals by Owner (VRBO); e.g., The initiative of the Dutch prosecution and police to work with all sex sites to counter forced prostitution; cf. launderettes for the hotel sector in relation to human trafficking for the purpose of labor exploitation. Court of Amsterdam, ECLI:NL:RBAMS:2018:1631.

109    Cf. Eric Williams who states: '(t)hese economic changes are gradual, imperceptible, but they have an irresistible cumulative effect. Men, pursuing their interests, are rarely aware of the ultimate results of their activity. The commercial capitalism of the eighteenth century developed the wealth of Europe by means of slavery and monopoly. But in so doing it helped to create the industrial capitalism of the nineteenth century, which turned round and destroyed the power of commercial capitalism, slavery, and all its works. Without a grasp of these economic changes the history of the period is meaningless'. (Williams [1944] 2000, p. 199). Given the emphasis on the Netherlands, it is important to note that, according to Eric Williams, the Dutch East India Company VOC made five thousand percent in profits, and that the profits from the slave trade were smaller than those made by the British East India company (Williams [1944] 2000, p. 33).

110    Cf. Eric Williams: '(t)he features of the man, his hair, color and dentifrice, his 'subhuman' characteristics so widely pleaded, were only the later rationalizations to justify an economic fact: that the colonies needed labor and resorted to Negro labor because it was cheapest and best'. (Williams [1944] 2000, p. 17). Cf. Jill Lepore citing Ohio Democrat Thomas Morris who in 1839 held the fiercest antislavery speech on the US Senate floor yet: '(b)orrowing from the Jacksonian indictment of the 'money power', he coined the phrase 'slave power'. Morris described the struggle as a battle between democracy and two united aristocracies: 'the aristocracy of the North', operating 'by the power of a corrupt banking system', and 'the aristocracy of the South', which operated 'by the power of the slave system'. (Lepore 2018, p. 224).

111    Cf. Jill Lepore who explains 'slavery seemed like a monster that, each time it was decapitated, grew a new head'. (Lepore 2018, p. 318). On corrosion from within in the United States, MacLean argues 'But take a longer view—follow the story forward to the second decade of the twenty-first century—and a different picture emerges, one that is both a testament to Buchanan's intellectual powers and, at the same time, the utterly chilling story of the ideological origins of the single most powerful and least understood threat to democracy today: the attempt by the billionaire-backed radical right to undo democratic governance'. (MacLean 2018, p. xv).

112    Cf. Williams: '(t)he hope has been expressed that the white servants were spared the lash so liberally bestowed upon their Negro comrades. They had no such good fortune. Since they were bound for a limited period, the planter had less interest in their welfare than in that of the Negroes who were perpetual servants and therefore 'the most useful appurtenances' of a plantation'. (Williams [1944] 2000, p. 14). But there is an important difference insofar as the legal protections for lifelong enslavement are concerned: 'Defoe bluntly stated that the white servant was a slave. He was not. The servant's loss of life was of limited duration, the Negro was slave for life'. (Williams [1944] 2000, p. 15).

113    So, liberal democracies will have to work in concert to tackle the corrosive effect of human trafficking and its related money laundering and corruption within their states and when such corrupting influence comes from outside by autocratic states. The latter should ideally also themselves want to counter this crime. A government of a liberal democracy must be active in countering human trafficking by taking a whole-of-society approach and, where relevant, collaborating internationally including with the private sector (Reich 2021). Corporations should do enough to end their role in facilitating this crime, also through corporate social responsibility (CSR; Reich 2021). Banks specifically should be wary that human trafficking usually involves so much money that perpetrators have to give it a pretend legal origin. Even if earned in cash bills or coins, in our increasingly cash-free societies they or their stooges must usually 'bring' these 'profits' into the legitimate digital financial sector. When reinvesting proceeds in the crime business or using it on their luxurious lifestyle, this intersects with the digital financial market (Tjeenk Willink 2021; Reich 2021). Even cryptocurrencies or underground banking can be understood as modeled on financial flows through this infrastructure. NGOs should support victims of human trafficking, while also seeking to examine related corruption. The general public in a liberal democracy must be wary to not inadvertently contribute to human trafficking and related corrosion of their state, for example, when purchasing goods made or services provided by victims of trafficking in persons. As a final example, academia should study both human trafficking and related corrosion of liberal democracies.

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
