# Peer review of "How Human Trafficking Fuels Erosion of Liberal Democracies—In Fiction and Fact, and from within and without"

_socsci, doi:10.3390/socsci11120560_

Round 1
Reviewer 1 Report
This is an interesting topic to research in the field of human trafficking. Please see the comments that you may find useful.
Some clarifications on page 2, line 61 would be helpful about ‘the anger, frustration, and cynicism about democracy and the rule of law is corroding the moral and social foundation of liberal democracies’. Who exactly has that anger and frustration about democracy and the rule of law? Did you mean criminal groups or ?
It would be helpful to have a clear definition of the context of organized subversive crime in the Netherlands. It seems inadequate analysis of this concept in the case of drug trafficking and then human trafficking on page 8, lines 244-248. On the same page, on lines 255-257 the author says:
‘For example, in a democratic society governed by the rule 255 of law, we only hold to account those who intend(ed) to commit a crime rather than those 256 who were forced to do so. This refers to the non-prosecution and non-punishment principle established in relation to forced drug couriers’.
OK but such an important point just ended there without any more analysis and the fact that there have been some cases in breach of this principle across Europe.
Footnote 6 in line 70 has an inconsistent referencing style. If you are doing name-date (Harvard) style then that should be consistent throughout the text.
the concept of a new crime with the intent to, or effect of, subverting a liberal democracy
there are some repetitions on ‘to consider the eroding effect of human trafficking on liberal democracy around the globe’. This or a similar statement has been repeated several times.
There is a mistake in footnote 17 on page 6 lines 177-178 to say ‘European Union’ is a transnational organisation.
Similarly, the analysis on lines 179-181 doesn’t seem to be convincing and it lacks authority.
There is a discussion on how human traffickers in non-democratic regimes such as Russia place their proceeds of crime in democratic countries. This point needs more analysis and clarification, i.e. why and how do democratic countries allow those traffickers to invest their ‘dirty money’ in democratic countries? Why the rule of law is not effectively operating to counter them?
Some of the analyses mainly relied on the TIP report whereas a diversity of sources would be helpful to be used also. There are also some claims used from generic sources without careful consideration of the context of relevant sources. E.g. see page 9, footnote 30 and again on page 10.
The discussion on conspiracists ‘abuse of the crime of human trafficking’ needs some more in-depth analysis and discussion. Particularly on how this is relevant to the main argument of the ‘effect of human trafficking on liberal democracy’.
In terms of methodology, clarification is needed about the empirical research that the author mentioned.
Finally, in conclusion, was mentioned ‘smart and comprehensive responses’ to tackle human trafficking. However, it is unclear what they meant by ‘smart’? Did the author mean digital technologies? Or just literally speaking they mentioned smart?
The presentation is good but the text would benefit from some headings and sub-headings when the author opens up a new theme.
Author Response
The author thanks the reviewer for their helpful and valuable remarks. Below a point-by-point response is provided.
Reviewer: This is an interesting topic to research in the field of human trafficking. Please see the comments that you may find useful.
Author: Thank you - I did, and have used them in the revised version.
Reviewer: Some clarifications on page 2, line 61 would be helpful about ‘the anger, frustration, and cynicism about democracy and the rule of law is corroding the moral and social foundation of liberal democracies’. Who exactly has that anger and frustration about democracy and the rule of law? Did you mean criminal groups or ?
Author: This sentence has been improved by adding: 'of the general population', thus reading: They worry that the anger, frustration, and cynicism of the general public about democracy and the rule of law is corroding the moral and social foundation of liberal democracies (Ibid.)
Reviewer: It would be helpful to have a clear definition of the context of organized subversive crime in the Netherlands. It seems inadequate analysis of this concept in the case of drug trafficking and then human trafficking on page 8, lines 244-248. On the same page, on lines 255-257 the author says:
‘For example, in a democratic society governed by the rule 255 of law, we only hold to account those who intend(ed) to commit a crime rather than those 256 who were forced to do so. This refers to the non-prosecution and non-punishment principle established in relation to forced drug couriers’.
OK but such an important point just ended there without any more analysis and the fact that there have been some cases in breach of this principle across Europe.
Author: Thanks, this has been improved by including this discussion. "What this novel concept of ‘crime that undermines democratic societies governed by the rule of law’ entails is not fully clear yet." Also,
Reviewer: Footnote 6 in line 70 has an inconsistent referencing style. If you are doing name-date (Harvard) style then that should be consistent throughout the text.
Author: Improved
Reviewer: the concept of a new crime with the intent to, or effect of, subverting a liberal democracy
there are some repetitions on ‘to consider the eroding effect of human trafficking on liberal democracy around the globe’. This or a similar statement has been repeated several times.
Author: improved by deleting repetitions.
Reviewer: There is a mistake in footnote 17 on page 6 lines 177-178 to say ‘European Union’ is a transnational organisation.
Author: Thanks, improved in international organization.
Reviewer: Similarly, the analysis on lines 179-181 doesn’t seem to be convincing and it lacks authority.
Author: Improved by a completely revised section 3. The author realizes that the initial version left the Epstein case too implicit. By using this case as an example, the revised article explains the critiqued lines of "As a final example provided here, the interwovenness with the ‘upperworld’ which will oftentimes be required to recruit victims, sometimes from abroad, ensure the laundering of crime proceeds and corrupt public or private sector representatives to look away or not intervene, for example, can also contribute to the erosion of liberal democracies. This has three further causes."
Reviewer: There is a discussion on how human traffickers in non-democratic regimes such as Russia place their proceeds of crime in democratic countries. This point needs more analysis and clarification, i.e. why and how do democratic countries allow those traffickers to invest their ‘dirty money’ in democratic countries? Why the rule of law is not effectively operating to counter them?
Author: Explained in the new section 3 by explaining inter alia how Russia was placed on the state-sanctioned human trafficking list in the newest TIP report. Additional sources have been added that show how democratic countries allow the state of Russia which is thus an important trafficker and its oligarchs to invest their ‘dirty money’, inter alia, thanks to the work of the aforementioned enablers.
Reviewer: Some of the analyses mainly relied on the TIP report whereas a diversity of sources would be helpful to be used also. There are also some claims used from generic sources without careful consideration of the context of relevant sources. E.g. see page 9, footnote 30 and again on page 10.
Author: Thanks, other reports including from the UNODC have been used. The footnote has been improved: Footnote 30 on page 9 has now been put next to the Big Tech giants to explain the reference to TikTok better (For instance, human trafficking is front and center in debates about how Meta and other Big Tech giants[1] fail to ensure integrity on their platforms.[2] [1] E.g. Kang, Cecilia; Frenkel, Sheera (June 27, 2020). "'PizzaGate' Conspiracy Theory Thrives Anew in the TikTok Era". The New York Times. Archived from the original on June 27, 2020. Available at: https://www.nytimes.com/2020/06/27/technology/pizzagate-justin-bieber-qanon-tiktok.html. [2] E.g. A leaked SEV (or Site Event) report shows, referenced briefly by The Wall Street Journal’s Facebook Files reporting, indicates that Apple threatened to pull Facebook and Instagram from iOS on October 23rd of 2019. Available at: https://www.theverge.com/22740969/facebook-files-papers-frances-haugen-whistleblower-civic-integrity.
I am afraid that there was no footnote on p. 10.
Reviewer:
The discussion on conspiracists ‘abuse of the crime of human trafficking’ needs some more in-depth analysis and discussion. Particularly on how this is relevant to the main argument of the ‘effect of human trafficking on liberal democracy’.
Author:
This discussion has been improved in section 5.
The text is:
The previous sections have demonstrated how human trafficking contributes to the subversion of liberal democracies, in fact and fiction, and from within and outside. For example, actual human trafficking – as exemplified by the Epstein case and confirmed by general facts and global data concerning human trafficking more generally – advances a sense of injustice and a loss of trust by the general public in democracy and the rule of law. Fictional human trafficking in the form of conspiracies equally adds to the assaults on liberal democracies due to the fundamental attacks on the truth, mobilization for violence, and risks for gradually and insidiously introducing totalitarianism. Fictions about human trafficking can even result in real-world consequences pertaining to actual human trafficking, such as the recent voting against anti-human trafficking legislation by a group of 20 US Republicans many of whom are well known for their spreading of conspiracies (the Frederick Douglass Trafficking Victims Prevention and Reauthorization Act was passed on 28 July 2022 in a 401-20 vote).[1] While some of these facts of and fictions about human trafficking erode a liberal democracy from within[2], other attacks rather come from outside. Such external erosion helps to promote autocracy at home by claiming, in short, that liberal democracy is a sham.[3] Worse yet, as much as fact and fiction interact, it is no longer as easy to distinguish between external and internal attacks on liberal democracies because of the involvement of Western enablers (e.g., Guriev and Treisman 2022; Miller-Idriss 2022; Snyder 2018; Stanley 2018; Vogl 2022). This leaves us with many questions as to whether or not those living in liberal democracies want to ensure they contribute to democracy and the rule of law and worldwide equality or rather to autocratic and oligarchical rule. So, how should we respond, both in actuality and through research in academia?
The remainder of this article argues that there are instances in which it would be helpful to charge and conduct research into this new Dutch concept of organized subversive crime for both human trafficking and other such offenses as well as their corrosive effect on liberal democracies, as follows.
[1] Also, the other way around, the Epstein case still translates into many (QAnon) conspiracies including about not only his suicide – also because Mr. Brunel’s suicide similarly happened by way of hanging with bed linen as Mr. Epstein’s – but also how the Clintons – rather than former president Trump or others – are supposedly connected thereto.
[2] Leaders of Western democracies have done little to guard against the backflow of ‘freed capital flows, deregulated business, and opened trade with their former adversaries.’ (Guriev and Treisman 2022, p. 209).
[3] Cf. Sergei Guriev and Daniel Treisman who explain how the new class of autocrats which they call ‘spin dictators’ because, just like conspiracies, they have changed into a more subtle and insidious group of dictators who use spin rather than the less sophisticated propaganda and force from previous centuries, ‘recruit and corrupt Western elites much as they co-opt their own educated class.’ (Guriev and Treisman 2022, p. 147). ‘Today’s spin dictators turn Tito’s double game into an art. They participate in Western institutions in order to extract benefits, exploiting the design flaws and weaknesses of these bodies. They trade with Western countries, while denouncing them. They recruit networks of corrupt partners in the West, simultaneously pursuing concrete goals and eroding Western cohesion. At the same time, they make hypocritical speeches about the West’s hypocrisy.’ (Guriev and Treisman 2022, p. 152).
In terms of methodology, clarification is needed about the empirical research that the author mentioned.
Author:
1.1. Methodology and justification of substantive and geographical focus
The methodology used to answer this central research question is a combination of normative (‘how should it be?’) and, to a limited extent where taken from other predominantly doctrinal sources, empirical research methods (‘what is it?’), while social scientific, historical, and political scientific literature is referred to where relevant. One reason for stressing the importance of needing to obtain empirical findings in future research is the criminological research finding that police interventions that are founded only in theoretical assumptions risk contributing to strengthening criminal networks instead of weakening them (e.g., Luna-Pla and Nicolás-Carlock 2020; Thurner et al 2018; De Domenico et al 2019; Helbing et al 2015; Van Engers et al 2017; Duijn et al 2014). The predominantly doctrinal analysis in this article points out ways to find useful empirical insights in further research. Moreover, the substantive emphasis of this article is on criminal law. Geographically, the focus is on the Netherlands, and in a more limited manner on other European states, the United States, and Russia. Because of the limited scope of this article; other fields of law and countries will have to be reserved for future research.[1] The reasons for this geographical focus are explained further below.
1.2.1. Further justification of geographical focus
This article leads with the Netherlands because scholars and policymakers in that country seek to address a new challenge of crime that erodes its established liberal democracy. The Netherlands did not (yet) face an outright violent attack on its democracy and rule of law such as the storming of the Capitol on January 6th, 2021, in the United States. However, just like the Americans, the Dutch struggle with finding adequate solutions to damage done by crime to those essential tenets of their state, inter alia, because of high-impact murders. While American scholars and policymakers call for dealing with January 6th by introducing a not yet existent domestic terrorism charge, the Dutch are planning on establishing the concept of a crime with the intent to, or effect of, subverting a liberal democracy.[2] This article therefore explores to what extent this solution sought by the Dutch could be helpful in other contexts such as in the United States. This article does so, by focusing on a crime that has not been fully examined in relation to this new Dutch concept either: human trafficking. This examination of the relevance of this new Dutch concept for human trafficking is twofold. First, this article explores whether this novel Dutch concept should or should not encompass human trafficking. Second, the study determines whether a focus on human trafficking can explain the more systemic issues of this new concept in the broader sense. Turning as a final issue to the third country included below, Russia, this country is mentioned because of its export of conspiracies including about human trafficking to liberal democracies (e.g., Snyder 2018; Miller-Idriss 2022). While the emphasis is on Russia because of this spreading of disinformation, this state is also relevant in relation to the crime of human trafficking itself. For example, on 19 July 2022, Russia was put on the state-sponsored human trafficking list, thus indicating an often-overlooked aspect of trafficking in persons not committed by organized criminal groups but by the state itself (TIP report 2022).
[1] This means that this article cannot delve into the entire range of democracies including its most populated, India, and the many others in North America (Canada), western Europe, Central and Eastern Europe, Latin America and the Caribbean, Asia and Australasia, the Middle East and North Africa, and Sub-Saharan Africa. See section 6.
[2] Another possibility could also be to charge the murder – and or in the future, potentially the person or persons who gave the order – with a crime that has been committed with ‘terrorist intent’. The latter is, unlike in the United States, in the Netherlands not reserved for foreign, international terrorism only.
Finally, in conclusion, was mentioned ‘smart and comprehensive responses’ to tackle human trafficking. However, it is unclear what they meant by ‘smart’? Did the author mean digital technologies? Or just literally speaking they mentioned smart?
The research question has been improved into
the following research question: Should some forms of human trafficking be dealt with as a new and unique category of crime or aggravating circumstance that emphasizes its corrosive effect on liberal democracies and, if so, under what conditions would that be useful in law and in research?[1]
[1] Supra note 1.
The presentation is good but the text would benefit from some headings and sub-headings when the author opens up a new theme.
Author:
Thanks, that is, as all of the feedback, very valuable. I have added headings and sub-headings, as can be seen in the new index:
Index
- Introduction. 3
1.1. Context and central research question. 3
1.2. Methodology and justification of substantive and geographical focus 3
1.2.1. Further justification of geographical focus. 4
1.3. Outline of this article. 4
- A new concept: Organized subversive crime. 4
2.1. Introduction. 4
2.2. Defining liberal democracy, democracy and the rule of law.. 5
2.3. Conceptual questions about this new Dutch concept of organized subversive crime. 6
2.4. Relevance of this new Dutch concept of organized subversive crime 7
- Human trafficking: Damage to democracy and the rule of law.. 7
3.1. Introduction. 7
3.2. Human trafficking cases including the Epstein case. 7
3.2.1. The Epstein case: Facts and procedure. 9
3.2.2. The Epstein case: Its corrosive effect 10
3.2.2. The Epstein case: Four factors that explain its corrosive effect 11
3.3. Four explanations for the corrosive effect of human trafficking. 13
3.3.1. Beyond the Epstein case: Understanding the corrosive effect of human trafficking in general 13
3.3.2. Beyond human trafficking: Understanding systemic issues of organized subversive crime. 18
3.4. Conclusion. 18
- Conspiracies about human trafficking: Damage to democracy and the rule of law.. 19
4.1. Introduction. 19
4.2. Conspiracies about human trafficking. 20
4.3. Four explanations for the corrosive effect of conspiracies about human trafficking. 22
4.4. Conclusion. 24
- In fact and fiction, from within and outside: The subversion of democracy and the rule of law.. 24
5.1. Introduction. 24
5.2. The usefulness of the new Dutch concept of organized subversive crime for law.. 25
5.3. The usefulness of the new Dutch concept of organized subversive crime for research. 27
5.4. Conclusion. 28
- Conclusion and recommendations. 28
Literature list. 30
Reviewer 2 Report
This manuscript presents an interesting claim in relation to the extensive contemporary research on the erosion of democracy—it suggests that the state of democracy may be approached by critically scrutinizing the legal and policy framework in place to combat human trafficking. While human trafficking is obviously outlawed from a human rights perspective, the article argues that, because of its impact on the state of democracy, it should be approached by legislators and policymakers innovatively as an act undermining the democratic regime, and outlawed on that basis.
The manuscript is based on a familiarity with a wide range of sources, and, besides a doctrinal approach, has traces of a broader contribution to political theory, particularly in the section which engages with how conspiracists operating in broadly democratic contexts present unwarranted claims of human trafficking to bolster their narrative that the existing regime, in fact, does not allow individuals to make free choices.
While these contributions are certainly worth discussing further and thus the manuscript has value, I believe that it requires considerable work to be publishable, particularly in terms of streamlining and structuring its arguments more effectively, but also with respect to a few methodology- and content- & concept- related elements.
With respect to the format and streamlining, I would strongly recommend to modify the title or the manuscript and rewrite the abstract. The title, while comparatively long, does not tell more than that there will be a study of the relationship between human trafficking and the decline of liberal democracy. If it reflected the key contribution beyond this general topic, while being shorter, it could be more appealing to the readers. Similarly, the abstract provides little information to the readers as to what the manuscript actually studies and how it does so.
Some segments are difficult to follow, see below.
- L. 60-62 speak of worries ‘that the anger, frustration, and cynicism about democracy and the rule of law is corroding the moral and social foundation of liberal democracies’. But this seems to be the opposite relationship to the one studied in the manuscript, tackling how erosion fuels crime (including, possibly, human trafficking), instead of how crime fuels erosion. Is there an ambition to engage with both directions of the relationship? If so, this would need clarification.
- L. 120-122 claim the need for ‘critical reflection on what does and does not constitute organized subversive crime and how to use research findings to determine priorities.’ But such a reflection seems unachievable if the focus is on human trafficking only, and especially within this article which is already on the lengthier end.
- Section 3 was a particularly challenging read. There are many insights, but in no particular order, and digressions (e.g. l. 198-201 on the fall of a Dutch government) make it even less structured. I would suggest to include a figure here, clearly depicting the range of actors who may affect the relationship between human trafficking and democracy (governments, banks, private corporations etc.). The role of banks is particularly interesting but is not clearly spelled out, and I would also recommend using more cautious language as there is no empirical evidence provided for the claim that these actors do, in fact, make a difference. L. 314-315 jump to a normative discussion, which then comes later, so it is not clear why the issue of CSR is mentioned here. L. 332-334 speak about a ‘negative spiral’, but do not provide any evidence – again, a more cautious language (these are possible directions of causation, stemming from theory, and would need to be backed by empirical research) would make the reasoning more convincing. I have major concerns with the claim on l. 339-341, arguing that private exploitation by a trafficker does not affect democracy: this does not just subscribe to a utilitarian position (which is not pursued/explained elsewhere in the manuscript), but, at the very least, is contradictory to the arguments on the loss of trust in democracy by individuals as a source of erosion – what else can be a greater source of distrust than if, in a democracy, an individual is not protected from private exploitation?
- In section 4, the claim that human trafficking is the ‘patient zero’ of disinformation is a very ambitious one, not supported by evidence. I do not think it is sustainable without more evidence.
- In section 5 (l. 643-646), once again, the claim that conspiracies about human trafficking and actual human trafficking should fall into the same category of crimes is not sufficiently justified. As much as conspiracies are dangerous, in democracies they need to be permitted to a degree at least given the right to free speech, and the suffering caused by trafficking seems incomparable to that triggered by (most) conspiracies.
- In l. 649-652 (same section), it is unclear whether the need for a new category of crime is prompted by the incapacity of the American legal system to prosecute those who stormed the Capitol. Are there problems with the current status of the legal framework for effectively addressing these instances? Without identifying such problems, the call for a new type of crime is less convincing.
In terms of the concepts, it is currently not clear how democracy, liberal democracy and the rule of law are related to each other, to what extent the authors use these concepts interchangeably versus intends to provide them with distinct meanings in the context of the article. I think it is important to clarify this relationship because it is a core point of contention in the vast scholarship on erosion of democracy.
I also struggle with the idea of a ‘democracy-undermining crime’ – all crimes are dangerous for democracy, are they not? To the extent that criminality undermines the social fabric, trust and (perceptions of) justice, it seems difficult to claim that a criminal act such as human trafficking is dangerous for democracy, while another act, let us say, a robbery, is not. Of course, there is extensive scholarship on ‘militant democracy’, which does rest on the fact that some acts that may (though do not necessarily have to be) criminalized are particularly dangerous for democracy—notably ‘hate speech’, but also manipulation of elections and (depending on the specific scholar) corruption. If the authors intend to pursue this as a core claim (which seems to be the case given the extensive borrowing from the Dutch literature on ‘organized subversive crime’), I would recommend them to embed this claim in the discussion on militant democracy, which is better known as the Dutch sources (largely not accessible in English). In this regard, the claim on the plans to introduce ‘the concept of a crime with the intent to, or effect of, subverting a liberal democracy’ in the Netherlands (l. 142-144) is puzzling. I am not familiar with the Dutch debate (and most readers of the journal cannot be expected to be), but it seems that such crimes are already in place in ‘militant democracies’ (e.g. prosecution of political party leaders for advocating regime change to an autocracy in their speeches). Some scholars whose work might be particularly relevant are B. Rijpkema (who also looks at the Dutch context of militant democracy), Jan-Werner Müller or András Sajó. There, then, one also needs to think of potential dangers of abuse of categories of crimes (relevant especially in relation to the authors’ proposal for a new crime, e.g. l. 658-666).
The Conclusion would benefit from addressing more clearly why a new type of crime would automatically (as that seems to be the argument) help enforcement, i.e. whether the focus should not instead be on implementation of the existing legal framework and in particular on judicial capacities/resources in this regard. The logic for why a new crime would be more conducive to democracy than, for example, adding an aggravating circumstance to human trafficking on the basis of its potential to undermine democracy is not qualified (and, once again, all crimes might be seen as having this potential). Generally, identifying the limitations of the study and some prospects for further research would help improve the conclusion in my view, to leave the reader with more takeaways.
The methodology would need improvement, particularly in terms of using more established and clear-cut presentation of the methods used in the manuscript. While this is open to interpretation, in my view the manuscript is essentially a doctrinal study with some traces of political theory and conceptual analysis. This is perfectly fine; however, the authors claim to be using ‘empirical’ methods in addition to normative, attributing the question ‘what it is’ to the former (p. 2, l. 27). This is not sustainable in my view, as empirical methods also address how and why questions, and there is no trace of empirical methods used in the manuscript (the closest here would be a case study of the Netherlands, but if that is the authors’ intention, then a restructuring of the manuscript would be necessary to have a section explicitly engaging with the Dutch case). In fact, the conclusion, which is almost completely normative, further underscores that empirical methods have not been used.
Related to the above, there is no justification given for the extensive references to the Dutch context and resources in the manuscript. If the intensity of the use of these sources is to be retained, alongside the ambition to make a general point, rather than one specific to the Dutch legal system, I think some justification is needed for the extensive use of these sources. More broadly, the choice of focus on the jurisdictions (p. 2, l. 34-36) is also not justified: it is clear that only some can be considered for space reasons, but why these ones in particular? Is it because there is a more vivid discourse on human trafficking in them (which would need to be demonstrated), or because they are established democracies (in which case Russia would obviously not work for the sample) or something else?
Below are a few more specific comments labelled according to the lines in the document.
- There are quite a few quotes by ‘Tjeenk Willink 2021’, but this source is missing from the reference list. A review of the compatibility of the sources cited and the source list at the end of the manuscript seems necessary.
- ‘20 Republicans of the Frederick Douglass Trafficking Victims Prevention and Reauthorization Act which was passed on 28 July 2022 in a 401-20 vote.’ This segment appears twice in the text (n. 32, first paragraph of sec. 5).
- L. 711-713 – unlike in the introduction, here the reader is left with the normative question only. It seems necessary to unify the actual question(s) the manuscript tackles between the introduction and the conclusion. In particular, the second question in the introduction is too broad (l. 23-25) and cannot be answered in an article, so focusing just on one question (the need for a separate crime category) might be more effective.
- L. 60: unclear how oligarchy enters the discussion here.
- L. 30 & 31, ‘police or other rule-of-law-based interventions’: police interventions might be at odds with the rule of law, equating the two does not work without further justification.
- N. 1 (Pinker): why should there be a source by this author from after 2018? There are other quoted sources which do not come from after 2018.
In sum, despite my reservations and opinion that the current form of the manuscript is not publishable, I encourage the authors to proceed with the revisions and develop these interesting ideas further, as there is a potential to connect them to several established threads in the literature on democracy protection.
Author Response
The author thanks the reviewer for the helpful and much appreciated review.
Please find below a point-by-point response.
Reviewer: This manuscript presents an interesting claim in relation to the extensive contemporary research on the erosion of democracy—it suggests that the state of democracy may be approached by critically scrutinizing the legal and policy framework in place to combat human trafficking. While human trafficking is obviously outlawed from a human rights perspective, the article argues that, because of its impact on the state of democracy, it should be approached by legislators and policymakers innovatively as an act undermining the democratic regime, and outlawed on that basis.
The manuscript is based on a familiarity with a wide range of sources, and, besides a doctrinal approach, has traces of a broader contribution to political theory, particularly in the section which engages with how conspiracists operating in broadly democratic contexts present unwarranted claims of human trafficking to bolster their narrative that the existing regime, in fact, does not allow individuals to make free choices.
While these contributions are certainly worth discussing further and thus the manuscript has value, I believe that it requires considerable work to be publishable, particularly in terms of streamlining and structuring its arguments more effectively, but also with respect to a few methodology- and content- & concept- related elements.
Author: Many thanks for these comments, and I have indeed streamlined and structured the arguments more effectively, and improved the few methodology- and content- & concept- related element.
The research question and explanations have been improved into:
1.1. Context and central research question
This article addresses human trafficking in the context of the reported decline of liberal democracies worldwide (e.g., Diamond 2019; Economist Intelligence Unit 2021; Mounk 2019; Levitsky and Ziblatt 2018; Guriev and Treisman 2022).[1] There is no better place than this Special Issue to examine if, and if so how, some forms of trafficking in persons advance the erosion of core tenets of liberal democracies including democracy and the rule of law. Both human trafficking and this potential corrosive effect on liberal democracies are complex and difficult to detect because of their mostly hidden nature (e.g., Shelley 2012; Levitsky and Ziblatt 2018; Snyder 2018; Coster van Voorhout 2020). Therefore, to study both issues comprehensively, this article takes all relevant angles: human trafficking in fact and in fiction, and corrosion from within and outside liberal democracies. From all four angles, this article seeks to answer the following research question: Should some forms of human trafficking be dealt with as a new and unique category of crime or aggravating circumstance that emphasizes its corrosive effect on liberal democracies and, if so, under what conditions would that be useful in law and in research?[2]
1.2. Methodology and justification of substantive and geographical focus
The methodology used to answer this central research question is a combination of normative (‘how should it be?’) and, to a limited extent where taken from other predominantly doctrinal sources, empirical research methods (‘what is it?’), while social scientific, historical, and political scientific literature is referred to where relevant. One reason for stressing the importance of needing to obtain empirical findings in future research is the criminological research finding that police interventions that are founded only in theoretical assumptions risk contributing to strengthening criminal networks instead of weakening them (e.g., Luna-Pla and Nicolás-Carlock 2020; Thurner et al 2018; De Domenico et al 2019; Helbing et al 2015; Van Engers et al 2017; Duijn et al 2014). The predominantly doctrinal analysis in this article points out ways to find useful empirical insights in further research. Moreover, the substantive emphasis of this article is on criminal law. Geographically, the focus is on the Netherlands, and in a more limited manner on other European states, the United States, and Russia. Because of the limited scope of this article; other fields of law and countries will have to be reserved for future research.[3] The reasons for this geographical focus are explained further below.
1.2.1. Further justification of geographical focus
This article leads with the Netherlands because scholars and policymakers in that country seek to address a new challenge of crime that erodes its established liberal democracy. The Netherlands did not (yet) face an outright violent attack on its democracy and rule of law such as the storming of the Capitol on January 6th, 2021, in the United States. However, just like the Americans, the Dutch struggle with finding adequate solutions to damage done by crime to those essential tenets of their state, inter alia, because of high-impact murders. While American scholars and policymakers call for dealing with January 6th by introducing a not yet existent domestic terrorism charge, the Dutch are planning on establishing the concept of a crime with the intent to, or effect of, subverting a liberal democracy.[4] This article therefore explores to what extent this solution sought by the Dutch could be helpful in other contexts such as in the United States. This article does so, by focusing on a crime that has not been fully examined in relation to this new Dutch concept either: human trafficking. This examination of the relevance of this new Dutch concept for human trafficking is twofold. First, this article explores whether this novel Dutch concept should or should not encompass human trafficking. Second, the study determines whether a focus on human trafficking can explain the more systemic issues of this new concept in the broader sense. Turning as a final issue to the third country included below, Russia, this country is mentioned because of its export of conspiracies including about human trafficking to liberal democracies (e.g., Snyder 2018; Miller-Idriss 2022). While the emphasis is on Russia because of this spreading of disinformation, this state is also relevant in relation to the crime of human trafficking itself. For example, on 19 July 2022, Russia was put on the state-sponsored human trafficking list, thus indicating an often-overlooked aspect of trafficking in persons not committed by organized criminal groups but by the state itself (TIP report 2022).
1.3. Outline of this article
This article is divided into six parts. First, a description follows of the aforementioned new Dutch concept of crimes that undermine democratic societies governed by the rule of law that scholars and policy-makers are currently still developing (section 2). Thereafter, a factual examination as to whether human trafficking can qualify as a crime that undermines democratic societies governed by the rule of law follows (section 3). In the subsequent part, this article turns from fact to fiction, by focusing on how conspiracists abuse the phenomenon of human trafficking (section 4). Thereupon, the penultimate section deals with the usefulness of this new Dutch concept including for human trafficking (section 5). Finally, a conclusion is drawn and recommendations for further research are made (section 6).
[1] For an overview of data used to undergird this thesis and the conclusion itself, see the report by the Economist Intelligence Unit from 2021, available at: https://www.economist.com/graphic-detail/2021/02/02/global-democracy-has-a-very-bad-year. Cf. under the term ‘democratic deconsolidation’, e.g. Yascha Mounk and Larry Diamond (Yascha Mounk, People vs. Democracy – Why Our Freedom Is in Danger and How to Save It, Cambridge: Harvard University Press, 2019, and Larry Diamond, Ill Winds: Saving Democracy from Russian Rage, Chinese Ambition, and American Complacency, New York: Penguin Press, 2019. Some argue that these concerns about the decline of democracies worldwide are somewhat exaggerated but even they express concern about the degree to which what they call ‘spin dictators’ damage liberal democracies, oftentimes through erosion from outside and within (See Sergei Guriev and Daniel Treisman, Spin Dictators – The changing face of tyranny in the 21st Century, Princeton: Princeton University Press, 2022). Similarly, on the trend of the decline of liberal democracies itself, see for example Levitsky and Ziblatt, How Democracies Die: What History Reveals About Our Future, New York: Penguin Random House, 2018. Rather than decline through an outright coup, they argue that currently democracies rather ‘erode slowly, in barely visible steps’ (Levitsky and Ziblatt 2018, p. 3). More positive is a scholar like Pinker (Steven Pinker, Enlightenment now: The case for reason, science, humanism, and progress, New York: Viking, 2018). However, the author did not find any writings on this topic from Pinker after 2018, who may also have changed his opinion at least somewhat after January 6th, 2021 . Pinker’s language expertise was used by the Epstein defense (Brown 2021, p. 156).
[2] Supra note 1.
[3] This means that this article cannot delve into the entire range of democracies including its most populated, India, and the many others in North America (Canada), western Europe, Central and Eastern Europe, Latin America and the Caribbean, Asia and Australasia, the Middle East and North Africa, and Sub-Saharan Africa. See section 6.
[4] Another possibility could also be to charge the murder – and or in the future, potentially the person or persons who gave the order – with a crime that has been committed with ‘terrorist intent’. The latter is, unlike in the United States, in the Netherlands not reserved for foreign, international terrorism only.
Reviewer: With respect to the format and streamlining, I would strongly recommend to modify the title or the manuscript and rewrite the abstract. The title, while comparatively long, does not tell more than that there will be a study of the relationship between human trafficking and the decline of liberal democracy. If it reflected the key contribution beyond this general topic, while being shorter, it could be more appealing to the readers. Similarly, the abstract provides little information to the readers as to what the manuscript actually studies and how it does so.
Author:
The title and abstract have been improved into:
How human trafficking fuels erosion of liberal democracies
In fact and fiction, and from within and outside
Abstract
On the same day that the human trafficker Ms. Ghislaine Maxwell was sentenced to 20 years’ imprisonment, many people closely watched the sixth hearing of the House Select Committee on the attack of the Capitol on January 6th (28 June 2022). What, if anything, do these ostensibly varied crimes have in common? Seeking to answer this fundamental question, this article explores the usually under-researched connection between trafficking in persons and the documented decline of liberal democracies worldwide. Globally, democratic societies governed by the rule of law appear to be under assault, and therefore this article explores relevant examples of how human trafficking contributes to the erosion of liberal democracy, in fact and fiction, and from within and outside. In other words, this article takes us from profits to ‘Pizzagate’.
The abstract continues to be meant to not give away too much, so as to entice the reader to read the entire article.
Some segments are difficult to follow, see below.
- L. 60-62 speak of worries ‘that the anger, frustration, and cynicism about democracy and the rule of law is corroding the moral and social foundation of liberal democracies’. But this seems to be the opposite relationship to the one studied in the manuscript, tackling how erosion fuels crime (including, possibly, human trafficking), instead of how crime fuels erosion. Is there an ambition to engage with both directions of the relationship? If so, this would need clarification.
Author: Thanks, this has been improved by clarifying:
They worry that the anger, frustration, and cynicism of the general public about democracy and the rule of law is corroding the moral and social foundation of liberal democracies (Ibid.)
In the context:
Dutch scholars and policy-makers are developing a novel concept to fundamentally understand a potentially new phenomenon of organized crime that oftentimes includes but is not limited to human trafficking (Eski et al 2021; Huisman and Kleemans 2017; Tops and Tromp 2017). For this, they have converted the verb ‘to undermine’ into a noun: ‘crime that undermines liberal democracies’.[1] Such crime is arguably deserving of special attention, because it not only harms victims and, by extension, their families and communities, but also subverts liberal democracies or at least has the potential to do so. Alternatively described as ‘the interwovenness of upper- and underworld’ or ‘organized subversive crime’, it contributes to, or even underlies, an overall sense of injustice and loss of trust from citizens in the state (Tjeenk Willink 2021). Some even deem this potentially newer type of criminality a crisis, arguing that it is on a par with ‘climate change, migration questions, and increased societal divisions’ (Ibid.; cf. Reich about the US, 2021; cf. on US oligarchy[2] Bernstein 2020; Giridharadas 2019; Mayer 2017; Snyder 2018). They worry that the anger, frustration, and cynicism of the general public about democracy and the rule of law is corroding the moral and social foundation of liberal democracies (Ibid.) The analyses of this current purported crisis in liberal democracies around the world have pinpointed everything from the influence of polarization, economic inequality, social media to white male rage. But as valid as these points may be, few of them fully explore the deep historical roots of how liberal democracies get eroded by what types of crime and the extent to which human trafficking plays a role therein. Before exploring these roots in the substantive sections of this article (sections 3 to 5), it is important to first address the most relevant conceptual notions of liberal democracy, democracy, and the rule of law.[3]
[1] In Dutch: derived from the noun of ‘to undermine’: ondermijnen, the noun ondermijning, which is a shorthand for rechtsstaat-ondermijnende criminaliteit). This is not to say that there is not some debate between Dutch scholars as to what defines this as of yet, as is the case for organized crime itself, of undefined category of crimes. See below in this section (section 2).
[2] In keeping with Snyder’s explanation, this article follows the meaning given to ‘oligarchy’ as used by Aristotle, meaning rule by the wealthy few; the word in this sense was revived in the Russian language in the 1990s, and then, with good reason, in English in the 2010s. (Snyder 2018, p. 11).
[3] Cf. Sergei Guriev and Daniel Treisman 2022, p. 7.
- L. 120-122 claim the need for ‘critical reflection on what does and does not constitute organized subversive crime and how to use research findings to determine priorities.’ But such a reflection seems unachievable if the focus is on human trafficking only, and especially within this article which is already on the lengthier end.
Author: agreed, and therefore the improvements are as follows:
This dearth of conceptual clarity requires critical reflection on what does and does not constitute organized subversive crime and how to use research findings to determine priorities more generally. More specifically, this article explores one aspect of its conceptual scope by focusing on its potential relevance for human trafficking insofar as its possible corrosive effect on liberal democracies is concerned. While a more familiar research theme may very well be how crimes like ‘hate speech’ or corruption erode democracy and the rule of law, this article explores if, and if so to what extent, human trafficking should also be understood to have this corrosive effect.[1] This raises the question why the Netherlands is developing this new concept of organized subversive crime in the first place.
[1] The author thanks the two anonymous reviewers for their remarks that ‘hate speech’ and corruption crimes are more familiar to the audience of this journal.
- Section 3 was a particularly challenging read. There are many insights, but in no particular order, and digressions (e.g. l. 198-201 on the fall of a Dutch government) make it even less structured. I would suggest to include a figure here, clearly depicting the range of actors who may affect the relationship between human trafficking and democracy (governments, banks, private corporations etc.). The role of banks is particularly interesting but is not clearly spelled out, and I would also recommend using more cautious language as there is no empirical evidence provided for the claim that these actors do, in fact, make a difference. L. 314-315 jump to a normative discussion, which then comes later, so it is not clear why the issue of CSR is mentioned here. L. 332-334 speak about a ‘negative spiral’, but do not provide any evidence – again, a more cautious language (these are possible directions of causation, stemming from theory, and would need to be backed by empirical research) would make the reasoning more convincing. I have major concerns with the claim on l. 339-341, arguing that private exploitation by a trafficker does not affect democracy: this does not just subscribe to a utilitarian position (which is not pursued/explained elsewhere in the manuscript), but, at the very least, is contradictory to the arguments on the loss of trust in democracy by individuals as a source of erosion – what else can be a greater source of distrust than if, in a democracy, an individual is not protected from private exploitation?
Agreed, and I therefore included an entirely new section 3. I realize that I left the Epstein case too implicit, and have therefore added a better explanation, as follows:
3. Human trafficking: Damage to democracy and the rule of law
3.1. Introduction
Today’s perhaps best-known human trafficking case around the world, the Epstein case, helps demonstrate damage done to – in addition to victims, their families and their communities – democracy and the rule of law. Mr. Epstein’s conduct up until his suicide in pre-trial detention prompted a group of four bipartisan US Senators, for example, to state that ‘these events have ignited a crisis of public trust in the Department [of Justice] and exacerbated the erosion of trust that the American people have in our institutions of republican self-government more broadly.’[1] The fact that a human trafficking case can cause a sense of injustice and a loss of public trust in liberal democracy raises a number of questions. Can human traffickers (in)directly corrupt public and private institutions and representatives and, if so, how do they do so? Does human trafficking contribute to the erosion of liberal democracy, for example by compromising the integrity of the justice and financial sector? Under what circumstances does human trafficking contribute to a sense of injustice and loss of public trust in liberal democracy and, if so, how?
3.2. Human trafficking cases including the Epstein case
Before describing in more detail at first the Epstein case and then facts and global data about human trafficking more generally, it is important to provide context. Most significantly, oftentimes it is wrongly assumed that human trafficking only affects a few states around the globe and, given that the Epstein case is a sex trafficking case, it is important to dispel at the outset the notion that the latter is the only form of human trafficking.
First, human trafficking occurs in every region of the world. For example, about 50,000 human trafficking victims were detected and reported by 148 of 155 reporting countries in 2019 (UNODC 2020). States can be the origin, transit or destination country for victims, or even a combination of all (Ibid.).
Second, human trafficking covers an enormous spectrum, ranging from the sex sphere to business sectors like agriculture, restaurants, transportation, forced commission of –drug[2] or other– crimes or begging and organ removal (Shelley 2013; Coster van Voorhout 2009 and 2020). The commodities produced by trafficked persons range from clothing to electronics and food, and the profits made by their sale are often not easily distinguished from licit flows of money.
Last, human trafficking is a highly complex crime[3]; a fundamental human rights violation[4]; and an offense that cannot always easily be distinguished from other crimes like migrant smuggling – if a migrant first consents to illegal entry but must pay off the travel debt through forced (sex) labor, for example (cf. Aronowitz 2009; Triantafyllidou and Mouroukis 2012, Cho 2015; Campana and Varese 2016; UNODC 2020).
Before turning to the Epstein case, it must be emphasized that it is difficult to know all aspects of human trafficking because it is a predominantly hidden crime. The same accounts for (human-trafficking related) corruption. This limits our understanding about both offenses. This refers to what in criminological literature is known as the dark figure (Biderman and Reiss 1967; Skogan 1977). We may never know how many human trafficking and/or related corruption crimes there really are, and how many offenders, facilitators and victims are involved.[5]
Another important issue as context for the below description of both the Epstein case and facts and global data about human trafficking more generally is that trafficking in persons is usually a low-risk high-profit crime, because few countries in the world have prioritized its prosecution or confiscation of assets (Shelley 2012, p. 242; Coster van Voorhout 2020). Increasingly, there is recognition that human trafficking can only be addressed and hopefully prevented if the whole of the global society works in concert. Insofar as responsibility of companies conducting corporate social responsibility to counteract human trafficking in supply chains is concerned, this crime is nowadays often known as modern slavery (Coster van Voorhout 2020).
On a final note, the importance of countering human trafficking in concert by states, companies, NGOs, the general public, and others has grown more urgent, because, in the last five years, the number of estimated victims has gone up with 25%, to 50 million people worldwide (ILO, IOM and Walk Free 2022). Globally, since its first year of registration of 2015, there is a 45% drop in prosecutions of human trafficking (Ibid.; e.g., TIP report 2022, p. 62).[6] Also, there is a 24% decrease in the victim identification rate worldwide since the year of the highest mark, 2019 (Ibid.). Moreover, recently, due to Covid-19, the share of children among detected human trafficking victims around the globe has tripled (e.g., UNODC 2021, p. 3). As a final trend that indicates this urgency of addressing human trafficking, people are increasingly vulnerable to being trafficked within and across borders. In future, more persons are exposed to trafficking, since globally the number of people forced to flee their homes is the highest since World War II (e.g., UNHCR 2022, p. 3). All expectations are that climate change, conflict over new resources like water and essential raw elements, and increased social and economic inequality will create further (internal) migration. This urgency about human trafficking itself also shows the significance of exploring its (potentially increasing) corrosive effect on (possibly more) liberal democracies, which is here done first by turning to the Epstein case.
3.2.1. The Epstein case: Facts and procedure
The Epstein case started with a criminal investigation after a 2005 report by a mother of a 14-year-old girl who alleged that the then 52-year-old Mr. Epstein had sexually abused her daughter.[7] That investigation by initially local police and thereafter the FBI revealed that, between approximately 1999 and 2007, Mr. Epstein and multiple co-conspirators assembled a network of at least thirty-four underage girls whom he sexually abused at his mansion in Palm Beach, Florida. Following the FBI’s investigation, by May 2007, the prosecution office completed an 82-page prosecution memo and a 53-page draft indictment against Mr. Epstein, charging him with federal crimes related to the sex trafficking of minor victims.[8] The federal prosecutor, Mr. Acosta, set a tentative date to indict Mr. Epstein of May 15, 2007. However, on September 24, 2007, Mr. Acosta concluded a non-prosecution agreement[9] with Mr. Epstein, without notifying the named thirty-four victims moreover.[10] In return for federal immunity, Mr. Epstein agreed to plead guilty to two low-level state solicitation of prostitution charges and serve eighteen months in the county jail.[11] Mr. Epstein received work release from that jail and spent considerable time not imprisoned but on house arrest. On 8 July 2019, at least fourteen years after the initial criminal investigation, the prosecutor in New York’s southern district, Mr. Berman, indicted Mr. Epstein for sex trafficking of minors. In addition to Palm Beach, Florida, the sex trafficking operation was suspected of also having reached into at least New York, New Mexico, the US Virgin Island of St. James and London, the United Kingdom, while also being connected most likely to another co-conspirator in France.[12] Mr. Acosta, later on 12 July 2019, resigned from his position of Secretary of Labor in the Trump Administration. The US labor department oversees the large government agency dedicated to anti-human trafficking and anti-child labor efforts. On 10 August 2019, due to Mr. Epstein’s suicide in pre-trial detention, the case against Mr. Epstein was discontinued by the aforementioned New York prosecutor Berman.[13] Other cases including a defamation case against one of Mr. Epstein’s lawyers, Mr. Dershowitz, by one of Epstein’s named victims, Ms. Virginia Giuffre, are still ongoing.[14]
3.2.2. The Epstein case: Its corrosive effect
The corrosive effect of the Epstein case in the United States, in addition to the yet cited quotation by the group of US Senators, has been best described by the investigative journalist Ms. Julie K. Brown who reported on the Epstein case relatively early on in 2018 and who aptly refers to perversion of justice (Brown 2021).[15] This corrosive effect was also felt beyond America, despite the fact that most of the sex trafficking occurred on that territory.[16] First, in the United Kingdom, after no apparent criminal investigation by the British, Mr. Epstein’s known British co-conspirator, Ms. Ghislaine Maxwell, was convicted in the US for conspiracy to commit and actual commission of sexual exploitation including in her residence in London, the United Kingdom.[17] As another example, Prince Andrew from Britain laid down all his functions upon a BBC interview that sought to address allegations by the aforementioned Ms. Virginia Giuffre at, inter alia, that residence of Ms. Maxwell. Second, in France, seemingly without a previous criminal investigation by the French, Mr. Jean-Luc Brunel, who reportedly founded his modeling agency with Mr. Epstein’s money, was arrested for using this agency to recruit victims that also ended up in Epstein’s scheme.[18] The French have also had to discontinue this criminal investigation because, like Mr. Epstein, Mr. Brunel was found to have committed suicide in a similar fashion as Mr. Epstein. In addition to thus affecting at least three established liberal democracies, the Epstein case also negatively impacted Mr. Epstein’s bank. Deutsche Bank entered into a $150 million settlement because of its failure for years to properly monitor account activity conducted on behalf of the life-long registered sex offender Mr. Epstein.[19] The settlement record shows that Mr. Epstein made suspicious payments to his (alleged) co-conspirators, wired money to Russian models and made a connected cash withdrawal of $100,000 for ‘tips and household expenses’.
3.2.2. The Epstein case: Four factors that explain its corrosive effect
In recognition of the fact that erosion of liberal democracies happens in ‘barely visible steps’,[20] as Levitsky and Ziblatt argue, as was explained above, the cumulative effect of such steps in the Epstein case was aptly summarized by Brown as: ‘Epstein got away with his crimes because nearly every element of society allowed him to get away with them. Professional, legal, and moral ethics were set aside for a broken system of values that places corporate profits, personal wealth and political connections, and celebrity above some of the most sacred tenets of our faiths, our teachings, and our democracy’ (Levitsky and Ziblatt 2018; Brown 2021, p. xiv).[21] Seeking to detect as much as possible the separate steps in the Epstein case[22], it shows that there were at least four corroding factors at play.
First, the Epstein case damaged the integrity of the rule of law and to impartial justice, as can best be detected by following the ‘wealth and influence’ which he used stealthily rather than openly[23] in order to ‘from the beginning of the case [to] marshal[…] the weaknesses of the criminal justice system to his benefit’ (Brown 2021, p. 77).[24] As far as the origin of Mr. Epstein’s wealth is concerned, it is unclear how much money he made through sex trafficking itself. However, it is likely that Mr. Epstein had set up surveillance of all his properties that may have been used to blackmail those who sexually exploited the underage victims trafficked by him and his co-conspirators.[25] What is known about Mr. Epstein’s (possible other) source of wealth is that he advised affluent others about tax evasion and avoided taxes himself.[26] While tax avoidance is usually lawful, it can still be ‘awful’ in the sense that criminologist Passas explains it (‘lawful but awful’; Passas 2005). This latter aspect will be revisited below in the next sub-section because of the corrosive effect of extracting money that could otherwise be spent publicly including on strengthening democracy and the rule of law.[27] But returning to Mr. Epstein for now, his career may not have consisted of only legal tax avoidance. Others connected to Mr. Epstein like Mr. Hoffenberg, who testified to the grand jury that Mr. Epstein was the mastermind behind their scheme, were held accountable for crimes of mail fraud, tax evasion, and obstruction of justice in 1995. Mr. Epstein was not convicted for such crimes. Whatever the (il)legal source of Mr. Epstein’s wealth and influence, it allowed him to pay for the legal assistance of most victims and his co-conspirators who mostly were from poorer socioeconomic backgrounds.[28] The effect of the latter can also be seen in a court ruling about how Mr. Epstein gained immunity for sex trafficking through the aforementioned non-prosecution agreement (NPA): ‘Worse, it appears that prosecutors worked hand-in-hand with Epstein’s lawyers—or at the very least acceded to their requests—to keep the NPA’s existence and terms hidden from victims. And to be clear, the government’s efforts appear to have graduated from passive nondisclosure to (or at least close to) active misrepresentation.’[29] The latter means that the thirty-four named victims remained in the dark about the immunity provided to Mr. Epstein in exchange for still unknown activities in support of law enforcement, if any.[30] Mr. Epstein has thus been able to use his wealth and influence to not be fully prosecuted in 2008[31], only to have reportedly continued and widened his sex trafficking operation for at least a decade.[32]
Second, in terms of potential damage done to the integrity of democracy, some of Mr. Epstein’s wealth and influence later appears to have contributed to ensuring his local, state and federal political connections[33] from both political parties in the United States[34] and other powerful contacts in science, the legal community and academia, for example.[35] It is difficult to assess the degree to which money spent by Mr. Epstein on politics has played a role in the Epstein case because of lacking transparency on the role of money in US politics more generally (e.g., Mayer 2017; Bernstein 2020). Correspondingly, it is unknown to what extent in the Epstein case prosecutors were insulated from political interference or not. However, this case does indicate how the dividing line between politics and the criminal justice system cannot always easily be drawn in more general terms. For example, as mentioned above, the former federal prosecutor, Mr. Acosta, later became Secretary of Labor in the Trump Administration and resigned when his role in yielding his prosecutorial discretion to not at least fully prosecute Epstein became national news.
Third, in terms of social and economic inequality, Mr. Epstein was able to at a minimum exploit the poorer socioeconomic conditions of most of the thirty-four female victims and his known co-conspirators. So even very local social and economic inequality – here between Palm Beach County and Palm Beach “Millionaires Row”, Florida – can allow a trafficker to exploit victims who moreover did not obtain complete justice for the harm done to them.
Finally, in terms of lacking international coordination between liberal democracies and private actors, as said above, there is no evidence that Ms. Maxwell from the United Kingdom and most likely Mr. Brunel from France were investigated in those jurisdictions nor that these states cooperated with the US to investigate the Epstein case. Also, evidently a global financial institution did not do its due diligence under its anti-money laundering obligations for a high-risk client and did not support one or more state investigations either. It therefore stands to reason that the US Senators were correct about this effect which moreover went beyond the US and the public sector.
3.3. Four explanations for the corrosive effect of human trafficking
3.3.1. Beyond the Epstein case: Understanding the corrosive effect of human trafficking in general
First, damage done by human traffickers to the integrity of the rule of law and impartial justice can occur not only at the level of compromising prosecutorial discretion and the Department of Justice, as in the Epstein case which unlike many other trafficking in persons cases did come to light at least somewhat, but also at the level of police[36] and judicial independence[37] (e.g., Diamond 2019; Levitsky and Ziblatt 2018; Guriev and Treisman 2022).[38] As seen in the Epstein case, human traffickers can use their money and influence not only to sustain their criminal market, but also go beyond that by eroding the institutions and representatives that are essential for the detection and subsequent prosecution and – if not pleaded out beforehand through a non-prosecution agreement – adjudication of human trafficking cases. Additionally, human traffickers are known to have used their funds and influence to corrupt the integrity of the democratic system by bribing politicians, for example, or for private sector corruption by compromising the integrity of companies (UNODC 2011). For the funds needed for both public and private sector corruption, most human traffickers will, unlike Mr. Epstein, however, not have to ‘earn’ their money through any other activities than trafficking in persons itself. After all, human trafficking is, by estimation, a $150 billion ‘industry’ (ILO 2014).[39] To make this significant sum of money more tangible, this is more than the combined 2021 profits of four large companies: Meta[40], Disney, Starbucks, and Nike.[41] This amount, which is already significant, may even be an underestimation. Increasingly, international organizations like the European Union[42] and human rights courts like the European Court of Human Rights[43] recognize a new form of human trafficking in forced couriering of drugs. Potentially, some of the money now ascribed to the estimated proceeds from drug trafficking, $300 billion globally, may therefore have to be added to the human trafficking profits.[44] This may bring the criminal money derived from human trafficking closer to this only other crime that is estimated to be more profitable worldwide: drug trafficking. Some argue that human trafficking profits are even on the rise – a possibility that appears to be confirmed by the increasing number of victims (cf. Shelley 2021; cf. ILO, IOM and Walk Free 2022). This means that human traffickers can use their criminal money, ‘dirty money’ or, where spent anonymously, ‘dark money’ in a way that corrodes liberal democracies – an effect that US Supreme Court Justice Breyer has already condemned as: ‘Where enough money calls the tune, the general public will not be heard’.[45]
Before turning to the further corrupting influence on liberal democracies of both dirty and dark money made through trafficking in persons, a conceptual problem must be stressed. Despite the already detected examples, like in the Epstein case, of interwovenness of human traffickers with the ‘upperworld’ including recruiters, launderers of crime proceeds and corrupt public or private sector representatives who look away or decide not to intervene, the notion of criminality is itself not always clear-cut (Reich 2021; Korf et al 2018; Passas 2005; Boister 2018). Some government actions, business practices or individual’s activities can be ‘lawful but awful’, as was referenced above about Passas’s concept, in that the interfaces between mostly-legal and mostly-illegal actors become fundamentally questionable (Ibid.) Criminal networks, as seen above in the Epstein case, intersect at least with legitimate public and private institutions. Sometimes that is because criminal networks (ab)use the legitimate infrastructure for their illegitimate goals (Tops and Tromp 2017; Huisman and Kleemans 2017). However, oftentimes it is because those same legitimate public and private institutions are not ‘infiltrated’ but rather themselves develop improper dependencies (Tjeenk Willink 2021; Thompson 2017; Lessig 2015; Passas 2005; Reich 2021). As an effect, respected and legitimate actors do not only have to be crime victims or become ‘intruded’ or ‘corrupted’, but rather can and often are corrupting themselves (Ibid.). This means that a clear line cannot always be drawn between unethical or unknowingly facilitative actors, on the one hand, and organized crime, on the other.
This problem of drawing that dividing line shows in links between human trafficking and the legitimate economy. To give examples beyond the Epstein case, nowadays most corporations are aware that they may (in)directly cause or facilitate human trafficking or use goods or services made with forced labor (e.g., Shelley 2010). Tech companies, hotels, recruitment agencies, and other legitimate companies appear to play a central role in facilitating human trafficking.[46] For instance, human trafficking is front and center in debates about how Meta and other Big Tech giants[47] fail to ensure integrity on their platforms.[48] The Facebook papers showed that, after a BBC investigation into domestic servitude, Facebook scrambled to address human trafficking content after Apple threatened to kick its apps off the iOS App Store. Also, for sex trafficking, there is an intersection of the legitimate economy in the hotel sector and locales such as Airbnb or Vacation Rentals by Owner (VRBO).[49] Moreover, as in the Epstein case, human trafficking also affects banks. To give an additional example, an Australian human trafficking case involved its first bank and oldest company Westpac which settled for 1.3 billion dollar.[50] There is a twofold reason to emphasize this effect of human trafficking on the integrity of the financial sector. In the first place, most people are well aware that, after the financial economic crisis in 2008, none of the banks or their executives have been convicted or even indicted for any offence (Reich 2021; Tjeenk Willink 2021).[51] In the second place, many worry that banks may not have made fundamental changes to their operations, so that their capabilities in helping maintain the integrity of the financial system are put into question. Trust in banks after the financial economic crisis is low. This understanding of how money and influence can speak louder than the will of the people can be further illustrated by focusing on class justice. People notice that often lower rungs of criminal networks rather than masterminds are held accountable. This causes erosion of trust in the state because there is a flipside to this too. People do see how supposedly legitimate business, which are oftentimes more responsible for sustaining the (financial) structures of organized subversive crime, usually remain scot-free. Class justice even happens within criminal procedure because ‘(t)he more sophisticated and knowledgeable the criminal, the more valuable is his cooperation and the more benefit he can obtain and offset the punishment which might otherwise have been imposed’ (Burgis 2020, citing Judge Glaser, p. 87). So low-level drug dealers, couriers, or money mules who have no information to give to the government, do suffer the sentence which the law requires. It stands to reason that the corrosive effect of dirty and dark money on democratic and rule-of-law institutions is far greater where governments have placed decades-long austerity measures on the public sector including on criminal investigators and the prosecution (so-called ‘more with less’-policies; Reich 2021; Tjeenk Willink 2021). In liberal democracies, the oftentimes squeezed public sector will have to investigate the crime of human trafficking which is ever evolving moreover. Such continuous evolvement makes it hard for law enforcement to detect newer forms of this crime and for research to keep up. Now labeled the blind spot in the context of organized subversive crime in the Netherlands, there are not only indications of especially (vulnerable) youth being recruited into drug crime, but also their possible involvement in the above-referenced openly violent murders of Mr. De Vries and Mr. Wiersum as so-called ‘hitters’ or ‘spotters’ (CKM 2021). Hitters do the actual killing, usually for low payment. Spotters are on the lookout during such an assassination. As a related issue, criminals tend to be opportunistic in that they go where ‘business’ opportunities drive them. As a cynical remark about the latter, commodities like drugs can be used once only, whereas persons can be exploited over and over (Coster van Voorhout 2020 and 2016). Persons are thus a continuous source of ‘income’. So, if human traffickers like Mr. Epstein can commit their crime with near impunity and are not hurt in their wallet, the risk is that they grow more powerful and (increasingly) negatively impact democratic and rule-of-law institutions as well as their representatives.
Having focused mostly on corrosion that happens within the bounds of a liberal democracy, erosion coming from outside is equally relevant.[52] Foreign human traffickers or governmental representatives who benefit from cheaper forced labor in their economy such as in Russia usually store their dirty or dark money in liberal democracies because those states offer rule of law protections for their wealth (TIP report 2022).[53] For example, Russia’s government has a policy or pattern of human trafficking for instance insofar as North Korean forced laborers within its territory is concerned (Ibid.). Additionally, the fuller scale invasion of Ukraine on 24 February 2022 makes millions of Ukrainians and Ukrainian residents vulnerable to human trafficking (Ibid.). While trafficking in persons is often assumed to be conducted by organized criminal groups, a state can thus also be involved in this crime (Ibid.). The significant benefits to the Russian economy that human trafficking provides mean that President Putin is unlikely to address the problem in the near future (Ibid.). In law, it is moreover important that the first case before the European Court of Human Rights concerning sex trafficking involved a Russian national who was found dead on Cyprus.[54] Now Russia has left the Council of Europe, thereby leaving its nationals without protection from this Court. On financial matters, it is also of relevance that Russia’s current and former officials and oligarchs are known to store their illegal gains in established democracies (Transparency International 2022; Snyder 2018). The vast majority of Russian-owned foreign assets are shrouded in secrecy, however (Ibid.). The sanctions by the United States and the European Union to which the Netherlands is a member pierce that veil of secrecy to some extent. Further initiatives including by the Biden Administration to create a global register for ultimate beneficial ownership intends to prevent financial secrecy and promote financial integrity in the future.[55] Russian placement of illegal proceeds from human trafficking in liberal democracies can advance their erosion. This placement of dirty and dark money in countries with good governance and rule-of-law protections is deliberate, so that the funds remains available to them (e.g., Bullough 2019; Burgis 2020; Vogl 2022).[56] The sanctions on Russian companies and individuals by the United States and the European Union demonstrate may have an effect on human trafficking given that by estimation 90% of human trafficking happens in the private sector but benefits the Russian government.[57] Also, (foreign) human traffickers or oligarchs can still often hide that amount of money through shell companies and anonymous ownership. By estimation, the amount of current offshore money is $7 trillion, with 10% of world GDP held offshore (UNDESA 2020). Hence, dirty and dark money has its tentacles in a network of liberal democracies and other states around the globe in all the aforementioned ways. The historian Mr. Timothy Snyder who conducts historical research into this connection between Russia and its influence on the decline of liberal democracy in Europe and the United States explains the following about the importance of the corrosive effect of dirty and dark money, though not directly in relation to much of Russian wealth made through the human trafficking aspects of its labor economy (TIP report 2022). Snyder argues: ‘In the 2010s, the United States approached the Russian standard of inequality. Although no American oligarchical clan has as yet captured the state, the emergence of such groups in the 2010s (Kochs, Mercers, Trumps, Murdochs) was hard to miss. Just as Russians used American capitalism to consolidate their own power, Americans cooperated with the Russian oligarchy with the same purpose –in the 2016 Trump presidential campaign, for example. Most likely, Trump’s preference for Putin over Obama was not just a matter of racism or rivalry: it was also an aspiration to be more like Putin, to be in his good graces, to have access to greater wealth. Oligarchy works as a patronage system that dissolves democracy, law, and patriotism. American and Russian oligarchs have far more in common with one another than they do with their own populations.’ (Snyder 2018, pp. 263-264). Snyder adds: ‘Tyrants first hide and launder their money, then use it to enforce authoritarianism at home – or export it abroad. Money gravitates to where it cannot be seen, which in the 2010s was in a various offshore tax havens. This was a global problem: estimates of just how much money was parked offshore, beyond the reach of national tax authorities, ranged from $7 trillion to $21 trillion. The United States was an especially permissive environment for Russians who wanted to steal and then launder money. Much of the Russian national wealth that was supposed to be building the Russian state in the 2000s and 2010s found its way to shell corporations in offshore havens. Many of these were in America.’ (Snyder 2018, p. 261).
So, liberal democracies will have to work in concert to tackle the corrosive effect of human trafficking and its related corruption within their states and when such corrupting influence comes from outside by autocratic states. The latter should ideally also themselves want to counter this crime. A government of a liberal democracy must be active in countering human trafficking by taking a whole-of-society approach and, where relevant, collaborating internationally including with the private sector (Ibid.). Corporations should do enough to end their role in facilitating this crime, also through corporate social responsibility
(CSR; Reich 2021). Banks specifically should be wary that human trafficking usually involves so much money that perpetrators have to give it a pretend legal origin. Even if earned in cash bills or coins, in our increasingly cash-free societies they or their stooges must usually ‘bring’ these ‘profits’ into the legitimate digital financial sector. When reinvesting proceeds in the crime business or using it on their luxurious lifestyle, this intersects with the digital financial market (Tjeenk Willink 2021; Reich 2021). Even cryptocurrencies or underground banking can be understood as modeled on financial flows through this infrastructure. NGOs should support victims of human trafficking, while also seeking to examine related corruption. The general public in a liberal democracy must be wary to not inadvertently contribute to human trafficking and related corrosion of their state, for example, when purchasing goods made or services provided by victims of trafficking in persons. As a final example, academia should study both human trafficking and related corrosion of liberal democracies.
3.3.2. Beyond human trafficking: Understanding systemic issues of organized subversive crime
The above-provided understanding of human trafficking that also takes into account newer forms like the forced commission of (drug) crimes also helps us lay bare human aspects of organized subversive crime in and beyond human trafficking. It helps facilitating our understanding that criminal networks can force persons into being perpetrators, victims or (un)witting facilitators of a range of crimes that, like human trafficking, erode democracy and the rule of law. This focus on human trafficking assists exploring a systemic issue of organized subversive crime: the persons involved in organized subversive crime who should be recognized as victims of the crime of human trafficking. This also helps determining what are and what are not humane interventions like policing; regulation; business transparency through human rights due diligence; and repression in which vulnerable persons are protected by the state (Tjeenk Willink 2021; Cf. Rawls 2001 and Sandel 1998). For example, in a democratic society governed by the rule of law, we only hold to account those who intend(ed) to commit a crime rather than those who were forced to do so. This refers to the non-prosecution and non-punishment principle established in relation to forced drug couriers.[58] For instance, the European Court of Human Rights has recognized that victims of human trafficking should not be mistaken for perpetrators because one crime (human trafficking) can be hidden in another (drug trafficking).[59]
3.4. Conclusion
To conclude, there are at least four categories of reasons that explain why human trafficking contributes to the subversion of liberal democracies worldwide: (i) financially, because of its lucrativeness that has demonstrably been used to corrupt public and private institutions and their representatives; (ii) developmentally, because of its ever-evolvement into newer forms like forced commission of -drug- crimes, for example; (iii) structurally, because of its embedding in (il)licit state structures; and (iv) conceptually, because at least some behavior is not criminalized whereas it does support both human trafficking and corrosion of democracy and the rule of law in established liberal democracies. All four aspects can result in a negative spiral: the sense that a liberal democracy is out of the public’s control creates frustration, spurs anger and resentment, and drives polarization in the citizenry which can, in turn, result in even less trust in the state. Those effects hardly ever stay within the realm of one country in our interconnected world with a near global financial sector and technology realm.
This conclusion should not be misunderstood as if this article argues that perpetrators of human trafficking inevitably commit a crime that does not exploit but even undermines state structures and results in a sense of injustice and a loss of trust by the general public in democracy and the rule of law. For instance, there may be instances where a human trafficker abuses one victim without the larger effect of eroding liberal democracy more broadly. To give a potential example, usually a trafficker who exploits a single victim domestically in a one-on-one pretend loving relationship merely out of financial gain may not have the larger effect of hurting democracy or the rule of law to the extent needed to legitimately qualify as a corrosive effect on liberal democracies (the poor term developed by the Dutch: ‘loverboy’; Bovenkerk et al 2007). While the individual victim and their family or community may consequently lose trust in democracy and the rule of law, oftentimes it will require a larger scale human trafficking operation to have the wider more fundamental corrosive effect on liberal democracy more generally. This is not to say that the accumulation of many such cases cannot create the inadvertent permissive environment that results in erosion of liberal democracies.
To sum up, networked forms of human trafficking that moreover ‘make’ perpetrators the amounts of illegal proceeds mentioned and blur the divisions between under- and upperworld can oftentimes be better understood if examined as organized subversive crime. This means that there are instances of human trafficking that must be seen from this perspective of how the harm is not only done to victims, their families and their communities but also to, by extension, liberal democracy or, in our interconnected world, multiple liberal democracies worldwide. As has been explained above, human trafficking can thus qualify as organized subversive crime and even other offenses like drug trafficking involving forced couriers who are victims of human trafficking can fit that bill. This should not be misread to mean that all organized subversive crime involves trafficked persons who are forced to work in (sex) labor or commit offenses. But some may be. As a related point, this also does not imply that organized subversive crime cannot be committed without any laundering of illegal proceeds derived from criminality. For example, some offenses will still be committed through violence ‘only’ and some of the illegal proceeds will exclusively circulate among criminals. However, oftentimes these organized crimes that are specifically committed because of their significant financial or other material gains can be better understood if the damage to liberal democracies will also be taken into account. As a final qualifying remark, it does not mean either that all organized subversive crime necessarily involves ‘upperworld’ enablers like tax advisors, lawyers and consultants. Some of this criminality can fully stay within the realm of the criminal milieu. However, it does mean that sometimes human trafficking is coupled with money laundering or corruption so that it becomes necessary to fully understand its contribution to the erosion of democracy and the rule of law by also taking into account its corrosive effect on liberal democracy.
For all these reasons, human trafficking is, as a research notion, a wicked problem (e.g., Cels, De Jong & Groenleer 2017). A wicked problem ‘lacks clarity in both its aims and solutions, and is subject to real-world constraints which hinder risk-free attempts to find a solution’ (Ibid.). It is ‘a social or cultural problem that is difficult or impossible to solve because of its complex and interconnected nature’ (Ibid.). Consequently, research will have to pay special attention to the detection of human trafficking, particularly in spheres that are less known as susceptible to this crime such as labor sectors, domestic work, the forced commission of crimes, and organ removal. For this, it helps to use both empirical and normative findings to understand the systemic issues of organized subversive crime. But before explaining further how to better detect the forms of human trafficking that constitute organized subversive crime in fact, it is important to turn to fiction so as to examine how myths about this offense as conspiracies are also found in the mix of the purported decline of liberal democracies worldwide.
[1] Letter from US Senators Ben Sasse, Richard Blumenthal, Ted Cruz and Marsha Blackburn to the Inspector General of the US Justice Department, available at: https://www.sasse.senate.gov/public/index.cfm/2019/12/ben-sasse-to-doj-inspector-general-finish-the-epstein-investigation.
[2] European Court of Human Rights (ECtHR) 16 February 2021 V.C.L. and A.N. v. the United Kingdom concerning forced cannabis couriering.
[3] E.g. This complexity is exemplified by inter alia the fact that human trafficking has the longest definition in the Dutch criminal code (e.g. Article 273f Sr; cf. Article 3a of the Palermo Protocol on human trafficking).
[4] E.g. As the modern-day interpretations of the articles regarding slavery, enslavement, servitude and related phenomena, because the term of human trafficking was not common at the time, in Article 4 of the Universal Declaration of Human Rights (UDHR), Article 8 of the International Covenant on Civil and Political Rights (ICCPR) and Article 4 of the European Convention on Human Rights (ECHR). Later also referred to in Conventions like the one on the rights of the child and the prevention of discrimination against women.
[5] E.g., police statistics are usually based on filed police reports and not every victim reports a crime they were victimized by, while global data, such as from UNODC or as gathered in the US TIP reports, are taken from reports by governments based on contact of victims with authorities.
[6] The U.S. State Department’s annual Trafficking in Persons (TIP) Report reviews responses by governments to combat human trafficking worldwide.
[7] The description of the Epstein case with a view to explaining its corrosive effect is especially important because such human trafficking cases evidently stay hidden for a long time, at least insofar as their full scope and networked form are concerned. Due to the limited scope of this article, however, not all aspects of this large-scale sex trafficking case can be examined in full. Also, this would be impossible at present, given that, despite Mr. Epstein’s death, some elements are still being investigated in several jurisdictions around the world. The below description therefore focuses on the legal facts and procedure followed in the Epstein case that indicate how human trafficking can have a corrosive effect on democracy and the rule of law. Case details have been derived from, inter alia, Unites States Court of Appeals for the eleventh Circuit, case No. 19-13843, re: Courtney Wild, D.C. Docket No. 9:08-cv-80736-KAM, p. 5, available at: https://www.courthousenews.com/wp-content/uploads/2021/04/wild-epstein-ca11.pdf. United States District Court Southern District of New York, United States v. Ghislaine Maxwell, Defendant, Sealed Indictment, 20 Cr. 330, available at https://www.justice.gov/usao-sdny/press-release/file/1291491/download. Cf. https://www.justice.gov/usao-sdny/pr/ghislaine-maxwell-sentenced-20-years-prison-conspiring-jeffrey-epstein-sexually-abuse. Cf. Brown 2021.
[8] By then the state prosecutor, according to the police, had already changed his position of seeking to prosecute Mr. Epstein. E.g., “Privately, Recarey and Reiter were baffled by the change in Krischer’s [the Palm Beach Florida state attorney] approach to the case. He went from ‘let’s get him’ to ‘why do you want to subpoena those records?’ “Recarey recalled.” (Brown 2021, p. 83).
[9] This non-prosecution agreement listed the following federal crimes: (1) using and conspiring to use a facility of interstate commerce to persuade, induce, or entice minors to engage in prostitution, in violation of 18 U.S.C. §§ 2422(b), 371, and 2; (2) traveling and conspiring to travel in interstate commerce for the purpose of engaging in illicit sexual conduct with minors, in violation of 18 U.S.C. § 2423(b) and (e); and (3) recruiting, enticing, and obtaining a minor to engage in a commercial sex act, in violation of 18 U.S.C. §§ 1591(a)(1) and 2. The Agreement extended immunity to Epstein’s named co-conspirators, ‘Sarah Kellen, Adriana Ross, Lesley Groff, [and] Nadia Marcinkova,’ as well as ‘any potential co-conspirators’ of Epstein’s. The latter are thus unnamed possible co-conspirators.
[10] ‘Acosta would later contend that he agreed to give Epstein federal immunity from sex trafficking charges based on the unlikely success that prosecutors felt they would have at trial. Even with the little bit that I [Brown] knew about the case in 2016, this never made sense to me. After all, immunity is a benefit granted in exchange for something else of value to prosecutors. What, if anything, did federal authorities get for giving Epstein and his co-conspirators –both named and unnamed – immunity?’ (Brown 2021, p. xiii).
[11] The introduction of prostitution, rather than trafficking, charges may have put additional pressure on victims to not press for prosecution of Mr. Epstein and his co-conspirators because prostitution is an illegal offense in Florida. “Belohlavek [the female prosecutor] was also concerned that, under state law at the time, minors as young as fourteen could be prosecuted for prostitution, meaning the girls could be charged.” (Brown 2021, p. 83).
[12] Brown 2021, pp. xi-xii.
[13] According to the Unites States Court of Appeals for the eleventh Circuit ‘Following a tip in 2005, the Palm Beach Police Department and the FBI conducted a two-year investigation of Epstein’s conduct’ (p. 3). The Dissenting opinion of Judge Branch, who is joined by Martin, Jill Pryor and Hull, makes this even more explicit ‘following a 2005 report by the parents of a 14-year-old girl that then 52-year-old billionaire Jeffrey Epstein sexually abused their daughter, local Florida authorities—and later the FBI—began investigating Epstein’. (p. 100). ‘In June 2008, Epstein pleaded guilty to the state crimes as agreed and was sentenced to 18 months’ imprisonment, 12 months’ home confinement, and lifetime sex-offender status.’ (p. 5). In case No. 19-13843, re: Courtney Wild, D.C. Docket No. 9:08-cv-80736-KAM, available at: https://www.courthousenews.com/wp-content/uploads/2021/04/wild-epstein-ca11.pdf
[14] E.g. Case 19 Civ. 3377 (LAP), Virginia L. Giuffre, Plaintiff, v. Alan Dershowitz, Defendant, 10-16-2019, available at: https://casetext.com/case/giuffre-v-dershowitz.
[15] Both her series in the Miami Herald and her book, from which citations will be provided below, are entitled Perversion of Justice. For example, Brown compares the regular course of justice with the one in the Epstein case: ‘Recarey [the lead detective on this case in South Florida], in a fatherly tone, tried to reassure them [the victims] that they would indeed be safe and that Epstein would be arrested, telling them: “It doesn’t matter how much money you have or how many connections you have, if you commit a crime then you will be punished. That’s the way our justice system works.’.’ (Brown 2021,p. 16). She adds: “One of the many mysteries of the Epstein case is how he got away with such flagrant sex crimes at a time when the FBI was cracking down on child exploitation and putting away men for decades for far lesser sex crimes. In 2006, the Justice Department under President George W. Bush had launched a task force focused on sex crimes against children. Hundreds of arrests and prosecutions happened during these years. Although the effort focused largely on child pornography, combating human trafficking was also one of its aims, even though I later came to learn that trafficking, at least back then, was seen by law enforcement as a largely foreign phenomenon perpetrated mostly by black and brown people who came from other countries. The Justice Department didn’t seem to fathom that sex trafficking could be a pervasive crime committed by well-to-do and powerful people in the United States. Or that pornography – especially child pornography – was fast becoming a multibillion-dollar worldwide industry.” (Brown 2021, pp. 17-18).
[16] Supra note 13.
[17] United States District Court Southern District of New York, United States v. Ghislaine Maxwell, Defendant, Sealed Indictment, 20 Cr. 330, available at https://www.justice.gov/usao-sdny/press-release/file/1291491/download. Cf. https://www.justice.gov/usao-sdny/pr/ghislaine-maxwell-sentenced-20-years-prison-conspiring-jeffrey-epstein-sexually-abuse.
[18] E.g. Robertson, Linda; Brown, Julie K.; Nehamas, Nicholas (20 December 2019). "Did a Miami-based modeling agency fuel Jeffrey Epstein's 'machine of abuse'?". Miami Herald. Retrieved 21 March 2020. Cf. "Jean-Luc Brunel: three former models say they were sexually assaulted by Jeffrey Epstein friend". The Guardian. ISSN 0261-3077. Retrieved 21 March 2020.
[19]New York State, Department of Financial Services, in the matter of Deutsche Bank AG, Deutsche Bank AG New York Branch and Deutsche Bank AG Trust Company of the Americas, consent order under New York Banking Law §§ 39 and 4 , available at: https://www.dfs.ny.gov/system/files/documents/2020/07/ea20200706_deutsche_bank_consent_order.pdf.
[20] The description of these corrosive factors is inevitably hampered by the fact that it will take years –if not decades– to fully understand the Epstein case in its networked form, in part because the integrity of public and private institutions as well as their representatives who should have investigated the crimes in full seem to have gotten eroded themselves. However, it is possible to trace some of the most relevant corrosive factors from the Epstein case based on literature that seeks to explain the decline of liberal democracies worldwide (e.g., Diamond 2019; Mounk 2019; Levitsky and Ziblatt 2018; Guriev and Treisman 2022).
[21] On the issue of celebrity, former R&B star R. Kelly was sentenced to 30 years in prison on 28 June 2022, the same day when Ms. Ghislaine Maxwell was sentenced to 20 years imprisonment, for racketeering and sex trafficking, charges stemming from nearly 30 years of allegations that he physically and sexually abused women and minors. Available at: https://www.justice.gov/usao-edny/pr/r-kelly-convicted-all-counts-federal-jury-brooklyn.
[22] The UNODC which examined human trafficking-related corruption examined consequences such as fraudulent travel or identity documentation made by bribed customs officials or actual illegal border crossings facilitated by agents who turned a blind eye (UNODC 2011). From such consequences, the potential corruption can be inferred.
[23] ‘Epstein didn’t need to use flamboyant charisma or to showboat his smarts to build his international network of influential people; he did that the old-fashioned way – with his money.’ (Brown 2021, p. 128)
[24] Brown 2021, p. 77.
[25] ‘I also learned that Epstein likely conducted video surveillance in every home he owned. As insurance, he probably had tapes and photographs of important visitors – mainly men – in compromising situations. Whether that was true or not, even the possibility that he had blackmail material was enough motive for many powerful people to do everything possible to cover up Epstein’s crimes.’ (Brown 2021, p. xiv).
[26] ‘One of his specialties was helping the super wealthy – as well as himself – to avoid paying taxes. Yet he was never in the Forbes 400 list of the wealthiest Americans, largely because the magazine was never able to determine the true size of his fortune.’ (Brown 2021, p. 128)
[27] See sub-section 3.4.1.2.
[28] E.g. ‘This seemed suspicious to Michelle [one of Epstein’s named victims]. So she hired another lawyer, not paid by Epstein. As a mediation, Epstein’s lawyers gave her an offer she would have to accept or reject immediately. She was troubled when her attorney informed her she couldn’t talk to her parents about the settlement first. By then, Epstein’s lawyers had already sent a message that they intended to destroy her and her family.’ (Brown 2021, pp. 111-112)
[29] Citation from Unites States Court of Appeals for the eleventh Circuit, case No. 19-13843, re: Courtney Wild, D.C. Docket No. 9:08-cv-80736-KAM, p. 5, available at: https://www.courthousenews.com/wp-content/uploads/2021/04/wild-epstein-ca11.pdf.
[30] Initially Judge Marra ruled that this lack of informing the victims about the NPA had violated the rights of the petitioning victim Ms. Courtney Wild under the Crime Victims’ Rights Act, 18 U.S.C. § 3771 (VCRA), but the cited court’s majority did not reach that conclusion. Supra note 35.
[31] ‘Two weeks later, Epstein donated money for the firearms simulator. Reiter [the local police chief] assigned someone to start looking into purchasing the equipment. But shortly thereafter, he learned about the investigation into Epstein and put a hold on the purchase. As time went on with the case, Reiter began to suspect that Epstein’s altruistic endeavors were aimed more at influencing the police than they were at helping them. Reiter was smart enough not to return the money immediately. He didn’t want to alert Epstein or jeopardize their investigation.’ (Brown 2021, p. 125). ‘ “His enthusiasm in making contact and in finalizing the donation was somewhat suspicious, different in manner than when he made previous donations and suggested that he may have become aware of the investigation at that time,” Reiter wrote in a later report.’ (Brown 2021, p. 126)
[32] This reference to at least a decade has been made, because it is hard to know when the Epstein human trafficking operation began. As Brown states: ‘It’s difficult to know how and when Epstein’s scheme began. What is known is that in 1998, Epstein’s then girlfriend, the British socialite Ghislaine Maxwell, began visiting colleges, art schools, spas, fitness centers, and resorts in and around Palm Beach County [Florida, the United States], under the guise that she wanted to hire young and pretty masseuses or “assistants” to come to Epstein’s home and work for him.’ (p. 10). Also, ‘Epstein promised to rescue them, but at a cost: not only were they expected to perform for him sexually, but in some cases, they were pressured to have sex with other men old enough to be their grandfathers.’ (p. 10). ‘Epstein’s houseman, Juan Alessi, was ordered to drive Maxwell from resort to resort for her to hand out business cards and recruit massage therapists for Epstein. Alessi was skeptical of her motives, especially when the first who began coming to the house looked as young as Alessi’s daughter.’ Brown 2021, (p. 10).
[33] ‘Recarey [the investigating agent] knew full well that the grand jury schedule set by prosecutors – and Epstein’s lawyers – was designed to fail, and, in an interview, he told me how and why he believed that they were throwing the case. It was part political, he said, because Epstein was a million-dollar donor to the Democratic Party, which controls Palm Beach. Krischer was reminded that Clinton was friends with Epstein, and there were a lot of other political heavyweights also tied to Epstein, including George Mitchell. If Epstein’s secrets got out in a big way, it would hurt the party. Kirscher was a powerful force in Palm Beach politics, and it was up to him to contain the case.’ (Brown 2021, p. 90).
[34] E.g., then Former President Mr. Bill Clinton whose former top advisor attests he had been on the island St. James where some of the abuse took place and now Former President Mr. Donald Trump who is also alleged to have sexually assaulted some of the victims). In New York Magazine: ‘ “I’ve known Jeff for fifteen years. Terrific guy,” said Donald Trump, fifteen years before he would be elected president. ‘He’s a lot of fun to be with. It is even said that he likes beautiful women almost as much as I do, and many of them are on the younger side. No doubt about it – Jeffrey enjoys his social life.”.’ (Brown 2021, p. 42). For example, Brown details: ‘The truth is there were powerful people on both sides of the political rails – as well as people ion the worlds of finance, academia, and science – who were involved with Epstein or, at the very least, complicit with what he was doing.’ (Brown 2021, p. xiv).
[35] ‘The message pads also contained a who’s who of influential people calling Epstein: Donald Trump, Les Wexner, former J.P. Morgan banker Jes Staley, real estate mogul Mort Zuckerman, former Maine senator George Mitchell, and Hollywood producer Harvey Weinstein.’ (Brown 2021, p. 81).
[36] To some extent that also happened in the Epstein case though the local police did not have it affect them too much: ‘The character assassination against Reiter and Recarey mounted as Epstein’s lawyers went beyond reasonable efforts to defend their client.’ ‘They knew that Epstein was trying to smear them in order to discredit their case – and it was working. Furthermore, Epstein made it clear, through his emissaries, that his victims had better keep their mouths shut or they would regret it. some of them were afraid for their lives.’ (Brown 2021, p. 138 and p. 139 respectively)
[37] Obviously in the Epstein case no prosecution and thus no adjudication by judges of his case took place. It has not been examined by investigative journalists and others yet where the difference between the initial judge and the majority in the victims’ case comes from. This case could end up at the US Supreme Court. Allegations that US courts in general and the Supreme Court in particular have been ‘captured’ have been made, inter alia, by Senate Democrats, led by Democratic Policy and Communications Committee (DPCC) Chairwoman Debbie Stabenow (D-MI) and Senate Judiciary Courts Subcommittee Chairman Sheldon Whitehouse (D-RI), released a new report in their Captured Courts series on the dark-money group the Judicial Crisis Network (JCN) and its role in right-wing donor interests’ scheme to capture and control the Supreme Court. Available at: https://www.democrats.senate.gov/imo/media/doc/Captured%20Courts%20Report%204-5-22.pdf. Globally, corrupt criminal justice professionals have also been found to facilitate ineffective investigations or prosecutions in exchange for bribes (UNODC, 2011).
[38] Cf. Money can also corrupt politics, as in the US has been investigated for legitimate business representatives that as stealthily as Epstein have used such funds – in their case – to weaken labor unions and labor law standards, like the Kochs and DeVoses (e.g., Mayer 2017; Bernstein 2020).
[39] ILO 2014. Less than 1% of illegal proceeds are confiscated or frozen, according to the United Nations Office on Drugs and Crimes (UNODC 2011). I use this estimation, while I realize its limitations to which I also want to bring your attention. Even within countries there is a lot of uncertainty about even the definition of crime, so that a global estimation of profit is inevitably hindered by many assumptions. But nonetheless it gives us a – though imperfect – indication of the money potentially made through this crime. In the European Union and several other developed countries the income of sex trafficking has been estimated to be 23,5 billion euros (ILO 2014).
[40] Formerly known as Facebook.
[41] This amount is also comparable to the economy of Morocco in 2022 and Kuwait in 2021. Also, it is some $11 billion more than the estimated wealth of Mr. Bill Gates in 2021.
[42] Directive 2011/36 EU 1.
[43] ECtHR 16 February 2021 (V.C.L. and A.N. v. the United Kingdom).
[44] This critique can be added to the fact that the $300 billion made through drug crime is based on street value rather than raw materials which is remarkable because this is the only crime for which calculations are based on price inflation (cf. Wainwright 2017).
[45] His dissenting opinion in McCutcheon et al v. Federal Election Commission.
[46] E.g. The Texas Supreme Court on 27 June 2021 ruled that Facebook can be held liable for sex traffickers that use its platform to recruit and prey on child victims. Available at https://search.txcourts.gov/SearchMedia.aspx?MediaVersionID=a1c6da82-96ba-48b8-87d3-e5556420b8b4&coa=cossup&DT=OPINION&MediaID=2b5b90ae-2e7e-41ac-9ad6-aceaa1159fc0.
[47] E.g. Kang, Cecilia; Frenkel, Sheera (June 27, 2020). "'PizzaGate' Conspiracy Theory Thrives Anew in the TikTok Era". The New York Times. Archived from the original on June 27, 2020. Available at: https://www.nytimes.com/2020/06/27/technology/pizzagate-justin-bieber-qanon-tiktok.html.
[48] E.g. A leaked SEV (or Site Event) report shows, referenced briefly by The Wall Street Journal’s Facebook Files reporting, indicates that Apple threatened to pull Facebook and Instagram from iOS on October 23rd of 2019. Available at: https://www.theverge.com/22740969/facebook-files-papers-frances-haugen-whistleblower-civic-integrity.
[49] E.g. The initiative of the Dutch prosecution and police to work with all sex sites to counter forced prostitution; cf. launderettes for the hotel sector in relation to human trafficking for the purpose of labor exploitation. Court of Amsterdam, ECLI:NL:RBAMS:2018:1631.
[50] Cf. the Australian bank Westpac has agreed to pay a record penalty of $1.3bn to settle legal action over money laundering and human trafficking for the purpose of child exploitation allegations levelled against it by the financial intelligence agency, Austrac. Westpac is also one of the largest banks in New Zealand. Available at: https://www.austrac.gov.au/news-and-media/media-release/austrac-and-westpac-agree-penalty.
[51] Cf. ‘Epstein’s brother, Mark, who lives in New York, said that his brother was a math whiz who mastered Wall Street at a time when it was “the wild, wild West.” Mark said Jeffrey would often mention how easy it was to manipulate investors. “He said if the general public knew what was taking place on Wall Street, there would be a revolution. People would be appalled by how corrupt it was.’ (Brown 2021, p. 130)
[52] Cf. Sergei Guriev and Daniel Treisman who explain ‘Today’s autocracies pose new challenges to the democracies of the West.’ (p. 205). Several of these autocracies are leaders in government-run human trafficking (TIP report 2022). Russia is one of 11 governments with a documented “policy or pattern” of human trafficking, trafficking in government-funded programs, forced labor in government-affiliated medical services or other sectors, sexual slavery in government camps, or the employment or recruitment of child soldiers: Afghanistan, Burma, China, Cuba, Eritrea, North Korea, Iran, Russia, South Sudan, Syria, and Turkmenistan.
[53] Cf. Frank Vogl, the founder of Transparency International, in his 2022 book: ‘Across the world, leaders of authoritarian governments, and their cronies, are robbing their people. These leaders are kleptocrats and they are pocketing staggering sums of cash, which they move through the world’s financial system into investments in the wealthiest Western nations. These crimes perpetrated by kleptocrats governing Russia, China, Iran, Egypt, Hungary, Nigeria, and many more nations not only impoverish their own citizens but all of us. More gallingly, we are assisting them in their greed and their grand corruption. Even more worrying, we are complicit in their quest for ever greater power. Central to Western complicity with kleptocrats and their associates across the globe are the armies of financial and legal advisors, real estate and luxury yacht brokers, art dealers and auction house managers, diamond and gold traders, auditors, and consulting firms, based in London and New York and in other important global business centers, who aid and abet the kleptocrats in return for handsome fees– these are the enablers. They are motivated not only by the widespread failures of law enforcement across the Western democracies to impose punishments that are sufficient to serve as meaningful disincentives. At the major banks, for example, who have been prosecuted at times for multi-billion dollar laundering of dirty cash, not a single chairman or chief executive officer has personally faced criminal charges for such activities, while the fines that are agreed to settle legal actions appear, quite simply, to be viewed by bankers as just the costs of doing business.’ (Vogl 2022, pp. 1-2)
[54] ECtHR, Rantsev v. Cyprus and Russia, Application no. 25965/04, 10 October 2010.
[55] E.g. Memorandum on Establishing the Fight Against Corruption as a Core United States National Security Interest, 3 June 2021, Presidential actions, available at: https://www.whitehouse.gov/briefing-room/presidential-actions/2021/06/03/memorandum-on-establishing-the-fight-against-corruption-as-a-core-united-states-national-security-interest/.
[56] Corrupted enablers in Western democracies assist organized crime (cf. Vogl 2022 and Guriev and Treisman 2022). Also some of the actions of Western democracies themselves have a damaging effect on democracy and the rule of law including on the international rule of law such as the invasion of Iraq by the United States.
[57] E.g. Investigative journalists explore leaks like the Panama and Paradise Papers as well as the effects of US and EU sanctions on the Russian state and its oligarchs so as to examine what is known about secret finances of oligarchs in the West, available at: https://www.icij.org/investigations/pandora-papers/as-the-west-takes-aim-with-russian-sanctions-heres-what-we-know-about-oligarchs-secret-finances/.
[58] ECtHR 16 February 2021 (V.C.L. and A.N. v. the United Kingdom).
[59] This is not to say that globally including in Europe these principles are always complied with in practice, as UNODC has explained, for example in its report, UNODC, ICAT, The Inter-Agency Coordination Group
against Trafficking in Persons, Special Issue 8, available at: https://www.unodc.org/documents/human-trafficking/ICAT/19-10800_ICAT_Issue_Brief_8_Ebook.pdf.
- In section 4, the claim that human trafficking is the ‘patient zero’ of disinformation is a very ambitious one, not supported by evidence. I do not think it is sustainable without more evidence.
Agreed, and I have therefore improved into:
4. Conspiracies about human trafficking: Damage to democracy and the rule of law
4.1. Introduction
The two most influential conspiracies of our time, Pizzagate and QAnon, demonstrate how conspiracists spread false claims about human trafficking like wildfire, thus complicating the understanding of the general public of what actually constitutes human trafficking and what does not. Both conspiracies refer to sex trafficking rings, as will be explained further below. To put it in all too familiar Covid-19 terms, human trafficking may therefore very well be the present-day ‘patient zero’ of disinformation. Disinformation sows distrust in liberal democracies by attacking the truth[1] and eroding trust in public and private institutions and their representatives.[2] The fact that human trafficking is such a popular theme for conspiracists raises a number of questions. How do they abuse the crime of human trafficking in their conspiracies? Do they focus on a specific form of human trafficking and, if so, why do they do so? What are some of the effects that we can empirically notice? Do these conspiracies indeed contribute to the erosion of liberal democracy, for example because they motivate real-world, physical violence?
[1] As a consequence, this may also result in polarization which according to Levitsky and Ziblatt is a sure way to end democracy.
[2] Cf, though in relation to education rather than reporting research findings and conceptualizing the intent or effect of subverting liberal democracies, B. Benton and D. Peterka-Benton, ‘Truth as a Victim: The challenge of anti-trafficking education in the age of Q’, Anti-Trafficking Review, issue 17, 2021, pp. 113-131, https://doi.org/10.14197/atr.201221177. J Cook, ‘“It’s Out of Control”: How QAnon undermines legitimate anti-trafficking trafficking efforts’, Huffington Post, 14 September 2020, retrieved 25 September 2020, https://www.huffpost.com/entry/how-qanon-impedes-legitimate-anti-trafficking-groups_n_5f4eacb9c5b69eb5c03592d1.
- In section 5 (l. 643-646), once again, the claim that conspiracies about human trafficking and actual human trafficking should fall into the same category of crimes is not sufficiently justified. As much as conspiracies are dangerous, in democracies they need to be permitted to a degree at least given the right to free speech, and the suffering caused by trafficking seems incomparable to that triggered by (most) conspiracies.
Author: Agreed, and I have clarified it by making this exact point while improving the point intended that the incitement to (mob) violence as a consequence thereof is the reason why it should be considered as such.
Similarly to those actual examples of human trafficking and related crimes, criminals who spread conspiracies about sex trafficking with the intent or effect of damaging democracy and rule of law who thereby incite (mob) violence, could also fall in this category of organized subversive crime. To clarify this point, in liberal democracies, some degree to spreading of conspiracies will have to be permitted out of respect for free speech. However, where such conspiracies spill over in criminal conduct in the form of (mob) violence that line between legal and illegal conduct can be drawn.
- In l. 649-652 (same section), it is unclear whether the need for a new category of crime is prompted by the incapacity of the American legal system to prosecute those who stormed the Capitol. Are there problems with the current status of the legal framework for effectively addressing these instances? Without identifying such problems, the call for a new type of crime is less convincing.
Author, thanks, I've explained the difficulties in the US on this issue immediately in the first chapter, the introduction (especially in para. 1.2.1.):
1. Introduction
1.1. Context and central research question
This article addresses human trafficking in the context of the reported decline of liberal democracies worldwide (e.g., Diamond 2019; Economist Intelligence Unit 2021; Mounk 2019; Levitsky and Ziblatt 2018; Guriev and Treisman 2022).[1] There is no better place than this Special Issue to examine if, and if so how, some forms of trafficking in persons advance the erosion of core tenets of liberal democracies including democracy and the rule of law. Both human trafficking and this potential corrosive effect on liberal democracies are complex and difficult to detect because of their mostly hidden nature (e.g., Shelley 2012; Levitsky and Ziblatt 2018; Snyder 2018; Coster van Voorhout 2020). Therefore, to study both issues comprehensively, this article takes all relevant angles: human trafficking in fact and in fiction, and corrosion from within and outside liberal democracies. From all four angles, this article seeks to answer the following research question: Should some forms of human trafficking be dealt with as a new and unique category of crime or aggravating circumstance that emphasizes its corrosive effect on liberal democracies and, if so, under what conditions would that be useful in law and in research?[2]
1.2. Methodology and justification of substantive and geographical focus
The methodology used to answer this central research question is a combination of normative (‘how should it be?’) and, to a limited extent where taken from other predominantly doctrinal sources, empirical research methods (‘what is it?’), while social scientific, historical, and political scientific literature is referred to where relevant. One reason for stressing the importance of needing to obtain empirical findings in future research is the criminological research finding that police interventions that are founded only in theoretical assumptions risk contributing to strengthening criminal networks instead of weakening them (e.g., Luna-Pla and Nicolás-Carlock 2020; Thurner et al 2018; De Domenico et al 2019; Helbing et al 2015; Van Engers et al 2017; Duijn et al 2014). The predominantly doctrinal analysis in this article points out ways to find useful empirical insights in further research. Moreover, the substantive emphasis of this article is on criminal law. Geographically, the focus is on the Netherlands, and in a more limited manner on other European states, the United States, and Russia. Because of the limited scope of this article; other fields of law and countries will have to be reserved for future research.[3] The reasons for this geographical focus are explained further below.
1.2.1. Further justification of geographical focus
This article leads with the Netherlands because scholars and policymakers in that country seek to address a new challenge of crime that erodes its established liberal democracy. The Netherlands did not (yet) face an outright violent attack on its democracy and rule of law such as the storming of the Capitol on January 6th, 2021, in the United States. However, just like the Americans, the Dutch struggle with finding adequate solutions to damage done by crime to those essential tenets of their state, inter alia, because of high-impact murders. While American scholars and policymakers call for dealing with January 6th by introducing a not yet existent domestic terrorism charge, the Dutch are planning on establishing the concept of a crime with the intent to, or effect of, subverting a liberal democracy.[4] This article therefore explores to what extent this solution sought by the Dutch could be helpful in other contexts such as in the United States. This article does so, by focusing on a crime that has not been fully examined in relation to this new Dutch concept either: human trafficking. This examination of the relevance of this new Dutch concept for human trafficking is twofold. First, this article explores whether this novel Dutch concept should or should not encompass human trafficking. Second, the study determines whether a focus on human trafficking can explain the more systemic issues of this new concept in the broader sense. Turning as a final issue to the third country included below, Russia, this country is mentioned because of its export of conspiracies including about human trafficking to liberal democracies (e.g., Snyder 2018; Miller-Idriss 2022). While the emphasis is on Russia because of this spreading of disinformation, this state is also relevant in relation to the crime of human trafficking itself. For example, on 19 July 2022, Russia was put on the state-sponsored human trafficking list, thus indicating an often-overlooked aspect of trafficking in persons not committed by organized criminal groups but by the state itself (TIP report 2022).
1.3. Outline of this article
This article is divided into six parts. First, a description follows of the aforementioned new Dutch concept of crimes that undermine democratic societies governed by the rule of law that scholars and policy-makers are currently still developing (section 2). Thereafter, a factual examination as to whether human trafficking can qualify as a crime that undermines democratic societies governed by the rule of law follows (section 3). In the subsequent part, this article turns from fact to fiction, by focusing on how conspiracists abuse the phenomenon of human trafficking (section 4). Thereupon, the penultimate section deals with the usefulness of this new Dutch concept including for human trafficking (section 5). Finally, a conclusion is drawn and recommendations for further research are made (section 6).
[1] For an overview of data used to undergird this thesis and the conclusion itself, see the report by the Economist Intelligence Unit from 2021, available at: https://www.economist.com/graphic-detail/2021/02/02/global-democracy-has-a-very-bad-year. Cf. under the term ‘democratic deconsolidation’, e.g. Yascha Mounk and Larry Diamond (Yascha Mounk, People vs. Democracy – Why Our Freedom Is in Danger and How to Save It, Cambridge: Harvard University Press, 2019, and Larry Diamond, Ill Winds: Saving Democracy from Russian Rage, Chinese Ambition, and American Complacency, New York: Penguin Press, 2019. Some argue that these concerns about the decline of democracies worldwide are somewhat exaggerated but even they express concern about the degree to which what they call ‘spin dictators’ damage liberal democracies, oftentimes through erosion from outside and within (See Sergei Guriev and Daniel Treisman, Spin Dictators – The changing face of tyranny in the 21st Century, Princeton: Princeton University Press, 2022). Similarly, on the trend of the decline of liberal democracies itself, see for example Levitsky and Ziblatt, How Democracies Die: What History Reveals About Our Future, New York: Penguin Random House, 2018. Rather than decline through an outright coup, they argue that currently democracies rather ‘erode slowly, in barely visible steps’ (Levitsky and Ziblatt 2018, p. 3). More positive is a scholar like Pinker (Steven Pinker, Enlightenment now: The case for reason, science, humanism, and progress, New York: Viking, 2018). However, the author did not find any writings on this topic from Pinker after 2018, who may also have changed his opinion at least somewhat after January 6th, 2021 . Pinker’s language expertise was used by the Epstein defense (Brown 2021, p. 156).
[2] Supra note 1.
[3] This means that this article cannot delve into the entire range of democracies including its most populated, India, and the many others in North America (Canada), western Europe, Central and Eastern Europe, Latin America and the Caribbean, Asia and Australasia, the Middle East and North Africa, and Sub-Saharan Africa. See section 6.
[4] Another possibility could also be to charge the murder – and or in the future, potentially the person or persons who gave the order – with a crime that has been committed with ‘terrorist intent’. The latter is, unlike in the United States, in the Netherlands not reserved for foreign, international terrorism only.
In terms of the concepts, it is currently not clear how democracy, liberal democracy and the rule of law are related to each other, to what extent the authors use these concepts interchangeably versus intends to provide them with distinct meanings in the context of the article. I think it is important to clarify this relationship because it is a core point of contention in the vast scholarship on erosion of democracy.
Author, Thanks, I have explained this further in:
2.1. Defining liberal democracy, democracy and the rule of law
In this article, a liberal democracy is defined as a state that combines free and fair elections with the rule of law, fundamental rights, and institutional checks and balances. Democracy and the rule of law are thus deemed to be of equal significance to a liberal democracy (the German concept of ‘demokratischer Rechtsstaat’). This is the case because both democratic processes and the rule of law, rather than rule by men, are necessary but insufficient conditions for a liberal democracy. This understanding is in keeping with the definition used by the United Nations:
‘the rule of law refers to a principle of governance in which all persons, institutions and entities, public and private, including the State itself, are accountable to laws that are publicly promulgated, equally enforced and independently adjudicated, and which are consistent with international human rights norms and standards. It requires, as well, measures to ensure adherence to the principles of supremacy of law, equality before the law, accountability to the law, fairness in the application of the law, separation of powers, participation in decision-making, legal certainty, avoidance of arbitrariness and procedural and legal transparency.’ (UN SG report 2004).
This article puts this emphasis on how both democracy and the rule of law are intricately linked and mutually reinforcing in a liberal democracy[1], because political scientific, historical and social scientific literature cited below in this article oftentimes refers ‘only’ to democracy (Levitsky and Ziblatt 2018; Stanley 2018; Snyder 2018; Miller-Idriss 2022). Damage to the rule of law is usually left implicit. However, given that this article focuses on criminal law, here, the impact on the rule of law will also be made explicit. As a related point, in this more democracy-focused scholarly work, this form of governance is often juxtaposed with authoritarianism in a manner that is oversimplified, as scholars like Foa and Mounk have criticized.[2] Consequently, this article will explore erosion of liberal democracies by some forms of human trafficking from within and outside without taking a binary democracy versus authoritarian lens. Such care is important because it is also recognized here that liberal democracies no longer get violently overthrown as often but rather ‘erode slowly, in barely visible steps’ (Levitsky and Ziblatt 2018, p. 3). Therefore, the effects on the interplay between the rule of law and democracy in all its complexity must be highlighted. As a final point, this article considers countries around the globe as interconnected, especially insofar as the global economy and financial flows are concerned. Because of globalization and digitization, most countries have become part of a more interwoven, networked web. This also means that negative impact in one state is bound to have an effect on other countries, and that, for example, the transfer of criminally obtained wealth from one nation is bound to influence others.
[1] The indices measuring the quality of democracy—such as those of the Economist Intelligence Unit, Freedom House, Varieties of Democracy, and International IDEA— includes disrespect for the rule of law in addition to crackdowns on civil liberties, populism’s rise, declines in popular trust in politics and political parties, and falling democratic participation.
[2] Roberto Stefan Foa & Yascha Mounk (2021) America after Trump: from “clean” to “dirty” democracy?, Policy Studies, 42:5-6, 455-472, DOI: 10.1080/01442872.2021.1957459.
I also struggle with the idea of a ‘democracy-undermining crime’ – all crimes are dangerous for democracy, are they not? To the extent that criminality undermines the social fabric, trust and (perceptions of) justice, it seems difficult to claim that a criminal act such as human trafficking is dangerous for democracy, while another act, let us say, a robbery, is not. Of course, there is extensive scholarship on ‘militant democracy’, which does rest on the fact that some acts that may (though do not necessarily have to be) criminalized are particularly dangerous for democracy—notably ‘hate speech’, but also manipulation of elections and (depending on the specific scholar) corruption. If the authors intend to pursue this as a core claim (which seems to be the case given the extensive borrowing from the Dutch literature on ‘organized subversive crime’), I would recommend them to embed this claim in the discussion on militant democracy, which is better known as the Dutch sources (largely not accessible in English). In this regard, the claim on the plans to introduce ‘the concept of a crime with the intent to, or effect of, subverting a liberal democracy’ in the Netherlands (l. 142-144) is puzzling. I am not familiar with the Dutch debate (and most readers of the journal cannot be expected to be), but it seems that such crimes are already in place in ‘militant democracies’ (e.g. prosecution of political party leaders for advocating regime change to an autocracy in their speeches). Some scholars whose work might be particularly relevant are B. Rijpkema (who also looks at the Dutch context of militant democracy), Jan-Werner Müller or András Sajó. There, then, one also needs to think of potential dangers of abuse of categories of crimes (relevant especially in relation to the authors’ proposal for a new crime, e.g. l. 658-666).
Author: Thanks, I have improved this into:
2.1. Conceptual questions about this new Dutch concept of organized subversive crime
What this novel concept of ‘crime that undermines democratic societies governed by the rule of law’ entails is not fully clear yet. The same holds true for organized crime. There is no definitive list of organized crime either. Rather, internationally legally defined are ‘merely’ its notions like an organized criminal group, serious crime, structured group, and criminal proceeds (Article 2 of the United Nations Convention against Transnational Organized Crime (UNTOC)).[1] The UNTOC does refer to crimes like money laundering and corruption, while human trafficking is included in the Protocol thereto (Article 3 (a) Palermo Protocol). Just like with organized crime, this conceptual flexibility allows for social, technological and other changes to be captured in our understanding of the phenomenon as well as for specific local variations within a country or between countries. Nonetheless, generally agreed-upon tenets of organized subversive crime are that, at worst, such criminality erodes public and private institutions and their representatives’ integrity, sometimes even irreparably so (Kruisbergen et al 2021; Tjeenk Willink 2021; Tops and Tromp 2017; Boutellier et al 2019; Huisman and Kleemans 2017). Some scholars argue that, for organized subversive crime, the criminal intent is determining, whereas others rather focus on the crime’s corrosive effects (Eski et al 2021; Boutellier et al 2019). This lacking clarity also results in both its over- and under-use (Ibid). For example, critics argue that politicians abuse this undefined concept to combat drug offences despite widely recognized failures of the ‘war on drugs’ (e.g., Bruijn 2019’s ‘disguised war on drugs’). This dearth of conceptual clarity requires critical reflection on what does and does not constitute organized subversive crime and how to use research findings to determine priorities more generally. More specifically, this article explores one aspect of its conceptual scope by focusing on its potential relevance for human trafficking insofar as its possible corrosive effect on liberal democracies is concerned. While a more familiar research theme may very well be how crimes like ‘hate speech’ or corruption erode democracy and the rule of law, this article explores if, and if so to what extent, human trafficking should also be understood to have this corrosive effect.[2] This raises the question why the Netherlands is developing this new concept of organized subversive crime in the first place.
[1] Article 2 (a) ‘Organized criminal group’ shall mean a structured group of three or more persons, existing for a period of time and acting in concert with the aim of committing one or more serious crimes or offences established in accordance with this Convention, in order to obtain, directly or indirectly, a financial or other material benefit. Article 2 (b) ‘Serious crime’ shall mean conduct constituting an offence punish able by a maximum deprivation of liberty of at least four years or a more serious penalty. Article 2 (c) ‘Structured group’ shall mean a group that is not randomly formed for the immediate commission of an offence and that does not need to have formally defined roles for its members, continuity of its membership or a developed structure. There is no definitive list of organized crime, although human trafficking is included as defined in the Protocol to Prevent, Suppress and Punish Trafficking in Persons, Especially Women and Children to the United Nations Convention on Organized Crime (Article 3 (a) Palermo Protocol).
[2] The author thanks the two anonymous reviewers for their remarks that ‘hate speech’ and corruption crimes are more familiar to the audience of this journal.
The Conclusion would benefit from addressing more clearly why a new type of crime would automatically (as that seems to be the argument) help enforcement, i.e. whether the focus should not instead be on implementation of the existing legal framework and in particular on judicial capacities/resources in this regard. The logic for why a new crime would be more conducive to democracy than, for example, adding an aggravating circumstance to human trafficking on the basis of its potential to undermine democracy is not qualified (and, once again, all crimes might be seen as having this potential). Generally, identifying the limitations of the study and some prospects for further research would help improve the conclusion in my view, to leave the reader with more takeaways.
Author: Thanks, I have improved this into:
5. In fact and fiction, from within and outside: The subversion of democracy and the rule of law
5.1. Introduction
The previous sections have demonstrated how human trafficking contributes to the subversion of liberal democracies, in fact and fiction, and from within and outside. For example, actual human trafficking – as exemplified by the Epstein case and confirmed by general facts and global data concerning human trafficking more generally – advances a sense of injustice and a loss of trust by the general public in democracy and the rule of law. Fictional human trafficking in the form of conspiracies equally adds to the assaults on liberal democracies due to the fundamental attacks on the truth, mobilization for violence, and risks for gradually and insidiously introducing totalitarianism. Fictions about human trafficking can even result in real-world consequences pertaining to actual human trafficking, such as the recent voting against anti-human trafficking legislation by a group of 20 US Republicans many of whom are well known for their spreading of conspiracies (the Frederick Douglass Trafficking Victims Prevention and Reauthorization Act was passed on 28 July 2022 in a 401-20 vote).[1] While some of these facts of and fictions about human trafficking erode a liberal democracy from within[2], other attacks rather come from outside. Such external erosion helps to promote autocracy at home by claiming, in short, that liberal democracy is a sham.[3] Worse yet, as much as fact and fiction interact, it is no longer as easy to distinguish between external and internal attacks on liberal democracies because of the involvement of Western enablers (e.g., Guriev and Treisman 2022; Miller-Idriss 2022; Snyder 2018; Stanley 2018; Vogl 2022). This leaves us with many questions as to whether or not those living in liberal democracies want to ensure they contribute to democracy and the rule of law and worldwide equality or rather to autocratic and oligarchical rule. So, how should we respond, both in actuality and through research in academia?
The remainder of this article argues that there are instances in which it would be helpful to charge and conduct research into this new Dutch concept of organized subversive crime for both human trafficking and other such offenses as well as their corrosive effect on liberal democracies, as follows.
5.2. The usefulness of the new Dutch concept of organized subversive crime for law
This article, as explained in the introduction, emphasizes a criminal law perspective. Criminal law should, because of its usually most significant impact on human rights norms and standards, be used as a last resort (ultimum remedium). Nonetheless, the preceding sections gave an initial indication as to how important it is to be able to specifically charge and conduct research into crimes that have the intent to, or effect of, subverting liberal democracies in fact and fiction, from within and outside.
The examples mentioned above demonstrate how some criminals like human traffickers go beyond ‘only’ exploiting state structures for financial or other material gain but rather undermine them. For example, some rather intend to, or commit actions or omissions that result in, damage to democracy and the rule of law. They appear to do so to make more money, not get investigated or prosecuted, or make citizens lose trust in public and private institutions as well as their representatives so as to gain more power over or than the state. While there have been calls in the US to respond to such actions with a to-be-introduced domestic terrorism charge, it does not seem that all those who we would need to hold accountable seek to terrorize an entire population or impose their will on government or an international organization. Instead, at least some of them do subtle and more insidious damage to democracy and the rule of law. Therefore, it is important that, in criminal law, a further distinction with terrorism – whether international or domestic terrorism – is made.[4] To explain this position on terrorism further, it is important to note that there is no universally agreed upon definition of terrorism. However, the only treaty that does define terrorism helps to further explain both the acts and intent required: an unlawful act ‘intended to cause death or serious bodily injury to a civilian, or to any other person not taking an active part in the hostilities in a situation of armed conflict, when the purpose of such act, by its nature or context, is to intimidate a population, or to compel a government or an international organization to do or to abstain from doing any act.’[5] This definition fits poorly with criminals who do not seek to intimidate the general population or coerce the government or an international organization into an act or omission by way of an outright act of violence but rather seem to gradually erode trust in public and private institutions and their representatives. Whereas some may have attacked the US Capitol for the purpose of halting the certification of the election, others may have created the overall climate in the background so that the attackers would do such outright harm to democracy and the rule of law.
So, it seems that it would be useful to investigate criminals for the specific conduct of intending to, or acting in a way that results in, erosion of democracy and the rule of law. Especially for criminals who commit crimes that are also fundamental rights violations, like human trafficking or other offenses like crimes against life (murder and manslaughter) or against physical integrity (torture and inhumane and degrading treatment), it may thus be helpful to introduce a new research concept and criminal charge: organized subversive crime.
As a first example, a criminal who ordered or executed the murder of a well-known member of the free press who investigates organized crime like the aforementioned Mr. Peter R. de Vries causes the death of someone in this profession that is essential for a democratic society. This can have a chilling effect on others in this same profession and thus erode democracy and the rule of law. The same holds true for the above-mentioned assassinations of defense lawyer Mr. Derk Wiersum and potentially the brother of the insider witness if committed out of retaliation for his brother testifying in an international drug trial. Equally, it may be helpful to specifically label trafficked ‘hitters’ or ‘spotters’ on these murders as engaged in organized subversive crime. Comparably, other forms of human trafficking including in sex and labor sectors or for the purpose of organ removal can constitute organized subversive crime as well. Similarly to those actual examples of human trafficking and related crimes, criminals who spread conspiracies about sex trafficking with the intent or effect of damaging democracy and rule of law who thereby incite (mob) violence, could also fall in this category of organized subversive crime. To clarify this point, in liberal democracies, some degree to spreading of conspiracies will have to be permitted out of respect for free speech. However, where such conspiracies spill over in criminal conduct in the form of (mob) violence that line between legal and illegal conduct can be drawn. As a final example in this non-exhaustive list, organized criminals who were involved in the storming of the US Capitol could perhaps better be charged with a newly introduced organized subversive crime of conspiring[6] to damage democracy and the rule of law. This seems to better fit some of the crimes leading up to or committed on January 6th, 2021, than introducing another new charge of terrorism, which is non-existent at the moment as well because America has not ensured enforcement of domestic terrorism yet either.[7] This is especially important since reportedly more than 38,000 Oath Keepers members in the United States, more than 470 of them work in law enforcement or are members of the military.[8] This makes it complicated to draw the dividing line between law enforcement and military as public institutions on the one hand, and those involved in the erosion of democracy and the rule of law, on the other.
As will have been noted, this new concept of organized subversive crime requires that crimes are committed with the intent to, or effect of, damage to democracy and the rule of law. Both the subjective and objective elements of this crime as well as the threshold of what constitutes damage that erodes democratic and rule-of-law institutions or their representatives’ integrity will have to be determined on a case-by-case basis. Nonetheless, in this manner, criminalizing organized subversive crime does help draw the historical record on criminals who sought to subvert democracy and the rule of law; centralize truth finding which has become increasingly relevant in our day and age with all these attacks on the truth; and form the accurate basis for restoration including redress for victims among whom their next of kin and society at large.
Of course, a charge of organized subversive crime could be brought alongside or in addition to terrorism crimes or crimes committed with terrorist intent. But the latter two crimes are specific in their aim at wanting to terrorize the general population for a political motive, on the one hand, or seeking the much more subtle and hidden creeping poison of organized subversive crime, on the other.
5.3. The usefulness of the new Dutch concept of organized subversive crime for research
Even if legislators do not (yet) introduce this new crime or aggravating circumstance, it remains relevant to conduct research into crimes that undermine democratic societies governed by the rule of law. One of the most effective ways to conduct such research is by identifying especially the larger, more hidden criminal networks and take the profit out of this crime. In other words, it helps to follow the financial trails human traffickers leave behind (Coster van Voorhout 2020; TIP report 2022). By following the money, it becomes possible to examine erosion of democracy and the rule of law.[9] Such an emphasis on erosion helps, because, as Snyder mentioned, ‘(e)rosion reveals what resists, what can be reinforced, what can be reconstructed, and what must be reconceived.’ (Snyder 2018, p. 13). This also means introducing greater transparency in liberal democracies so that the exploiters of banking secrecy, tax havens, intelligence networks and organized crime can be detected.[10] Such research into what institutions get eroded through organized subversive crime should be done with due regard for history.[11] Historically, human trafficking is relevant in a twofold manner. As long as education does not correctly address the history of slavery and indeed the imperial histories or slave-based histories of most Western states, it allows for the twofold problems of racism and human trafficking. As Eric Williams explains: ‘[slavery was] basically an economic phenomenon. Slavery was not born of racism; rather, racism was the consequence of slavery.’ (Williams 1944, pp. xi-xii).[12] For example, we still notice the morphing of justifications of slavery into language that hides that connotation such as, in the context of the United States, the reference to state rights as well as the perpetuation of a nondemocratic system with an electoral college and with giving two senators to each state (Lepore 2018). In Europe, although states differ, Germany, France, Britain, Italy, the Netherlands, Spain, and Portugal have all had significant slave trade and slavery pasts before integrating their empires into the European Union, and in all these countries populist antimigration rhetoric appears to be on the rise. These are also exactly the weak points that conspiracists and autocrats appear to abuse, for example in referring to the false so-called ‘great reset’ in which a supposed world elite would want to replace the white race through mass migration, which has already led to actual violence in Germany, for example.[13] Plus, this lack of a real historical understanding is the reason why many argue that it is better to refer not to human trafficking but rather to modern slavery. It is a reasonable question whether slavery and the slave trade have morphed into a new form[14] given the increasing world population and social and economic inequality which makes more people vulnerable to being trafficked or, in the words of Kevin Bales, ‘disposable’.[15]
5.4. Conclusion
A good response to human trafficking in all its complexities will require a whole-of-society response by many liberal democracies acting in concert. Revisiting the previous example of Russia, seeing the significant benefits to the Russian labor economy that human trafficking provides, President Putin is unlikely to address the problem in the near future. So this also means that human trafficking is another fault line for oligarchies and liberal democracies. This leads to the following conclusion that research into crimes that undermine democratic societies governed by the rule of law is indeed urgently required, as follows.
[1] Also, the other way around, the Epstein case still translates into many (QAnon) conspiracies including about not only his suicide – also because Mr. Brunel’s suicide similarly happened by way of hanging with bed linen as Mr. Epstein’s – but also how the Clintons – rather than former president Trump or others – are supposedly connected thereto.
[2] Leaders of Western democracies have done little to guard against the backflow of ‘freed capital flows, deregulated business, and opened trade with their former adversaries.’ (Guriev and Treisman 2022, p. 209).
[3] Cf. Sergei Guriev and Daniel Treisman who explain how the new class of autocrats which they call ‘spin dictators’ because, just like conspiracies, they have changed into a more subtle and insidious group of dictators who use spin rather than the less sophisticated propaganda and force from previous centuries, ‘recruit and corrupt Western elites much as they co-opt their own educated class.’ (Guriev and Treisman 2022, p. 147). ‘Today’s spin dictators turn Tito’s double game into an art. They participate in Western institutions in order to extract benefits, exploiting the design flaws and weaknesses of these bodies. They trade with Western countries, while denouncing them. They recruit networks of corrupt partners in the West, simultaneously pursuing concrete goals and eroding Western cohesion. At the same time, they make hypocritical speeches about the West’s hypocrisy.’ (Guriev and Treisman 2022, p. 152).
[4] In the absence of a universally agreed definition of the term, various terminology describing the notion of "terrorism" can be found within its outputs. One notable exception though is the example discussed here of article 2 of the International Convention for the Suppression of the Financing of Terrorism of 1999.
[5] Article 2 (b) of the International Convention for the Suppression of the Financing of Terrorism of 1999.
[6] Intended in common law terms; in the Netherlands this would be membership of an organized crime group because, in short, civil law countries in these instances place more emphasis on conduct than intent (the objective element also known as actus reus rather than the subjective element also known as mens rea), whereas in common law countries generally the subjective element takes priority.
[7] The United States has a criminal definition of domestic terrorism ex title 18, section 2331 of the United States Code, but has no specific domestic terrorism laws for its enforcement. Per that section, he term “domestic terrorism” means activities that— (A) involve acts dangerous to human life that are a violation of the criminal laws of the United States or of any State; (B) appear to be intended— (i) to intimidate or coerce a civilian population; (ii) to influence the policy of a government by intimidation or coercion; or (iii) to affect the conduct of a government by mass destruction, assassination, or kidnapping; and (C) occur primarily within the territorial jurisdiction of the United States; and (6) the term “military force” does not include any person that— (A) has been designated as a— (i) foreign terrorist organization by the Secretary of State under section 219 of the Immigration and Nationality Act (8 U.S.C. 1189); or (ii) specially designated global terrorist (as such term is defined in section 594.310 of title 31, Code of Federal Regulations) by the Secretary of State or the Secretary of the Treasury; or (B) has been determined by the court to not be a “military force”. (Added Pub. L. 102–572, title X, §â€¯1003(a)(3), Oct. 29, 1992, 106 Stat. 4521; amended Pub. L. 107–56, title VIII, §â€¯802(a), Oct. 26, 2001, 115 Stat. 376; Pub. L. 115–253, §â€¯2(a), Oct. 3, 2018, 132 Stat. 3183.)
[8] Anti-Defamation League, ‘The Oath Keepers Data Leak: Unmasking Extremism in Public Life’, 9 June 2022, available at: https://www.adl.org/resources/report/oath-keepers-data-leak-unmasking-extremism-public-life.
[9] Even in the Mueller investigation into the interference with the US election by Russia no structural financial investigation took place; cf. one of its prosecutors Andrew Weissman: ‘In this investigation, that tenacity was as much an asset as a curse: The inability to chase down all financial leads, or to examine all crimes, gnawed at me, and still does.’ (Weissman 2020, p. 264).
[10] Cf. Guriev and Treisman 2022 and Vogl 2022.
[11] Cf. Eric Williams who states: ‘(t)hese economic changes are gradual, imperceptible, but they have an irresistible cumulative effect. Men, pursuing their interests, are rarely aware of the ultimate results of their activity. The commercial capitalism of the eighteenth century developed the wealth of Europe by means of slavery and monopoly. But in so doing it helped to create the industrial capitalism of the nineteenth century, which turned round and destroyed the power of commercial capitalism, slavery, and all its works. Without a grasp of these economic changes the history of the period is meaningless.’ (Williams 1944, p. 199). Given the emphasis on the Netherlands, it is important to note that, according to Eric Williams, the Dutch East India Company VOC made five thousand percent in profits, and that the profits from the slave trade were smaller than those made by the British East India company (Williams 1944, p. 33).
[12] Cf. Eric Williams: ‘(t)he features of the man, his hair, color and dentifrice, his ‘subhuman’ characteristics so widely pleaded, were only the later rationalizations to justify an economic fact: that the colonies needed labor and resorted to Negro labor because it was cheapest and best.’ (Williams 1944, p. 17). Cf. Jill Lepore citing Ohio Democrat Thomas Morris who in 1839 held the fiercest antislavery speech on the US Senate floor yet: ‘(b)orrowing from the Jacksonian indictment of the ‘money power,’ he coined the phrase ‘slave power.’ Morris described the struggle as a battle between democracy and two united aristocracies: ‘the aristocracy of the North,’ operating ‘by the power of a corrupt banking system’, and ‘the aristocracy of the South,’ which operated “by the power of the slave system.’ (Lepore 2018, p. 224).
[13] Anna Sauerbrey (2016), Germany caught between two violent extremes. Retrieved from: https://www.nytimes.com/2016/07/28/opinion/germany-caught-between-two-violent-extremes.html.
[14] Cf. Jill Lepore who explains ‘slavery seemed like a monster that, each time it was decapitated, grew a new head.’ (Lepore 2018, p. 318).
[15] Cf. Eric Williams: ‘(t)he hope has been expressed that the white servants were spared the lash so liberally bestowed upon their Negro comrades. They had no such good fortune. Since they were bound for a limited period, the planter had less interest in their welfare than in that of the Negroes who were perpetual servants and therefore ‘the most useful appurtenances’ of a plantation.’ (Williams 1944, p. 14). But there is an important difference insofar as the legal protections for lifelong enslavement are concerned: ‘Defoe bluntly stated that the white servant was a slave. He was not. The servant’s loss of life was of limited duration, the Negro was slave for life.’ (Williams 1944, p. 15)
The methodology would need improvement, particularly in terms of using more established and clear-cut presentation of the methods used in the manuscript. While this is open to interpretation, in my view the manuscript is essentially a doctrinal study with some traces of political theory and conceptual analysis. This is perfectly fine; however, the authors claim to be using ‘empirical’ methods in addition to normative, attributing the question ‘what it is’ to the former (p. 2, l. 27). This is not sustainable in my view, as empirical methods also address how and why questions, and there is no trace of empirical methods used in the manuscript (the closest here would be a case study of the Netherlands, but if that is the authors’ intention, then a restructuring of the manuscript would be necessary to have a section explicitly engaging with the Dutch case). In fact, the conclusion, which is almost completely normative, further underscores that empirical methods have not been used.
Author, again, many thanks, I have improved this into:
1.1. Methodology and justification of substantive and geographical focus
The methodology used to answer this central research question is a combination of normative (‘how should it be?’) and, to a limited extent where taken from other predominantly doctrinal sources, empirical research methods (‘what is it?’), while social scientific, historical, and political scientific literature is referred to where relevant. One reason for stressing the importance of needing to obtain empirical findings in future research is the criminological research finding that police interventions that are founded only in theoretical assumptions risk contributing to strengthening criminal networks instead of weakening them (e.g., Luna-Pla and Nicolás-Carlock 2020; Thurner et al 2018; De Domenico et al 2019; Helbing et al 2015; Van Engers et al 2017; Duijn et al 2014). The predominantly doctrinal analysis in this article points out ways to find useful empirical insights in further research. Moreover, the substantive emphasis of this article is on criminal law. Geographically, the focus is on the Netherlands, and in a more limited manner on other European states, the United States, and Russia. Because of the limited scope of this article; other fields of law and countries will have to be reserved for future research.[1] The reasons for this geographical focus are explained further below.
1.2.1. Further justification of geographical focus
This article leads with the Netherlands because scholars and policymakers in that country seek to address a new challenge of crime that erodes its established liberal democracy. The Netherlands did not (yet) face an outright violent attack on its democracy and rule of law such as the storming of the Capitol on January 6th, 2021, in the United States. However, just like the Americans, the Dutch struggle with finding adequate solutions to damage done by crime to those essential tenets of their state, inter alia, because of high-impact murders. While American scholars and policymakers call for dealing with January 6th by introducing a not yet existent domestic terrorism charge, the Dutch are planning on establishing the concept of a crime with the intent to, or effect of, subverting a liberal democracy.[2] This article therefore explores to what extent this solution sought by the Dutch could be helpful in other contexts such as in the United States. This article does so, by focusing on a crime that has not been fully examined in relation to this new Dutch concept either: human trafficking. This examination of the relevance of this new Dutch concept for human trafficking is twofold. First, this article explores whether this novel Dutch concept should or should not encompass human trafficking. Second, the study determines whether a focus on human trafficking can explain the more systemic issues of this new concept in the broader sense. Turning as a final issue to the third country included below, Russia, this country is mentioned because of its export of conspiracies including about human trafficking to liberal democracies (e.g., Snyder 2018; Miller-Idriss 2022). While the emphasis is on Russia because of this spreading of disinformation, this state is also relevant in relation to the crime of human trafficking itself. For example, on 19 July 2022, Russia was put on the state-sponsored human trafficking list, thus indicating an often-overlooked aspect of trafficking in persons not committed by organized criminal groups but by the state itself (TIP report 2022).
[1] This means that this article cannot delve into the entire range of democracies including its most populated, India, and the many others in North America (Canada), western Europe, Central and Eastern Europe, Latin America and the Caribbean, Asia and Australasia, the Middle East and North Africa, and Sub-Saharan Africa. See section 6.
[2] Another possibility could also be to charge the murder – and or in the future, potentially the person or persons who gave the order – with a crime that has been committed with ‘terrorist intent’. The latter is, unlike in the United States, in the Netherlands not reserved for foreign, international terrorism only.
Related to the above, there is no justification given for the extensive references to the Dutch context and resources in the manuscript. If the intensity of the use of these sources is to be retained, alongside the ambition to make a general point, rather than one specific to the Dutch legal system, I think some justification is needed for the extensive use of these sources. More broadly, the choice of focus on the jurisdictions (p. 2, l. 34-36) is also not justified: it is clear that only some can be considered for space reasons, but why these ones in particular? Is it because there is a more vivid discourse on human trafficking in them (which would need to be demonstrated), or because they are established democracies (in which case Russia would obviously not work for the sample) or something else?
Author: Thanks, I have improved this in the above-cited 1.2.1. Further justification of geographical focus and also the corresponding further text such as pertaining to Russia in the fully rewritten sections 3 and 5.
Below are a few more specific comments labelled according to the lines in the document.
- There are quite a few quotes by ‘Tjeenk Willink 2021’, but this source is missing from the reference list. A review of the compatibility of the sources cited and the source list at the end of the manuscript seems necessary.
Author: Thanks, the source was missing and has now been correctly added! My apologies!
- ‘20 Republicans of the Frederick Douglass Trafficking Victims Prevention and Reauthorization Act which was passed on 28 July 2022 in a 401-20 vote.’ This segment appears twice in the text (n. 32, first paragraph of sec. 5).
Author: Thanks, it is now only mentioned once:
5.1. Introduction
The previous sections have demonstrated how human trafficking contributes to the subversion of liberal democracies, in fact and fiction, and from within and outside. For example, actual human trafficking – as exemplified by the Epstein case and confirmed by general facts and global data concerning human trafficking more generally – advances a sense of injustice and a loss of trust by the general public in democracy and the rule of law. Fictional human trafficking in the form of conspiracies equally adds to the assaults on liberal democracies due to the fundamental attacks on the truth, mobilization for violence, and risks for gradually and insidiously introducing totalitarianism. Fictions about human trafficking can even result in real-world consequences pertaining to actual human trafficking, such as the recent voting against anti-human trafficking legislation by a group of 20 US Republicans many of whom are well known for their spreading of conspiracies (the Frederick Douglass Trafficking Victims Prevention and Reauthorization Act was passed on 28 July 2022 in a 401-20 vote).[1]
[1] Also, the other way around, the Epstein case still translates into many (QAnon) conspiracies including about not only his suicide – also because Mr. Brunel’s suicide similarly happened by way of hanging with bed linen as Mr. Epstein’s – but also how the Clintons – rather than former president Trump or others – are supposedly connected thereto.
- L. 711-713 – unlike in the introduction, here the reader is left with the normative question only. It seems necessary to unify the actual question(s) the manuscript tackles between the introduction and the conclusion. In particular, the second question in the introduction is too broad (l. 23-25) and cannot be answered in an article, so focusing just on one question (the need for a separate crime category) might be more effective.
Author, agreed and please see the improved research question (and above the better explanation of the methodology used) in
6. Conclusion and recommendations
This article sought to answer the central research question: Should human trafficking be dealt with as a new and unique category of crime or aggravating circumstance that emphasizes its corrosive effect on liberal democracies and, if so, under what circumstances would that be useful in law and in research? To formulate an answer to this question, the preceding sections have demonstrated that some specific forms of human trafficking can indeed qualify as organized subversive crime. However, this article has also shown that normative and, in future, empirical research, including examinations that improve our historical understanding of disintegration of liberal democracies, can serve as a guide to repair and, hopefully, more proactively as an agenda for its prevention. Therefore, it indeed seems helpful that the Dutch have pioneered the concept of organized subversive crime so as to examine how an offense like human trafficking advances the erosion of liberal democracy or, in this interconnected world, several liberal democracies. For that, human trafficking first has to be detected, and fact has to be distinguished from fiction. After detection, we can build resilience and ensure prevention by especially focusing on how to make liberal democracies more democratic and strengthen the rule of law. For the rule of law, this means the interplay of civil, administrative and criminal law with due consideration to international human rights norms and standards. In this dynamic, criminal law has to particularly focus on holding accountable those who subvert liberal democracies, in fact and fiction, and from within and outside (or a combination of all such factors).
- L. 60: unclear how oligarchy enters the discussion here.
Author: thanks, improved already above in the fully revised section 3.
- L. 30 & 31, ‘police or other rule-of-law-based interventions’: police interventions might be at odds with the rule of law, equating the two does not work without further justification.
Author: thanks, improved by deleting the "other rule-of-law-based interventions’.
- N. 1 (Pinker): why should there be a source by this author from after 2018? There are other quoted sources which do not come from after 2018.
Author: thanks, improved by However, the author did not find any writings on this topic from Pinker after 2018, who may also have changed his opinion at least somewhat after January 6th, 2021 . Pinker’s language expertise was used by the Epstein defense (Brown 2021, p. 156).
In sum, despite my reservations and opinion that the current form of the manuscript is not publishable, I encourage the authors to proceed with the revisions and develop these interesting ideas further, as there is a potential to connect them to several established threads in the literature on democracy protection.
Author: Many thanks, much appreciated, and I hope that all revisions have indeed had the intended effect along the much appreciated suggestions and helpful feedback from the reviewer.
Round 2
Reviewer 2 Report
The revised version of the manuscript has been improved in some respects in my view. In particular, the justification for the expected growing importance of human trafficking given increased migration due to climate change and Russia’s invasion of Ukraine is a very relevant point (author response, p. 9). The author also addressed the segments on the research question, methodology and the case selection in an, in principle, understandable manner. They also removed the controversial concept of ‘democracy-undermining crime’ (which suggested that not all crimes undermine democracy—possibly the case, but not explicable within the framework of this manuscript) and the (arguably implausible) claim that private exploitation does not undermine democracy.
Limitations of the research and avenues for further research have not been included in my reading. Perhaps there was a reason for doing so, but I could not discern it from the author response.
As for the abstract, the author claims not to intend to reveal too much in the abstract for the reader (author response, p. 5). This is a rather outdated approach to abstract writing, which does not fit to a digital world, where, on the contrary, it is appreciated if the abstract provides a clear overview of the argument which then interested readers may be prompted to engage with in greater detail. I would not appreciate, as a reader, the kind of abstract provided in this article, as it is missing a clear statement and blurs the overall message of the text. Yet, I recognize that there are different perspectives on this point, and the author has provided an explanation for their preference in the author response, which is taken note of.
A few more specific comments:
- ‘They worry that the anger, frustration, and cynicism of the general public about democracy and the rule of law is corroding the moral and social foundation of liberal democracies (Ibid.)’ (author response, p. 5) – I would suggest removing this sentence altogether. It has not become clearer with the small change, and appears unnecessary in the paragraph.
- ‘More specifically, this article explores one aspect of its conceptual scope by focusing on its potential relevance for human trafficking insofar as its possible corrosive effect on liberal democracies is concerned’ (author response, p. 6) – it is unclear what ‘one aspect of its conceptual scope’ refers to. I recommend using specific labels instead of pronouns here.
- I assume that section 3.2.2 was added given another reviewer’s remarks; it does provide some valuable insights, particularly through underscoring the ‘power of money’ that provides quality legal services to the trafficker. Contrary to the earlier suggestion here, however, no figure, clearly depicting the range of actors who may affect the relationship between human trafficking and democracy (governments, banks, private corporations etc.), was included, and no explanation was provided for the non-inclusion. As it is often said, one picture is worth thousand words, and I think it would help streamline the discussion also in Section 3.3., which could then be made considerably shorter and more accessible to the reader (with more integration of the ‘four factors’ explaining the negative effects of human trafficking into the rest of the manuscript). Yet, the author might have had a specific reason against doing so.
- While there is a relevant point referring to the potential role of banks in the process, the ‘causal chain’ to erosion of democracy in relation to human trafficking is still very loose. I would suggest to use more cautious phrasing or to eliminate this paragraph, unless illustrated with more specific cases of involvement of banks in facilitating human trafficking, instead of general reflections on the role of banks in facilitating austerity measures that, in turn, damage trust in democracy (see author response, pp. 12-13).
- While the inclusion of the section on definitions of democracy, liberal democracy and the rule of law is appreciated, the distinction between democracy and liberal democracy the author seems to be making remains unconvincing. I think the problem could be addressed by simply using ‘democracy’ instead of ‘liberal democracy’ throughout the text. However, if accepted in the present form, the article would definitely not be the only one published which struggles with establishing a meaningful distinction between the two concepts (see author response, p. 32).
On the whole, the manuscript does prompt some new thoughts and contains several interesting observations for lawyers and (other) social scientists. It has a greater promise than what it delivers, but it may indeed prompt further research particularly in the context of further erosion of democracy in the wake of the Russian invasion of Ukraine. It could be written in a more compelling (and concise) manner (except a few very well-written segments, as I comment above), but, if the journal does not have a word limit, the choice for a lengthier format more demanding for the reader is, to an extent, the author’s as well.
Author Response
Dear reviewer,
Given that I have decided, upon your valuable feedback, to reverse the order of the sections on conspiracies and actual human trafficking cases (sections 3 and 4), I have decided against using track changes because this would have made the text illegible. Consequently, I have marked the text in red and underlined it, as track changes would do, so that you can see the changes easily. I have also been able to use more sources, such as the book of the prosecutor of Epstein, Geoff Berman, and the work on the January 6th committee by Denver Riggleman whose work are added in new footnotes similarly marked.
Please find the author's response as the attached document, hoping that this would make an easier read for you.
I look forward to your feedback!
Kind regards, the author
